# List-Decodable Mean Estimation
# in Nearly-PCA Time

**Ilias Diakonikolas**
Department of Computer Science
University of Wisconsin, Madison
Madison, WI 53706
ilias@cs.wisc.edu

**Daniel M. Kane**
Department of Computer Science
University of California, San Diego
La Jolla, CA 92093
dakane@cs.ucsd.edu

**Daniel Kongsgaard**
Department of Mathematics
University of California, San Diego
La Jolla, CA 92093
dkongsga@ucsd.edu

**Jerry Li**
Microsoft Research
Redmond, WA 98052
jerrl@microsoft.com

**Kevin Tian** *
Department of Computer Science
Stanford University
Stanford, CA 94305
kjtian@stanford.edu

## Abstract

Robust statistics has traditionally focused on designing estimators tolerant to a minority of contaminated data. *List-decodable learning* [CSV17] studies the more challenging regime where only a minority $\frac{1}{k}$ fraction of the dataset, $k \geq 2$, is drawn from the distribution of interest, and no assumptions are made on the remaining data. We study the fundamental task of list-decodable mean estimation in high dimensions. Our main result is a new algorithm for bounded covariance distributions with optimal sample complexity and near-optimal error guarantee, running in *nearly-PCA time*. Assuming the ground truth distribution on $\mathbb{R}^d$ has identity-bounded covariance, our algorithm outputs $O(k)$ candidate means, one of which is within distance $O(\sqrt{k \log k})$ from the truth.

Our algorithm runs in time $\widetilde{O}(ndk)^2$, where $n$ is the dataset size. This runtime nearly matches the cost of performing $k$-PCA on the data, a natural bottleneck of known algorithms for (very) special cases of our problem, such as clustering well-separated mixtures. Prior to our work, the fastest runtimes were $\widetilde{O}(n^2 dk^2)$ [DKK20], and $\widetilde{O}(ndk^C)$ [CMY20] for an unspecified constant $C \geq 6$. Our approach builds on a novel soft downweighting method we term SIFT, arguably the simplest known polynomial-time mean estimator in the list-decodable setting. To develop our fast algorithms, we boost the computational cost of SIFT via a careful "win-win-win" analysis of an approximate Ky Fan matrix multiplicative weights procedure we develop, which may be of independent interest.

---

*Part of this work was done as an intern at Microsoft Research.

[2] Throughout this work, the $\widetilde{O}$ notation hides logarithmic factors in $n$ and the failure probability.

35th Conference on Neural Information Processing Systems (NeurIPS 2021).

# 1 Introduction

Mean estimation has emerged as one of the cornerstone tasks in robust statistics, as the most basic in a hierarchy of increasingly complex estimation problems. The problem is straightforward to state: given samples from a "nice" ground-truth distribution $\mathcal{D}$, where an adversary has (arbitrarily) corrupted a fraction of the data, recover the mean of $\mathcal{D}$ as accurately as possible. Due to its fundamental nature, robust mean estimation has received extensive study in the statistics, theoretical computer science, and machine learning communities, starting from the 1960s [Ans60, Tuk60, Hub64, Tuk75].

Despite the apparent simplicity of the problem, efficient algorithms achieving near-optimal error rates were not known in high-dimensional settings until recently [LRV16, DKK$^+$19]. These works studied mean recovery in the traditional setting where a majority of the data is "trusted," i.e. the fraction of corruptions is strictly less than $\frac{1}{2}$. For the standard formulation of robust mean estimation, this assumption is necessary. Indeed, if only an $\alpha \leq \frac{1}{2}$ fraction of points can be trusted, the dataset could consist of $O(\frac{1}{\alpha})$ well-separated clusters of "good" points. The mean of each individual cluster is an equally valid solution to the estimation problem, so asking for a single solution is ill-posed.

In many settings of theoretical and practical interest, having a majority of inlier points is too strong of an assumption. To circumvent the issue of well-posedness in the $\alpha \leq \frac{1}{2}$ regime, [CSV17] proposed a relaxed learning notion termed *list-decodable learning*. Rather than restricting to a single hypothesis, the algorithm may output a list of $O(\frac{1}{\alpha})$ hypotheses, guaranteeing at least one of them is close to the truth. In the context of robust mean estimation, this amounts to outputting $O(\frac{1}{\alpha})$ candidate means.

A natural problem in its own right, list-decodable mean estimation also generalizes a number of other problems. A prototypical example is learning well-separated mixture models, which has received extensive treatment in the literature [Das99, VW04, AM05, DS07, AK05, RV17, HL18, DKS18, KSS18]. In this problem, data is drawn from a uniform mixture[3] of $k$ "nice" distributions $\mathcal{D}_1, \ldots, \mathcal{D}_k$ with far apart means, and the goal is to recover clusters coming from each component. By running a list-decodable estimation procedure with $\alpha = \frac{1}{k}$, each cluster is an equally valid "ground truth," so the output must contain candidate means close to each true mean. If the candidates are sufficiently close, standard techniques recover the true clustering. List-decodable mean estimation robustly extends this problem to tolerate adversarial noise or non-uniformity, up to constants in the output size.

Moreover, list-decodable mean estimation can be used to model important data science applications such as crowdsourcing (where a majority of respondents could be unreliable or malicious) [SVC16, MV18], or semi-random community detection in stochastic block models [CSV17]. This primitive is particularly useful in the context of semi-verified learning [CSV17, MV18], where a learner can audit a small amount of trusted data. Even if the trusted dataset is too small to directly learn from, in conjunction with a list-decodable learning procedure it can pinpoint a candidate hypothesis consistent with the verified data (indeed, only roughly $\log \frac{1}{\alpha}$ vetted points are required).

The first tractable algorithm for high-dimensional list-decodable mean estimation was given by [CSV17]. Their work considered the setting where $\mathcal{D}$ has (unknown) covariance $\mathbf{\Sigma} \preceq \sigma^2 \mathbf{I}$ for some known $\sigma$ (i.e. a second moment bound). In this setting, [CSV17] gave a sample-optimal, polynomial-time algorithm outputting $O(\frac{1}{\alpha})$ candidate means, so some candidate is within $\ell_2$ distance $O(\sigma\sqrt{\alpha^{-1}\log\alpha^{-1}})$ from the mean of $\mathcal{D}$. As later demonstrated in [DKS18], this error is optimal up to logarithmic factors. However, the [CSV17] algorithm heavily relies on black-box semidefinite programming solvers, and thus the runtime is prohibitively large in high-dimensional instances.

The goal of our work is to develop much faster, truly scalable algorithms for list-decodable mean estimation which achieve optimal statistical guarantees. This goal fits broadly into a larger line of work focused on understanding the computational cost of robustness for basic statistical tasks. In some settings, this line has demonstrated strong evidence that robustness comes at an inherent computational cost [DKS17, HL19]. In contrast, recent algorithms have been developed that achieve robustness essentially "for free" in many other settings [CDG19, DHL19, CDGW19, LY20, JLT20].

While list-decodable mean estimation has received a fair amount of attention, there have only been a few results achieving improved runtimes. One line of work proposed an algorithm framework termed *multi-filtering* [DKS18, DKK20], based on learning multiple candidate "weight functions." In particular, [DKK20] uses this approach to design an algorithm achieving near-optimal error in time $\widetilde{O}(n^2 d\alpha^{-2})$, where $n$ is the overall dataset size. While this dramatically improves over the runtime

---

[3]Some algorithms extend beyond the uniform setting; we present it this way for simplicity of exposition.

in [CSV17], the quadratic dependence on $n$ is not ideal in high-dimensional settings. Concurrently, the work [CMY20] proposes a different, descent-based algorithm based on (approximate) positive semidefinite programming, achieving optimal error (up to constant factors) in time $\widetilde{O}(nd\alpha^{-C})$ for some constant $C \geq 6$. When $\alpha = \Theta(1)$, the [CMY20] runtime is nearly-linear in the problem input size. However, if $\alpha^{-1}$ scales polynomially with $d$, e.g. a mixture model with many components in moderate dimension, then this large dependence on $\alpha^{-1}$ may also be prohibitively slow.

In contrast to this somewhat murky runtime landscape, the state of affairs for clustering separated mixture models is relatively clear. The fastest known algorithm for clustering a mixture of $k$ well-separated components is almost twenty years old [VW04], and runs in time $\widetilde{O}(ndk)$, as a relatively simple and elegant application of (approximate) $k$-principal components analysis ($k$-PCA). Since list-decodable mean estimation can be thought of as the natural robust analog to clustering mixture models, it is natural to ask:

*Can we perform list-decodable mean estimation as efficiently as learning mixture models?*

Concretely, since clustering mixture models is an instance of list-decodable learning with $\alpha = k^{-1}$, the question becomes: can we solve list-decodable mean estimation in time $\widetilde{O}(\frac{nd}{\alpha})$? This presents itself as a natural barrier, since further improvement would imply faster learning of mixture models.

## 2 Our results

Our main contribution is to answer this question affirmatively with near-optimal statistical guarantees. We start by formally defining the problem we study, which we call "list-decodable mean estimation."

**Definition 1.** *Given $T \subset \mathbb{R}^d$ and $0 < \alpha \leq \frac{1}{2}$ such that an $\alpha$-fraction of $T$ are independent draws from a distribution $\mathcal{D}$ with unknown target mean $\mu^*$ and unknown covariance $\mathbf{\Sigma} \preceq \sigma^2 \mathbf{I}$, output a small list of vectors $\{\hat{\mu}_j\}_{j \in L}$ such that $\min_{j \in L} \|\hat{\mu}_j - \mu^*\|_2$ is minimized.*

We emphasize that no assumptions are made on the remaining $1 - \alpha$ fraction of $T$, i.e. they may be arbitrary and chosen by a computationally unbounded adversary allowed to inspect the inliers. In the setting of Definition 1, our main result states that we can solve list-decodable mean estimation nearly-optimally in the time it takes to perform $k$-PCA poly-logarithmically many times, for $k = \Theta(\frac{1}{\alpha})$.

**Theorem 1** (See Corollary 3 of the supplementary material). *Given $|T| = n = \Omega(\frac{d}{\alpha})$ samples, there is an algorithm with runtime $\widetilde{O}\left(\frac{nd}{\alpha}\right)$ that with high probability outputs a list of $m$ vectors $\{\mu_j\}_{j \in [m]}$, for $m = O(\frac{1}{\alpha})$, such that $\min_{j \in [m]} \|\mu^* - \mu_j\|_2 = O(\sigma \alpha^{-\frac{1}{2}} \sqrt{\log \alpha^{-1}})$.*

A few remarks are in order. First, a list size of $\Omega(\frac{1}{\alpha})$, sample complexity of $\Omega(\frac{d}{\alpha})$, and error of $\Omega(\sigma \alpha^{-\frac{1}{2}})$ are information-theoretically necessary for this problem [DKS18]. We also provide an algorithm in the supplementary material returning $m = O(\frac{1}{\alpha})$ hypotheses $\{\mu_j\}_{j \in [m]}$ attaining optimal error $\min_{j \in [m]} \|\mu^* - \mu_j\|_2 = O(\sigma \alpha^{-\frac{1}{2}})$, with an additive runtime overhead of $\widetilde{O}(\alpha^{-6})$. Without loss of generality $\alpha^{-1} = o(d)$, as otherwise there is a trivial algorithm (cf. Appendix A of the supplementary material), so the $\alpha^{-6}$ additive term is only dominant in the range $\alpha^{-1} \in [\omega(\sqrt{d}), o(d)]$ assuming $n = \Theta(\frac{d}{\alpha})$. Even with this overhead, our runtime is the best-known in all parameter regimes.

Our approach is inspired by the way in which fast algorithms for robust mean estimation in the $\alpha \to 1$ regime were built. First, a "simple" polynomial (but not nearly-linear) time algorithm — namely, the filter — was developed [DKK$^+$19, DKK$^+$17, Ste18]. Then, building on the framework of this simple algorithm, near-linear time algorithms were developed [CDG19, DHL19] using tools from the continuous optimization community — specifically, matrix multiplicative weights regret analyses. In this paper, we accomplish both these steps for the $\alpha \ll \frac{1}{2}$ regime. First, we design a simple "basic" algorithm for the list-decodable mean estimation problem, and then speed it up using matrix regret minimization tools. As explained below, both steps require substantially new ideas.

**SIFT: a new, simple algorithm for list-decodable learning.** Our first main contribution is a novel algorithm for list-decodable mean estimation, which we call SIFT (Subspace Isotropic FilTering),[4]

---

[4]After the initial dissemination of this work, we were made aware there is another (unrelated) algorithm named SIFT developed in the computer vision community [Low99].

| Reference | Runtime | Error guarantee |
|-----------|---------|-----------------|
| [CSV17] | $\text{poly}(n, d, \alpha^{-1})$ | $O(\sigma\alpha^{-\frac{1}{2}}\sqrt{\log\alpha^{-1}})$ |
| [DKK20] | $\widetilde{O}(n^2 d\alpha^{-2})$ | $O(\sigma\alpha^{-\frac{1}{2}}\log\alpha^{-1})$ |
| [CMY20] | $\widetilde{O}(nd(\alpha^{-1})^{\geq 6})$ | $O(\sigma\alpha^{-\frac{1}{2}})$ |
| **Our work** | $\widetilde{O}(nd\alpha^{-1})$ | $O(\sigma\alpha^{-\frac{1}{2}}\sqrt{\log\alpha^{-1}})$ |
| **Our work** | $\widetilde{O}(nd\alpha^{-1} + \alpha^{-6})$ | $O(\sigma\alpha^{-\frac{1}{2}})$ |

Table 1: List-decodable mean estimation algorithm runtimes. All algorithms listed return lists of size $O(\frac{1}{\alpha})$ and use sample complexity $n = O(\frac{d}{\alpha})$, which are information-theoretically optimal [DKS18]. The $\widetilde{O}$ notation hides polylogarithmic factors in failure probability and dimension.

achieving optimal statistical guarantees in time $\widetilde{O}(\frac{n^2 d}{\alpha})$. While by itself, SIFT does not achieve a nearly-linear runtime, its framework is vital in designing our more sophisticated algorithms. Crucially, SIFT is conceptually different from all prior approaches for our problem, with a complete decoupling of a "filtering" step and a "clustering" step; it is these differences that allow for our later speedups.

The main advantage of SIFT is its simplicity. All prior algorithms for list-decodable mean estimation were quite complicated, with involved and lengthy analyses, whereas an analysis of SIFT fits in just over two pages. We believe SIFT is of independent interest (both theoretically and practically), and its framework may find applications in other list-decodable learning settings.

FastSIFT**: speeding up** SIFT **via Ky Fan regret minimization.** While each SIFT iteration can be performed in time $\widetilde{O}(ndk)$, the algorithm requires $\Theta(n)$ iterations in the worst case, as there are instances where each iteration removes only one data point. The main challenge is to combine the SIFT analysis with a downweighting procedure guaranteeing termination in *polylogarithmically* many iterations. To achieve this, we use tools from semidefinite programming (SDP) to design iterative schemes with stronger termination guarantees. This mirrors, and is inspired by, the $\alpha \to 1$ regime, where tools such as packing SDP solvers [CDG19] and matrix multiplicative weights [DHL19] were used to speed up the basic filter. We note that [CMY20] also uses SDP tools to obtain their runtime, but in a substantially different way: they design a black-box Ky Fan norm packing SDP subroutine, whereas we directly use a (stronger) regret guarantee to guide weight removal. Finally, while there have been substantial recent runtime advances for solving SDPs (see e.g. [JKL$^{+}$20]), these guarantees are too weak for our purposes, requiring the development of new approximate SDP tools.

## 3 Preliminaries

**General notation.** We let $\mathcal{N}(\mu, \boldsymbol{\Sigma})$ denote the multivariate Gaussian distribution with specified mean and covariance, and $[d]$ denote the naturals $1 \leq j \leq d$. When applied to a vector argument, $\|\cdot\|_p$ is the $\ell_p$ norm. We denote the (solid) probability simplex in $n$ dimensions by $\Delta^n = \{w \in \mathbb{R}^n_{\geq 0} \mid \|w\|_1 \leq 1\}$. The all-ones vector in appropriate dimension is $\mathbb{1}$. The identity matrix in appropriate dimension is $\mathbf{I}$. We use the standard Loewner order $\preceq$ on the set of symmetric matrices $\mathbb{S}^d = \mathbb{R}^{d \times d}$ with positive semidefinite subset $\mathbb{S}^d_{\geq 0}$. For $k \in [d]$, the operation $\lambda_k(\cdot)$ on $\mathbb{S}^d$ returns the $k^{\text{th}}$ largest eigenvalue. When applied to a matrix in $\mathbb{S}^d_{\geq 0}$, $\|\cdot\|_k$ for $k \in [d]$ is the *Ky Fan* norm (sum of the top $k$ eigenvalues), and $\|\cdot\|_{\text{op}}$ in particular is the Ky Fan 1 norm. The inner product on $\mathbb{S}^d \times \mathbb{S}^d$ is $\langle \mathbf{A}, \mathbf{B} \rangle = \text{Tr}(\mathbf{AB})$.

**Distributions.** Let $T$ be a set of points in $\mathbb{R}^d$ with $|T| = n$, and let $w \in \Delta^n$. For any $T' \subseteq T$, $w_{T'} \in \Delta^n$ is the vector which equals $w$ on coordinates in $T'$, and is zero elsewhere. We refer to the empirical mean and covariance, parameterized by weights $w$ and subset $T' \subseteq T$, by

$$\mu_w(T') := \sum_{i \in T'} \frac{w_i}{\|w_{T'}\|_1} X_i, \ \text{Cov}_w(T') := \sum_{i \in T'} \frac{w_i}{\|w_{T'}\|_1} \left(X_i - \mu_w(T')\right)\left(X_i - \mu_w(T')\right)^{\top}.$$

We denote uniform samples $i$ from a set $S$ by the notation $i \sim S$.

**Deterministic assumption.** In the *list-decodable mean estimation* problem, we are given a set $T$ of $n$ points $\{X_i\}_{i \in T}$ in $\mathbb{R}^d$, under Definition 1.[5] Without loss, we assume $\sigma = 1$, which generalizes appropriately by scaling the dataset by $\sigma^{-1}$. Regarding the sample size $n$, we recall the following (note that the matrix in Assumption 1 is *not* the covariance, as it is centered at the true mean).

**Proposition 1** (Proposition B.1, [CSV17]). *For any constant $\epsilon \in (0, 1)$, there are constants $c, C > 0$ such that with probability at least $1 - \exp(-\Omega(n))$, for $n = \frac{Cd}{\alpha}$, if an $(1 + \epsilon)\alpha$ fraction of points in $\{X_i\}_{i \in T} \subseteq \mathbb{R}^d$ is drawn from $\mathcal{D}$ with covariance bounded by $c\mathbf{I}$, then Assumption 1 holds.*

**Assumption 1.** $\exists S \subseteq \{X_i\}_{i \in T} \subseteq \mathbb{R}^d$ *of size* $\alpha n = \Theta(d)$ *with* $\mathbb{E}_{i \sim S}[(X_i - \mu^*)(X_i - \mu^*)^\top] \preceq \mathbf{I}$.

In the remainder of the paper, we operate under Assumption 1, i.e. that there is a subset $S$ of the dataset whose centered covariance around $\mu^*$ is identity-bounded. Proposition 1 implies that this assumption holds for any failure probability larger than $\exp(-\Omega(d))$ (up to constants in parameter definitions); for any smaller failure probability, Proposition 1 implies that the assumption still holds by adjusting the sample size by a logarithmic factor. We also assume $\frac{1}{\alpha} \leq d$ for simplicity, and handle the regime $\frac{1}{\alpha} = \Omega(d)$ with a much simpler algorithm in Appendix A of the supplemental material.

Finally, $k$ will be reserved for values which are $\Theta(\frac{1}{\alpha})$ for explicitly stated constants. In particular, our algorithms will use operations such as principal components analysis in $\Theta(\frac{1}{\alpha})$ dimensions. This is because a substantial portion of the challenge is reducing to the problem of learning the mean in $\Theta(\frac{1}{\alpha})$ dimensions, at which point naïve random sampling suffices (up to logarithmic factors).

**Organization.** In this shortened version of our paper, we give a full statement and analysis of our "slow algorithm" SIFT in Section 4, achieving optimal sample complexity and error, at a logarithmic overhead in list size and an $O(n)$ overhead in runtime compared to our final method. We then provide a detailed outline of our techniques to improve this result in Section 5, focusing on the statement of key technical ingredients developed in this work. Due to space constraints, we defer a more rigorous treatment of proofs, as well as an extended exposition of our techniques and comparison to the prior work on this problem, to an unabridged version in the supplementary material.

# 4 Warmup: SIFT **algorithm and analysis**

We develop a conceptually new and simple algorithm for solving list-decodable mean estimation based on a "soft downweighting" approach, which forms the backbone of our faster methods. In brief, its design philosophy can be summarized as follows: first learn the mean in all but $\Theta(\frac{1}{\alpha})$ directions, and then randomly sample in the remaining subspace. All omitted proofs are either straightforward or from prior work, and are given in Section 3 of the supplementary material. We define two useful concepts in analyzing our downweighting methods.

**Definition 2** (Saturated weights). *We call $w \in \Delta^n$ "saturated" if $w \leq \frac{1}{n}\mathbb{1}$ and $\|w_S\|_1 \geq \alpha\sqrt{\|w\|_1}$.*

**Definition 3** (Safe scores). *We call scores $\{\tau_i\}_{i \in T} \in \mathbb{R}_{\geq 0}^n$ "safe with respect to $w \in \Delta^n$" if $\sum_{i \in S} \frac{w_i}{\|w_S\|_1} \tau_i \leq \frac{1}{2} \sum_{i \in T} \frac{w_i}{\|w\|_1} \tau_i$. When $w$ is clear from context, we simply call $\tau$ "safe".*

In algorithms based on soft filtering in the presence of a small amount of adversarial noise (e.g. [DKK+17, Li18, Ste18]), a typical goal is to iteratively remove more "good weight" than "bad weight." However, when the overwhelming majority of the initial weight is bad, this is too strong of a goal. The intuition for Definition 2 is that a weaker goal suffices for our guarantees; while the good weight decreases throughout, Definition 2 requires that the good weight becomes more saturated when more weight is removed. We now make the connection between these definitions formal.

**Lemma 1.** *Let weights $w$ be saturated. Then $w'$ is also saturated, where $w'$ is the result of the following update: let $\{\tau_i\}_{i \in T}$ be safe with respect to $w$, and update for all $i \in T$:*

$$w'_i \leftarrow \left(1 - \frac{\tau_i}{\tau_{\max}}\right) w_i, \text{ where } \tau_{\max} := \max_{i \in T | w_i \neq 0} \tau_i. \tag{1}$$

We next give three helper lemmas which reason about how the quality of empirical estimates based on $S$ deteriorate, as weight allocated to $S$ is reduced. The first shows how the quality of the empirical mean relates to the empirical covariance and proportion of weight in $S$.

---

[5]Abusing notation, we let $T$ denote both the point set and indices, interchangeably using $X_i \in T$ and $i \in T$.

**Lemma 2.** *If $w \in \Delta^n$ has $w \leq \frac{1}{n}\mathbb{1}$ entrywise, $\|\mu_w(T) - \mu^*\|_2 \leq \sqrt{2 \|\mathrm{Cov}_w(T)\|_{\mathrm{op}} \frac{\|w\|_1}{\|w_S\|_1}} + \frac{2\alpha}{\|w\|_1}$.*

The second shows how the empirical covariance of $S$ grows relative to how much of $S$ is kept.

**Lemma 3.** *Let $w \in \Delta^n$ have $w \leq \frac{1}{n}\mathbb{1}$ entrywise. Then $\mathrm{Cov}_w(S) \preceq \frac{\alpha}{\|w_S\|_1}\mathbf{I}$.*

The third shows how a bound on the saturation of $S$ in a weight vector can be used to bound the distance between empirical means in $S$ and $T$ via the weighted empirical covariance matrix.

**Lemma 4.** $(\mu_w(S) - \mu_w(T))(\mu_w(S) - \mu_w(T))^\top \preceq \frac{\|w\|_1}{\|w_S\|_1}\mathrm{Cov}_w(T)$.

We now present SIFT as Algorithm 1. It requires calls to an approximate $k$-PCA subroutine Power, the classical simultaneous power iteration method, whose guarantees we state here.

**Proposition 2** (Theorem 1, [MM15]). *For any $\delta \in (0,1)$ and $k \in [d]$, there is an algorithm, Power, which takes as input $k$, $\delta$, $\mathbf{A} \in \mathbb{S}_{\geq 0}^d$ and $\epsilon \in (0,1)$, and returns with probability $1 - \delta$ a set of orthonormal vectors $\mathbf{V} \in \mathbb{R}^{d \times k}$ such that if $\mathbf{V}_{:i}$ is column $i$ of $\mathbf{V}$,*

$$\langle \mathbf{V}_{:i}, \mathbf{A}\mathbf{V}_{:i} \rangle \in [1 \pm \epsilon]\lambda_i(\mathbf{A}) \text{ for all } i \in [k], \; \left\|(\mathbf{I} - \mathbf{V}\mathbf{V}^\top)\mathbf{A}(\mathbf{I} - \mathbf{V}\mathbf{V}^\top)\right\|_{\mathrm{op}} \leq (1 + \epsilon)\lambda_{k+1}(\mathbf{A}).$$

*When $\mathbf{A}$ is given in the form $\mathbf{M}^\top\mathbf{M}$ for some $\mathbf{M} \in \mathbb{R}^{n \times d}$, the runtime of Power is $O\left(\frac{ndk}{\epsilon}\log\left(\frac{d}{\delta\epsilon}\right)\right)$.*

---

**Algorithm 1** SIFT$(T, \delta)$

---

1: **Input:** $T \subset \mathbb{R}^d$ with $|T| = n$ satisfying Assumption 1, $\delta \in (0,1)$
2: $w^{(0)} \leftarrow \frac{1}{n}\mathbb{1}_T, t \leftarrow 0, \beta \leftarrow 1, k \leftarrow \lceil\frac{4}{\alpha}\rceil$
3: $\mathbf{V} \leftarrow \mathsf{Power}(\mathrm{Cov}_{w^{(t)}}(T), k, 0.2, \frac{\delta}{2n})$
4: **while** $\lambda_k(\mathbf{V}^\top \mathrm{Cov}_{w^{(t)}}(T)\mathbf{V}) \geq \frac{4}{\sqrt{\beta}}$ **do**
5: $\quad \tau_i^{(t)} \leftarrow \left\|\mathbf{\Sigma}^{-\frac{1}{2}}\mathbf{V}^\top(X_i - \mu_w(T))\right\|_2^2$ for all $i \in T$
6: $\quad w_i^{(t+1)} \leftarrow \left(1 - \frac{\tau_i^{(t)}}{\tau_{\max}^{(t)}}\right)w_i^{(t)}$ for all $i \in T$, where $\tau_{\max}^{(t)} := \max_{i \in T | w_i^{(t)} \neq 0} \tau_i^{(t)}$
7: $\quad t \leftarrow t + 1, \beta \leftarrow \|w^{(t)}\|_1$
8: $\quad \mathbf{V} \leftarrow \mathsf{Power}(\mathrm{Cov}_{w^{(t)}}(T), k, 0.2, \frac{\delta}{2n})$
9: **end while**
10: **return** $L := \{\mathbf{V}\mathbf{V}^\top X_i + (\mathbf{I} - \mathbf{V}\mathbf{V}^\top)\mu_{w^{(t)}}(T)$ where $i \in T$ is sampled uniformly at random$\}$, with list size $|L| = \lceil\frac{2}{\alpha}\log\frac{2}{\delta}\rceil$

---

As Lines 4 through 9 of Algorithm 1 constitute a weight removal method of the form given in (1), we now demonstrate the scores used are safe, which implies that $w^{(t)}$ is always saturated.

**Lemma 5.** *In each iteration $t$ of Algorithm 1 until termination, $\tau^{(t)}$ is safe with respect to $w^{(t)}$.*

*Proof.* Throughout this proof, let $w := w^{(t)}$ and $\tau := \tau^{(t)}$. Furthermore, let $\mathbf{V}$, $\beta$ correspond to the weights $w^{(t)}$, and let $\mathbf{\Sigma} := \mathbf{V}^\top\mathrm{Cov}_{w^{(t)}}(T)\mathbf{V}$. We first compute the average score in $S$:

$$\sum_{i \in S} \frac{w_i}{\|w_S\|_1}\tau_i = \sum_{i \in S} \frac{w_i}{\|w_S\|_1}\left\|\mathbf{\Sigma}^{-\frac{1}{2}}\mathbf{V}^\top(X_i - \mu_w(T))\right\|_2^2$$

$$= \sum_{i \in S} \frac{w_i}{\|w_S\|_1}\left(\left\|\mathbf{\Sigma}^{-\frac{1}{2}}\mathbf{V}^\top(X_i - \mu_w(S))\right\|_2^2 + \left\|\mathbf{\Sigma}^{-\frac{1}{2}}\mathbf{V}^\top(\mu_w(S) - \mu_w(T))\right\|_2^2\right)$$

$$= \left\langle \mathbf{\Sigma}^{-1}, \mathbf{V}^\top\mathrm{Cov}_w(S)\mathbf{V}\right\rangle + \left\|\mathbf{\Sigma}^{-\frac{1}{2}}\mathbf{V}^\top(\mu_w(S) - \mu_w(T))\right\|_2^2$$

$$\leq \left\langle \mathbf{\Sigma}^{-1}, \frac{\alpha}{\|w_S\|_1}\mathbf{I}\right\rangle + \frac{\|w\|_1}{\|w_S\|_1} \leq \frac{1}{4}\left\langle\sqrt{\beta}\mathbf{I}, \frac{1}{\sqrt{\beta}}\mathbf{I}\right\rangle + \frac{\sqrt{\beta}}{\alpha} \leq \frac{k}{2}.$$

The first three equalities expanded definitions; the first inequality is by Lemmas 3 and 4. The second inequality used Definition 2 twice, which implies that $\|w_S\|_1 \geq \alpha\sqrt{\beta}$, as well as the exit condition in

Line 4. The third inequality follows from the definition of $k$. Finally, we conclude that $\tau$ is safe, by

$$\sum_{i \in T} \frac{w_i}{\|w\|_1} \tau_i = \sum_{i \in T} \frac{w_i}{\|w\|_1} \left\| \mathbf{\Sigma}^{-\frac{1}{2}} \mathbf{V}^\top \left( X_i - \mu_w(T) \right) \right\|_2^2$$

$$= \left\langle \mathbf{\Sigma}^{-1}, \mathbf{V}^\top \left( \sum_{i \in T} \frac{w_i}{\|w\|_1} \left( X_i - \mu_w(T) \right) \left( X_i - \mu_w(T) \right)^\top \right) \mathbf{V} \right\rangle = \left\langle \mathbf{\Sigma}^{-1}, \mathbf{\Sigma} \right\rangle = k.$$

$\square$

**Theorem 2.** *Under Assumption 1, with probability $1 - \delta$, the output of Algorithm 1 satisfies $\min_{\mu \in L} \|\mu - \mu^*\|_2^2 \le \frac{22}{\alpha}$. The overall runtime of Algorithm 1 is $O\left( n^2 dk \log\left( \frac{n}{\delta} \right) \right)$.*

*Proof. Runtime.* There are at most $n$ iterations in Algorithm 1, since at least one weight is zeroed out in each iteration. Further, the bottleneck operation in each iteration is Power, since an eigendecomposition of $\mathbf{\Sigma}$ takes time $O(k^3) = O(ndk)$. Since $\epsilon$ is a constant in Proposition 2, this yields the runtime. We note that the algorithm must terminate before removing all the weight: since $w$ is saturated, $\|w\|_1 \ge \alpha^2$ via Definition 2 and $\|w\|_1 \ge \|w_S\|_1$. By a union bound, with probability $1 - \frac{\delta}{2}$, Proposition 2 applies every iteration; we condition on this event for the remainder of the proof.
*Correctness.* As in Lemma 5, we let $w$ denote the weights in the last iteration of the algorithm (after exiting on Line 10). Denote $\mathbf{P} := \mathbf{V}\mathbf{V}^\top$ and $Y_i := \mathbf{P}X_i$ for all $i \in T$. Since

$$\sum_{i \in S} \frac{1}{\alpha n} \left( Y_i - \mathbf{P}\mu^* \right) \left( Y_i - \mathbf{P}\mu^* \right)^\top = \mathbf{P} \left( \sum_{i \in S} \frac{1}{\alpha n} \left( X_i - \mu^* \right) \left( X_i - \mu^* \right)^\top \right) \mathbf{P} \preceq \mathbf{P},$$

by Assumption 1, the expectation of $\|Y_i - \mathbf{P}\mu^*\|_2^2$ for a uniformly random sample $i \in S$ is $\frac{4}{\alpha}$ by linearity of trace. By Markov, with probability $\ge \frac{1}{2}$ a sample from $S$ has $\|Y_i - \mathbf{P}\mu^*\|_2^2 \le \frac{8}{\alpha}$, so with probability $\ge 1 - \frac{\delta}{2}$, some element of $L$ uses an $X_i$ with $\|Y_i - \mathbf{P}\mu^*\|_2^2 \le \frac{8}{\alpha}$. For this $i$, we expand

$$\|(\mathbf{P}X_i + (\mathbf{I} - \mathbf{P})\mu_w(T)) - \mu^*\|_2^2 = \|Y_i - \mathbf{P}\mu^*\|_2^2 + \|(\mathbf{I} - \mathbf{P})(\mu_w(T) - \mu^*)\|_2^2$$

$$\le \frac{8}{\alpha} + \|(\mathbf{I} - \mathbf{P})(\mu_w(T) - \mu^*)\|_2^2.$$

To bound this second term, we apply Lemma 2 on the set of points $\{(\mathbf{I} - \mathbf{P})X_i\}_{i \in T}$. This implies

$$\|(\mathbf{I} - \mathbf{P})(\mu_w(T) - \mu^*)\|_2^2 \le \frac{2\beta}{\|w_S\|_1} \|(\mathbf{I} - \mathbf{P})\mathrm{Cov}_w(T)(\mathbf{I} - \mathbf{P})\|_{\mathrm{op}} + \frac{2\alpha}{\beta} \le \frac{12\sqrt{\beta}}{\|w_S\|_1} + \frac{2\alpha}{\beta} \le \frac{14}{\alpha}.$$

Here, the last inequality used that saturation implies $\beta \ge \alpha^2$. The second used that Power and the termination condition imply that, since eigenvalues of $\mathbf{\Sigma}$ are the same as $\mathbf{V}\mathbf{V}^\top \mathrm{Cov}_w(T)\mathbf{V}\mathbf{V}^\top$,

$$\|(\mathbf{I} - \mathbf{P})\mathrm{Cov}_w(T)(\mathbf{I} - \mathbf{P})\|_{\mathrm{op}} \le 1.2\lambda_k(\mathrm{Cov}_w(T)) \le 1.5\lambda_k(\mathbf{\Sigma}) \le \frac{6}{\sqrt{\beta}}. \qquad \square$$

While Theorem 2 achieves the desired error, it has a quadratic dependence on the sample size $n$, and a suboptimal list size by a factor of $O(\log \frac{1}{\delta})$. We address both issues with our fast algorithm.

## 5   Nearly-PCA time algorithm: overview of FastSIFT

In this section, we highlight the main technical ideas behind our fast algorithms. In particular, we show how to leverage tools built in developing SIFT and combine them with a weight removal scheme based on a Ky Fan variant of the matrix multiplicative weights (MMW) regret minimization framework we develop, to give our final algorithm. Throughout this overview, we define integer $k = \Theta(\frac{1}{\alpha})$ to represent some dimensionality of a linear subspace; constants will be specified in relevant algorithms in the supplementary material.

**Scoring via Ky Fan matrix multiplicative weights.** To obtain the main result of this paper, it remains to show how we can improve the number of iterations of SIFT to polylogarithmic. For this, we turn to a strategy originating in [DHL19] in the large-$\alpha$ regime, which is to use the *matrix multiplicative weights* regret minimization framework to define weights for stronger performance guarantees. The intuition is that by using scores defined by more than the top eigenvector of the current covariance matrix (or in this paper, the top $k$ eigenvectors), we capture more than one bad point at a time and obtain better worst-case iteration bounds. The main MMW regret guarantee makes this formal. Roughly, it says that if in each iteration we can downweight the current covariance so that its inner product with a matrix given by the MMW framework is small, then in $O(\log d)$ iterations we can halve the operator norm (the maximal inner product against *all* trace-1 nonnegative matrices).

A key technical contribution of this paper is to give a Ky Fan $k$-norm (sum of $k$ largest eigenvalues) generalization of MMW, which typically gives operator norm guarantees. We analyze our algorithm and show that it is tolerant to the error guarantees of approximate $k$-PCA procedures such as simultaneous power iteration [MM15]. Crucial to our tightest runtime bounds are strengthenings of the analysis of a similar procedure found in [CMY20] in several places, which save multiple $k$ factors in our guarantees and may be of independent interest; we now highlight a few here.[6]

The main idea of our Ky Fan MMW regret guarantee is to bound the cost of actions $\{\mathbf{Y}_t\}_{t\geq 0}$ against a sequence of positive semidefinite "gain matrices" $\{\mathbf{G}_t\}_{t\geq 0}$ as measured by inner products. The actions $\{\mathbf{Y}_t\}_{t\geq 0}$ are given by the algorithm (depending on the gain matrices), and live in

$$\mathcal{Y} := \{\mathbf{Y} \in \mathbb{R}^{d\times d} \mid \mathbf{0} \preceq \mathbf{Y} \preceq \mathbf{I}, \mathrm{Tr}(\mathbf{Y}) = k\}.$$

The reason for this choice of action set, the "$k$-Fantope," is because it satisfies $\sup_{\mathbf{U}\in\mathcal{Y}} \langle \mathbf{U}, \mathbf{G}\rangle = \|\mathbf{G}\|_k$, so the best action over $\mathcal{Y}$ in hindsight captures this norm. Ultimately, our re-weighting scheme requires matrix-vector query access to each $\mathbf{Y}_t$, which are defined by *Bregman projections* onto the set $\mathcal{Y}$. It was shown in [CMY20] that the natural choice of projection, induced by a regularizer $r(\mathbf{Y})$ chosen to be matrix entropy, is a truncated exponential, where truncation occurs on the top-$k$ eigenspace. The bottleneck cost is computing this space, which normally requires an eigendecomposition.

To this end, we show new guarantees on the performance of approximate $k$-PCA, which allow for their use in this process. In particular, we prove the following new (informal) fact; a more formal statement is in the supplementary material.

**Proposition 3** (Informal, see Proposition 7 of the supplementary material). *Let* $\mathbf{P} = \mathbf{V}\mathbf{V}^\top$ *where* $\mathbf{V} \in \mathbb{R}^{d\times k}$ *is the result of* $\widetilde{O}(\epsilon^{-1})$ *iterations of simultaneous power iteration (approximate top-$k$ eigenvectors) on* $\mathbf{S} \in \mathbb{S}_{\geq 0}^d$. *Then with high probability,*

$$(1-\epsilon)\mathbf{S} \preceq \mathbf{PSP} + (\mathbf{I}-\mathbf{P})\mathbf{S}(\mathbf{I}-\mathbf{P}) \preceq (1+\epsilon)\mathbf{S}.$$

This improves a similar analysis in [CMY20], which showed an approximation factor of $1 \pm k\epsilon$. The other technical piece required by our MMW algorithm is the following refined divergence bound.

**Proposition 4** (Informal, see Lemma 13 of the supplementary material). *Let* $r(\mathbf{Y}) = \langle \mathbf{Y}, \log \mathbf{Y}\rangle$ *be the matrix entropy function defined on* $\mathcal{Y}$, *let* $r^*$ *be its convex conjugate over* $\mathbb{S}^d$, *and let* $V_{\mathbf{A}}^{r^*}(\mathbf{B}) := f(\mathbf{B}) - f(\mathbf{A}) - \langle \nabla f(\mathbf{A}), \mathbf{B} - \mathbf{A}\rangle$ *be the induced Bregman divergence. Then for all* $\mathbf{S} \in \mathbb{S}^d$, $\mathbf{G} \in \mathbb{S}_{\geq 0}^d$ *and* $\eta > 0$ *satisfying* $\|\eta\mathbf{G}\|_{\mathrm{op}} \leq \frac{1}{2}$,

$$V_{\mathbf{S}}^{r^*}(\mathbf{S} + \eta\mathbf{G}) \leq \langle \eta\mathbf{G}, \mathbf{Y}\rangle, \text{ where } \mathbf{Y} := \nabla r^*(\mathbf{S}) \in \mathcal{Y}.$$

This is a strengthening of a much more straightforward bound of $k\|\eta\mathbf{G}\|_{\mathrm{op}}$, which follows easily from strong convexity of $r$ in the trace norm (and hence smoothness of its dual).[7] We require this strengthening so that we can use the action matrices $\{\mathbf{Y}_t\}_{t\geq 0}$ to define scores, to decrease inner products: the weaker bound above has no dependence on $\overline{\mathbf{Y}}$, so without the stronger bound it is unclear how to use the MMW update structure to downweight. A similar refined regret bound as Proposition 3 is derived in [Nes07], in the special case when $k = 1$ (and $\mathcal{Y} \subset \mathbb{S}_{\geq 0}^d$ is trace-1 matrices).

---

[6]We believe that similar wins following from our tighter analysis apply to [CMY20], and brings their overall runtime down to roughly $\widetilde{O}(ndk^4)$. We give a discussion of their $k$ dependence in the supplementary material.

[7]It is a strengthening since $\nabla r^*(\mathbf{S}) \in \mathcal{Y}$, so we can apply a matrix Hölder's inequality and use $\mathrm{Tr}(\mathbf{Y}) = k$, $\forall \mathbf{Y} \in \mathcal{Y}$.

We prove our refined divergence bound by adapting arguments from previous literature [CDST19, JLL$^+$20] on using Hessian formulae of spectral functions to prove divergence bounds, whenever the conjugate $r^*$ is twice-differentiable, and applying the Alexandrov theorem. Finally, up to (non-dominant) approximation error terms, our Ky Fan MMW procedure's main guarantee can be stated as: given a sequence of positive semidefinite matrices $\{\mathbf{G}_t\}_{t\geq 0}$, let step size $\eta > 0$ satisfy $\eta \mathbf{G}_t \preceq \mathbf{I}$ for all $t$. The procedure plays a sequence $\{\mathbf{Y}_t\}_{t\geq 0} \in \mathcal{Y}$, so that for any $T \in \mathbb{N}$,

$$\left\| \frac{1}{T} \sum_{t=0}^{T-1} \mathbf{G}_t \right\|_k \leq \frac{2}{T} \sum_{t=0}^{T-1} \langle \mathbf{G}_t, \mathbf{Y}_t \rangle + \frac{k \log d}{\eta T}. \tag{2}$$

**Win-win-win analysis of MMW: FastSIFT.** We now describe how to use the regret guarantee (2) to obtain a faster algorithm. In particular, when the sequence $\{\mathbf{G}_t\}_{t\geq 0}$ is monotonically non-increasing, we can choose $\eta = \|\mathbf{G}_0\|_{\mathrm{op}}^{-1}$ to meet all the boundedness conditions $\eta \mathbf{G}_t \preceq \mathbf{I}$. If we can guarantee that every $\langle \mathbf{G}_t, \mathbf{Y}_t \rangle$ is bounded by, say, $\frac{1}{5} \|\mathbf{G}_0\|_k$, and $k \|\mathbf{G}_0\|_{\mathrm{op}} \leq 2 \|\mathbf{G}_0\|_k$ (i.e. the top $k$ eigenvalues of $\mathbf{G}_0$ are roughly uniform), the above regret guarantee becomes

$$\|\mathbf{G}_T\|_k \leq \left\| \frac{1}{T} \sum_{t=0}^{T-1} \mathbf{G}_t \right\|_k \leq \frac{2}{5} \|\mathbf{G}_0\|_k + \frac{2 \|\mathbf{G}_0\|_k \log d}{T} .$$

Now, $T = O(\log d)$ iterations suffice to halve the Ky Fan-$k$ norm. Our strategy, following [DHL19], is to let $\mathbf{G}_t$ be the empirical covariance matrix with respect to $w_t$, for monotonically decreasing weight sequence $\{w_t\}_{t\geq 0}$ formed by safe weight removals (1). This suggests that if we can implement this strategy, we will reach the termination condition of SIFT in polylogarithmically many iterations of MMW (since an upper bound on the Ky Fan norm of the empirical covariance also implies a bound on the $k^{\mathrm{th}}$ largest eigenvalue). At this point, a few questions remain.

1. How do we define the sequence $\{\mathbf{G}_t\}_{t\geq 0}$ so that it is monotonically decreasing? For instance, our safety condition (Definition 3) is defined with respect to normalized scores, but normalizing the covariance matrices makes them no longer (necessarily) monotone.

2. How do we whiten the scores so that any one of the top $k$ eigenvalues does not dominate? This requirement arises in several places in the analysis (akin to in the analysis of SIFT), for example in our earlier assumption that $k \|\mathbf{G}_0\|_{\mathrm{op}} \leq 2 \|\mathbf{G}_0\|_k$. We note that using a trick similar to normalizing the top-$k$ eigenspace to be the identity, as in SIFT, is not effective here as these spaces may be incompatible, and break monotonicity of gain matrices.

3. How do we safely downweight the covariances to make them satisfy $\langle \mathbf{G}_t, \mathbf{Y}_t \rangle \leq \frac{1}{5} \|\mathbf{G}_0\|_k$?

We show that a careful analysis of each failure case leads to a different "win condition" in the algorithm, which lets us certify progress in a different way.

1. We restart the algorithm in phases where the $\ell_1$ norm of the weights halves, so that in each phase the normalizing constant is stable. There can only be logarithmically many phases.

2. We restart the algorithm whenever the $k^{\mathrm{th}}$ largest eigenvalue of the covariance matrix is smaller than half the largest, setting aside the $k$ eigendirections. The remainder of the algorithm works in the space orthogonal to these directions. Since each time we set aside $k$ directions we halve the operator norm on the remaining subspace, this only occurs logarithmically many times, and yields a "set-aside subspace" of dimension $\widetilde{O}(k)$.

3. Whenever $\Theta(\log d)$ iterations pass without meeting either of the above "exit criteria," we use binary searches to safely remove as much weight as possible so each covariance matrix $\mathbf{G}_t$ meets the inner product criteria through $\mathbf{Y}_t$ to progress, i.e. we use weight removal to bound the right hand side of the regret guarantee (2). This argument follows the safety analysis of SIFT closely, crucially using that the top $k$ eigenvalues are roughly uniform.

By carefully reasoning about when each of the above three cases occurs, we eventually conclude that we are able to return in polylogarithmically many iterations a pair $(\mathbf{B}, w)$ such that $\mathbf{B}$ is an orthonormal basis of a subspace of dimension roughly $k \log d$, and $w$ is some weight vector whose empirical covariance's projection into $\mathbf{B}^\perp$ has bounded operator norm. At this point, we can use the empirical mean in $\mathbf{B}^\perp$ to learn the mean in all but $k \log d$ dimensions. To learn the mean in $\mathbf{B}$, there are a number of much simpler strategies such as random sampling from the dataset, which we discuss when describing ListDecodableMeanEstimation in the sequel.

**Preprocessing and postprocessing.** The above argument assumed that we initially had a bounded dataset diameter. We show a simple equivalence class partitioning, PreProcess, based on one-dimensional projections efficiently produces clusters with polynomially bounded diameter, so that the entire good dataset lies in the same partition (with high probability), yielding the following claim.

**Lemma 6** (Informal, see Lemma 12 of the supplementary material). *There is an algorithm running in time $O(nd + n \log n)$ that partitions $T$ into disjoint clusters with radius $poly(n, \delta^{-1})$ so that all of $S$ is in the same cluster with probability $\geq 1 - \delta$.*

Furthermore, we give a greedy clustering step PostProcess which works in a low-dimensional subspace, reducing the size of a candidate list to the optimal $O(k)$ (removing extraneous logarithmic factors), while only affecting the error guarantee by a constant. The statement of PostProcess depends on the specific guarantees of SIFT and FastSIFT (i.e. the decomposition into a $\Theta(\frac{1}{\alpha})$-dimensional subspace and its complement), so we defer it to the supplementary material for brevity.

**Putting it all together:** ListDecodableMeanEstimation. Applying PreProcess and FastSIFT sequentially yields a $\widetilde{O}(\frac{nd}{\alpha})$-time algorithm which learns a subspace of dimension $\widetilde{O}(k)$ spanned by $\mathbf{B}$, where the empirical mean in $\mathbf{B}^\perp$ is an estimate attaining the information-theoretic limit. It remains to produce a list which learns the mean inside $\mathbf{B}$. At this stage, we employ two strategies: we can either randomly sample, as in the last line of SIFT, or we can simply run SIFT itself with a smaller dataset size on the subspace. Applying PostProcess to the output lists of these two strategies yield our full algorithms, whose guarantees are respectively summarized in the last two rows of Table 1.

## Acknowledgments and Disclosure of Funding

We thank Morris Yau for clarifying conversations about the prior work [CMY20]. Ilias Diakonikolas is supported by NSF Award CCF-1652862 (CAREER), a Sloan Research Fellowship, and a DARPA Learning with Less Labels (LwLL) grant. Daniel Kane is supported by NSF CAREER Award ID 1553288 and a Sloan fellowship. Kevin Tian is supported by NSF CAREER Award CCF-1844855 and NSF Grant CCF-1955039.

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
