# List-Decodable Mean Estimation in Nearly-PCA Time

Ilias Diakonikolas[*]     Daniel M. Kane[†]     Daniel Kongsgaard[‡]     Jerry Li[§]

Kevin Tian[¶]

## Abstract

Robust statistics has traditionally focused on designing estimators tolerant to a minority of contaminated data. *List-decodable learning* [CSV17] studies the more challenging regime where only a minority $\frac{1}{k}$ fraction of the dataset, $k \geq 2$, is drawn from the distribution of interest, and no assumptions are made on the remaining data. We study the fundamental task of list-decodable mean estimation in high dimensions. Our main result is a new algorithm for bounded covariance distributions with optimal sample complexity and near-optimal error guarantee, running in *nearly-PCA time*. Assuming the ground truth distribution on $\mathbb{R}^d$ has identity-bounded covariance, our algorithm outputs $O(k)$ candidate means, one of which is within distance $O(\sqrt{k \log k})$ from the truth.

Our algorithm runs in time $\widetilde{O}(ndk)$[1], where $n$ is the dataset size. This runtime nearly matches the cost of performing $k$-PCA on the data, a natural bottleneck of known algorithms for (very) special cases of our problem, such as clustering well-separated mixtures. Prior to our work, the fastest runtimes were $\widetilde{O}(n^2dk^2)$ [DKK20], and $\widetilde{O}(ndk^C)$ [CMY20] for an unspecified constant $C \geq 6$. Our approach builds on a novel soft downweighting method we term SIFT, arguably the simplest known polynomial-time mean estimator in the list-decodable setting. To develop our fast algorithms, we boost the computational cost of SIFT via a careful "win-win-win" analysis of an approximate Ky Fan matrix multiplicative weights procedure we develop, which may be of independent interest.

---

[*]University of Wisconsin, Madison, `ilias@cs.wisc.edu`

[†]University of California, San Diego, `dakane@cs.ucsd.edu`

[‡]University of California, San Diego, `dkongsga@ucsd.edu`

[§]Microsoft Research, `jerrl@microsoft.com`

[¶]Stanford University, `kjtian@stanford.edu`. Part of this work was done as an intern at Microsoft Research.

[1]Throughout this work, the $\widetilde{O}$ notation hides logarithmic factors in $n$ and the failure probability.

# Contents

# 1    Introduction

Mean estimation has emerged as one of the cornerstone tasks in robust statistics, as the most basic in a hierarchy of increasingly complex estimation problems. The problem is straightforward to state: given samples from a "nice" ground-truth distribution $\mathcal{D}$, where an adversary has (arbitrarily) corrupted a fraction of the data, recover the mean of $\mathcal{D}$ as accurately as possible. Due to its fundamental nature, robust mean estimation has received extensive study in the statistics, theoretical computer science, and machine learning communities, starting from the 1960s [Ans60, Tuk60, Hub64, Tuk75].

Despite the apparent simplicity of the problem, efficient algorithms that achieved nearly-optimal error rates were not known in high-dimensional settings until recently [LRV16, DKK+19a, DKK+17]. These works studied mean recovery in the traditional setting where a majority of the data is "trusted," i.e. the fraction of corruptions is strictly less than $\frac{1}{2}$. For the standard formulation of robust mean estimation, this assumption is necessary. Indeed, if only an $\alpha \le \frac{1}{2}$ fraction of points can be trusted, then the dataset could consist of $O(\frac{1}{\alpha})$ well-separated clusters of "good" points. Thus, the mean of each individual cluster is an equally valid solution to the robust mean estimation problem, so asking for a single solution is ill-posed.

In many settings of theoretical and practical interest, asking for a majority of inlier points is too strong of an assumption. To circumvent the issue of well-posedness in the $\alpha \le \frac{1}{2}$ regime, [CSV17] proposed a relaxed notion of learning termed *list-decodable learning*. Rather than being restricted to a single hypothesis, the algorithm is allowed to output a list of $O(\frac{1}{\alpha})$ hypotheses, with the guarantee that at least one of them is close to the truth. In the context of robust mean estimation, this amounts to outputting a list of $O(\frac{1}{\alpha})$ candidate means.

A natural problem in its own right, list-decodable mean estimation is also a generalization of a number of other well-studied problems. A prototypical example is learning well-separated mixture models, a task which has received extensive treatment in the literature [Das99, VW04, AM05, DS07, AK05, RV17, HL18, DKS18, KSS18]. In this problem, data is drawn from a uniform mixture[2] of $k$ "nice" distributions $\mathcal{D}_1, \ldots, \mathcal{D}_k$, whose means are far apart relative to their covariances, and the goal is to recover clusters which correspond to samples coming from each component. By running a list-decodable mean estimation procedure with $\alpha = \frac{1}{k}$, each true cluster of points is an equally valid "ground-truth distribution," so the output list must contain candidate means close to each of the true means. If the candidates are sufficiently close to the true means, standard techniques allow for recovery of the true clustering. List-decodable mean estimation robustly extends this clustering problem to tolerate adversarial noise or non-uniformity, up to constants in the output size.

Moreover, list-decodable mean estimation can be used to model important data science applications such as crowdsourcing (where a majority of respondents could be unreliable or malicious) [SVC16, MV18], or semi-random community detection in stochastic block models [CSV17]. This primitive is particularly useful in the context of semi-verified learning [CSV17, MV18], where a learner can audit a small amount of trusted data. Even if the trusted dataset is too small to directly learn from, in conjunction with a list-decodable learning procedure it can pinpoint a candidate hypothesis consistent with the verified data (indeed, only roughly $\log \frac{1}{\alpha}$ vetted points are required).

The first tractable algorithm for high-dimensional list-decodable mean estimation was due to [CSV17]. Their work considered the setting where $\mathcal{D}$ has (unknown) covariance $\boldsymbol{\Sigma}$, satisfying $\boldsymbol{\Sigma} \preceq \sigma^2 \mathbf{I}$ for some known $\sigma$ (i.e. a second moment bound). In this setting, [CSV17] gave an algorithm which is sample-optimal, runs in polynomial time, and which outputs a list of $O(\frac{1}{\alpha})$ candidate means, so

---

[2]Some algorithms extend beyond the uniform setting, but we present it this way here for simplicity of exposition.

that some candidate is within $\ell_2$ distance $O(\sigma \cdot \sqrt{\alpha^{-1} \cdot \log \alpha^{-1}})$ from the mean of $\mathcal{D}$. As was later demonstrated in [DKS18], this error rate is optimal up to logarithmic factors under a second moment bound. However, the [CSV17] algorithm heavily relies on black-box semidefinite programming solvers, and as a result the runtime is prohibitively large in high-dimensional problem instances.

The goal of our work is to develop much faster, truly scalable algorithms for list-decodable mean estimation which achieve optimal statistical guarantees. This goal fits broadly into a larger line of work focused on understanding the computational cost of robustness for basic statistical tasks. In some settings, this line has demonstrated strong evidence that robustness comes at an inherent computational cost [DKS17, HL19]. In contrast, recent algorithms have been developed that achieve robustness essentially "for free" in many other settings [CDG19, DHL19, CDGW19, LY20, JLT20].

While list-decodable mean estimation has received a fair amount of attention (cf. Section 1.2), there have only been a few results achieving improved runtimes. One line of work proposed an algorithm design framework termed *multi-filtering* [DKS18, DKK20], based on learning multiple candidate "weight functions." In particular, [DKK20] uses this approach to design an algorithm achieving nearly-optimal error, in time $\widetilde{O}(n^2 d \alpha^{-2})$, where $n$ is the size of the overall dataset. While this runtime dramatically improves over the runtime in [CSV17], the quadratic dependence on $n$ is not ideal in very high-dimensional problem settings. Concurrently to [DKK20], the work [CMY20] proposes a different, descent-based algorithm based on (approximate) positive semidefinite programming, achieving optimal error (up to constant factors) in time $\widetilde{O}(nd\alpha^{-C})$ for some constant $C \geq 6$. When $\alpha = \Theta(1)$, the [CMY20] runtime is nearly-linear in the problem input size. However, if $\alpha^{-1}$ scales polynomially with $d$, e.g. in learning a mixture model with many components in moderate dimension, then this large dependence on $\alpha^{-1}$ may also be prohibitively slow.

In contrast to this somewhat murky runtime landscape, the state of affairs for clustering separated mixture models is relatively clear. The fastest algorithm for clustering a mixture of $k$ well-separated components is almost twenty years old [VW04], and runs in time $\widetilde{O}(ndk)$, as a relatively simple and elegant application of (approximate) $k$-principal components analysis ($k$-PCA). Since list-decodable mean estimation can be thought of as the natural robust analog to clustering mixture models, it is natural to ask:

*Can we perform list-decodable mean estimation as efficiently as learning mixture models?*

Concretely, since clustering mixture models corresponds to an instance of list-decodable learning with $\alpha = k^{-1}$, the question becomes: can we solve list decodable mean estimation in time $\widetilde{O}(\frac{nd}{\alpha})$? This runtime presents itself as a natural barrier for our problem, since any further runtime improvement would also imply faster learning of mixture models.

## 1.1 Our results

Our main contribution is to answer this question affirmatively for a wide range of problem parameters. Our first result is the following, which states that we can nearly match the runtime of $k$-PCA while obtaining optimal statistical guarantees up to constants.

**Theorem 1** (informal, cf. Theorem 4)**.** *Let $\alpha \in (0, \frac{1}{2})$. Let $\mathcal{D}$ be a distribution with unknown mean $\mu^* \in \mathbb{R}^d$ and covariance matrix $\mathbf{\Sigma} \preceq \sigma^2 \mathbf{I}$. Let $T \subset \mathbb{R}^d$ have $|T| = n$, an $\alpha$ fraction of which is drawn independently $\sim \mathcal{D}$. For $n = \Omega(\frac{d}{\alpha})$, Algorithm 8 outputs a list of $m = O(\frac{1}{\alpha})$ hypotheses $\{\mu_j\}_{j \in [m]}$*

*so that* $\min_{j \in [m]} \|\mu^* - \mu_j\|_2 = O\left(\sigma\sqrt{\frac{1}{\alpha}}\right)$, *with high probability. The runtime of the algorithm is*

$$\widetilde{O}\left(\frac{nd}{\alpha} + \frac{1}{\alpha^6}\right).$$

We make a few remarks regarding this result. It is known that a list size of $\Omega(\frac{1}{\alpha})$, sample complexity of $\Omega(\frac{d}{\alpha})$, and error of $\Omega\left(\sigma\alpha^{-0.5}\right)$ are information-theoretically necessary [DKS18]. Further, without loss of generality $d = \omega(\alpha^{-1})$, as otherwise there is a trivial algorithm for this problem (cf. Appendix A), so the $\alpha^{-6}$ additive term in the runtime is only dominant when $\alpha^{-1} = \Omega(\sqrt{d})$. Notably, even with this additive overhead, our runtime is the best-known in all parameter regimes.

We also present an algorithm with an alternative postprocessing scheme which removes the $\alpha^{-6}$ dependence in the runtime, at the cost of a $\sqrt{\log \alpha^{-1}}$ factor in the final error.

**Theorem 2** (informal, cf. Corollary 2). *In the same setting as Theorem 1, Algorithm 8 using Algorithm 9 instead of Algorithm 2 outputs a list of $m = O(\frac{1}{\alpha})$ hypotheses $\{\mu_j\}_{j \in [m]}$ so that* $\min_{j \in [m]} \|\mu^* - \mu_j\|_2 = O\left(\sigma\sqrt{\frac{\log \alpha^{-1}}{\alpha}}\right)$, *with high probability. The runtime of the algorithm is*

$$\widetilde{O}\left(\frac{nd}{\alpha}\right).$$

| Reference | Runtime | Error guarantee |
|-----------|---------|-----------------|
| [CSV17] | $\mathrm{poly}(n, d, \alpha^{-1})$ | $O(\sigma\alpha^{-\frac{1}{2}}\sqrt{\log \alpha^{-1}})$ |
| [DKK20] | $\widetilde{O}(n^2 d\alpha^{-2})$ | $O(\sigma\alpha^{-\frac{1}{2}}\log \alpha^{-1})$ |
| [CMY20] | $\widetilde{O}(nd(\alpha^{-1})^{\geq 6})$ | $O(\sigma\alpha^{-\frac{1}{2}})$ |
| **Our work** | $\widetilde{O}(nd\alpha^{-1})$ | $O(\sigma\alpha^{-\frac{1}{2}}\sqrt{\log \alpha^{-1}})$ |
| **Our work** | $\widetilde{O}(nd\alpha^{-1} + \alpha^{-6})$ | $O(\sigma\alpha^{-\frac{1}{2}})$ |

Table 1: List-decodable mean estimation algorithm runtimes. All algorithms listed return lists of size $O(\frac{1}{\alpha})$ and use sample complexity $n = O(\frac{d}{\alpha})$, which are information-theoretically optimal [DKS18]. The $\widetilde{O}$ notation hides polylogarithmic factors in failure probability and dimension.

Our approach is inspired by the way in which fast algorithms for robust mean estimation in the $\alpha \to 1$ regime were built. At a high level, a "simple" polynomial (but not nearly-linear) time algorithm — namely, the filter — was first developed [DKK+19a, DKK+17, Ste18]. After the most basic tractable algorithm for the problem was discovered, it was sped up in subsequent works by combining it with tools developed by the continuous optimization community [CDG19, DHL19], specifically based on regret analyses of the matrix multiplicative weights (MMW) updates.

In this paper, we accomplish both of these steps for the $\alpha \ll \frac{1}{2}$ regime. First, we design a simple "basic" algorithm for the problem, and then we demonstrate how to speed it up using matrix regret minimization tools. Both of these steps require substantially new ideas from previous work, which we now briefly discuss, and survey in more detail in Section 1.3.

SIFT: **a new, simple algorithm for list-decodable learning.** Our first main contribution is

a novel algorithm for list-decodable mean estimation, which we call SIFT (Subspace Isotropic Filtering), achieving optimal statistical guarantees (up to constants) in time $\widetilde{O}(\frac{n^2 d}{\alpha})$. While by itself, SIFT does not achieve a nearly-linear runtime, its framework will be vital in designing our more sophisticated algorithms. Crucially, SIFT is conceptually different from all previous approaches for list-decodable mean estimation, and it is these differences that allow for our later speedups.

The main advantage of SIFT is its simplicity. All prior algorithms for list-decodable mean estimation were quite complicated, with rather involved and lengthy analyses, whereas a complete analysis of SIFT fits within roughly five pages. Because of this, we believe SIFT is of independent interest (both theoretically and practically), and can find applications in other list-decodable learning settings.

Prior list-decodable mean estimation algorithms [DKS18, DKK20, CMY20] sought to directly identify candidate clusters of points. However, the techniques developed to do so turn out to be quite complicated. In contrast, SIFT first seeks to solve an intermediate problem: find an $O(\alpha^{-1})$-dimensional subspace, containing (most of) the deviation of the true mean from the empirical mean. This is motivated by — and can be seen as a robust analog of — the application of $k$-PCA for clustering mixture models. After finding this subspace, we can then solve the problem in the low dimensional subspace via a naïve clustering method, to find all clusters at once.

This approach has a number of conceptual advantages. For one, the aforementioned prior algorithms often interlace "clustering" steps with "filtering" steps. Loosely speaking, a "clustering" step is one in which the algorithm identifies a potential cluster of good points, or a union of such clusters, and a "filtering" step is one in which the algorithm downweights points which are unlikely to be in any such cluster. The interplay between recursive calls of these two types of steps results in a variety of complications in speeding up prior algorithms. In contrast, we find all of the candidate clusters simultaneously, in the very last step of our algorithm.

To solve the intermediate problem of finding a low-dimensional subspace, we need two main technical innovations: (1) a new outlier-scoring function (for detecting which data points are likely to be outliers), and (2) a new safety condition (for maintaining an invariant on weights of the good set). Our scoring function leverages information about the subspace spanned by the top $k = \Theta(\alpha^{-1})$ eigenvectors of the empirical covariance simultaneously. In contrast, prior scores such as those used in the multi-filter [DKS18, DKK20] or in the basic filter for the $\alpha \to 1$ regime [DKK+17, Ste18], only used the top eigenvector. To ensure that no single cluster of points dominates the scores, we apply a whitening transformation on the subspace of top eigenvectors to make the data isotropic. We demonstrate that downweighting points based on these scores preserves a strong safety condition we call *saturation*. This condition guarantees that the total fraction of weight remaining on the good points actually increases as the overall total weight decreases, and ensures that we never lose too much information about the good points, as the process continues.

Finally, we terminate the procedure when the $k^{\text{th}}$ largest eigenvalue of the empirical covariance is small. We show that combining this with the saturation condition allows us to learn the mean outside of a $k$-dimensional subspace. By combining with a low-dimensional algorithm to estimate the mean within the subspace (i.e. naïve clustering), we obtain our overall SIFT algorithm.

**FastSIFT: speeding up SIFT via Ky Fan regret minimization.** While each iteration of the SIFT algorithm can be performed in time $\widetilde{O}(ndk)$, SIFT requires $\Theta(n)$ iterations in the worst case. This is because there are simple hard instances in which each iteration of SIFT removes only one data point. Consequently, the main challenge is to combine the analysis of SIFT with a downweighting procedure which guarantees termination in polylogarithmically many iterations.

To achieve this goal, we use tools from semidefinite programming (SDP) to design iterative schemes

with stronger termination guarantees. This mirrors, and is inspired by, the approach used in the $\alpha \to 1$ regime, where tools such as packing semidefinite program solvers [CDG19] and matrix multiplicative weights [DHL19] were used to speed up the basic filter [DKK+17, Ste18] to achieve nearly-linear runtimes. We note that [CMY20] also uses SDP tools to obtain their runtime improvements. However, their use of these tools differs substantially from our work.

As we will explain in more detail in Section 1.3, there are a number of new technical and conceptual challenges to adapting matrix optimization tools to our setting. The first main difficulty is that we require MMW-style regret guarantees against Ky Fan $k$-norms, for $k = \Theta(\frac{1}{\alpha})$, rather than the standard spectral norm. However, to our knowledge the only prior analysis of such a procedure was due to [CMY20], which lost multiple factors of $k$ in their regret guarantees. To circumvent this, we provide a novel analysis of a "lazy mirror descent" procedure adapted to a Ky Fan constraint set, and prove that it achieves the same sorts of "local norm" bounds as [ZLO15] achieved for spectral norm procedures. Proving these guarantees requires a great deal of technical care (particularly under approximate $k$-PCA operations), and we believe it may be of independent interest.

Even with this powerful primitive, it is still not clear how to plug in the faster Ky Fan solver we develop to speed up SIFT. This is because several of the operations in SIFT appear to not be compatible with the requirements of regret minimization procedures. To get around this difficulty, we introduce a number of "exit conditions" for our multiplicative weights updates that, if violated, guarantee a great deal of progress on a different potential. If these exit conditions are not violated, then the iterative updates are sufficiently stable, ensuring progress on the original SIFT objective.

## 1.2 Related work

Robust statistics in its current form was first proposed in a series of papers by statisticians in the 1960s and 1970s [Ans60, Tuk60, Hub64, Tuk75]. Since then, there has been a tremendous amount of work in the area from the statistics community, see e.g. [Hub04]. Despite this, efficient algorithms for fundamental high dimensional problems in this field were not known until quite recently [DKK+19a, LRV16, DKK+17]. These algorithms and the techniques developed therein have been used to give robust estimators for a range of more complex problems, including covariance estimation [DKK+19a], sparse estimation tasks [BDLS17, DKK+19c], learning graphical models [CDKS18], linear regression [KKM18, DKS19], stochastic optimization [PSBR18, DKK+19b], and defending backdoor attacks against neural networks [TLM18], to name a few. The reader is referred to [DK19, Ste18, Li18] for more comprehensive overviews of these advances.

The aforementioned papers study robust statistics in the setting where $\alpha \to 1$. List-decodable learning as studied in this paper was first considered in [CSV17]; a similar learning model was introduced in [BBV08], albeit in a different setting. Subsequent research on list-decodable learning can broadly be split into two lines of work, which we now describe.

The first sequence focuses on obtaining better error bounds when the distribution is assumed to have additional structure, typically in the form of some control over the higher moments [HL18, KSS18, DKS18]. While these algorithms are able to achieve better error when the unknown distribution is (say) Gaussian, these algorithms require estimating higher order moments and also often use heavy-duty tools such as the sum-of-squares hierarchy. As a result, they all require significantly more samples and expensive computation than is required in the setting we study (i.e. under a minimal second moment bound assumption). These techniques have also been extended to settings such as list-decodable regression [RY20, KKK19] and subspace recovery [RY20, BK20].

The second line of work — and the one we extend — is one focusing on developing more efficient algorithms for list-decodable mean estimation. Prior to our work, two different approaches have

been proposed for this problem. One, developed in [DKS18, DKK20], presents a method termed a *multi-filter*. The multi-filter recursively uses univariate projections of the data to either filter out a small fraction of clear outliers, or divide the data into overlapping clusters. Using this framework, [DKK20] achieve a runtime of $\widetilde{O}(n^2 dk^2)$, and an error guarantee of $O\left(\sigma \cdot \alpha^{-0.5} \log(1/\alpha)\right)$. The second approach, and arguably the closest to ours, is the one introduced in [CMY20], which achieves a runtime of $\widetilde{O}(nd\alpha^{-C})$ for some $C \geq 6$. Their algorithm also uses $O(\alpha^{-1})$-dimensional information and tools from fast matrix optimization, specifically, generalizations of packing SDPs for the Ky Fan norm (an approach which builds on [CDG19], which handled the $\alpha \to 1$ regime).

We emphasize that we use these tools in fundamentally different ways than [CMY20]. In particular, the algorithm in [CMY20] uses a primal-dual approach reminiscent of [CDG19] to directly find one candidate cluster at a time. They then remove this cluster, and repeat the process. This approach requires rather sophisticated scoring techniques, and as a result their algorithm requires solving generalizations of packing SDPs in Ky Fan norms, similar to how [CDG19] require black-box packing SDP solvers. However, these solvers lose several $O(\alpha^{-1})$ factors in their runtime bounds. Moreover, even in the mixture model case, any process which sequentially removes one cluster at a time, and performs operations in $O(\alpha^{-1})$ dimensions, must pay a quadratic overhead in $O(\alpha^{-1})$ in the runtime. To obtain a linear dependence on $\alpha^{-1}$ requires an algorithmic approach beyond iterative cluster removal (and also requires SDP solvers with faster rates).

In sharp contrast, the algorithms we develop do not require such heavy-duty SDP solvers, but rather only need a refined regret guarantee against the $k$-Fantope, which drives our weight removal process. Rather than directly trying to find candidate clusters, we achieve our runtime improvement by identifying a low-dimensional subspace, such that outside the subspace the problem is trivial. We can then find all the clusters simultaneously, allowing us to avoid the quadratic overhead inherent in the [CMY20] approach, and the reliance on Ky Fan norm packing SDPs.

## 1.3 Technical overview

We now highlight the main technical ideas behind our algorithms. We begin by developing our basic algorithm, SIFT (cf. Theorem 3 in Section 3), focusing on how we overcome challenges which arise in modifying prior work from the "large-$\alpha$" regime [DKK+17, Li18, Ste18, DHL19] to the setting where most of the points are outliers. We then show how to leverage the tools built in developing SIFT to be combined with a weight removal scheme based on a Ky Fan-norm variant of the MMW regret minimization framework, to develop our final algorithm (cf. Theorem 4 in Section 5).

Throughout this overview, we define integer $k = \Theta(\frac{1}{\alpha})$ to represent some dimensionality of a linear subspace; particular constants will be specified in relevant algorithms.

**New safety condition for weight removal.** A powerful meta-technique which has emerged in the design of robust estimation algorithms is soft downweighting, or "filtering". Consider for simplicity first a corrupted dataset where an $1 - \epsilon$ fraction of the points are drawn from a "ground-truth" distribution, for $\epsilon \ll \frac{1}{2}$. The strategy of filtering then consists of the following steps.

1. Initialize a set of uniform weights $w$. We will try to non-uniformly decrease these to (relatively) downweight the corrupted subset $B$.

2. Iteratively identify a "certificate" of corruption, whose presence indicates outliers (e.g. an eigenvalue which is too large, and could only have been caused by an adversary). Ideally, in the absence of a certificate, the algorithm can successfully terminate with a good estimate.

3. Use the certificate to define scores $\{\tau_i\}$, such that the weighted average score in $B$ is larger

than the weighted average in the good subset $S$. Concretely, the following "safety condition,"

$$\sum_{i \in S} w_i \tau_i \leq \sum_{i \in B} w_i \tau_i, \tag{1}$$

is used. The guarantee (1) is referred to as a safety condition because it allows us to conclude that $\sum_{i \in S} w_i - w_i' \leq \sum_{i \in B} w_i - w_i'$, where

$$w_i' \leftarrow \left(1 - \frac{\tau_i}{\max_{i'} \tau_{i'}}\right) w_i, \ \forall i. \tag{2}$$

Ergo, downweighting points proportionally to their score removes less good weight than bad.

Eventually, the goal is to argue that enough weight must have been removed so there are no more bad points remaining. This clearly is too weak a goal in the small-$\alpha$ regime, since even removing e.g. twice as much bad weight as good weight can quickly lead to a situation where there are no good points remaining, and yet only a $3\alpha$ fraction of the original total weight has been removed. To drive our filtering approach in this work, we use a different notion of safety. We propose a normalized variant of (1), e.g.

$$\sum_{i \in S} \frac{w_i}{\|w_S\|_1} \tau_i \leq \frac{1}{2} \sum_{i \in T} \frac{w_i}{\|w\|_1} \tau_i, \tag{3}$$

to be our safety condition. Here, $T = S \cup B$ is the whole dataset. This specific choice of safety condition is due to the fact that iteratively decreasing weights via (2), using scores which satisfy (3), maintains the invariant

$$\|w_S\|_1 \geq \alpha\sqrt{\|w\|_1}. \tag{4}$$

We call such a set of weights *saturated*; this is made formal in Lemma 1. In other words, the total weight of the good set becomes more saturated as the algorithm progresses, to combat the fact that there are less good points to work with. By carefully balancing this saturation invariant with a choice of termination condition, we show that no matter how much weight we have removed when the algorithm ends, (4) suffices to guarantee we attain the minimax estimation error.

**Learning the mean in all but $k$ dimensions: SIFT.** We now sketch how to use the invariant (4) for mean estimation. Our first observation is that in the regime where the ambient dimension $d = \Theta(k)$, it is straightforward (up to logarithmic factors) to attain estimation error $\sqrt{k}$ just by randomly sampling points, since a typical point from $S$ is at this distance. This observation breaks the learning problem into two pieces: it suffices to learn the mean in any $d - k$-dimensional subspace up to Euclidean error $\sqrt{k}$, and then randomly sample in the remaining $k$ dimensions.

It is thus natural to use the $k^{\text{th}}$ largest eigenvalue of the covariance matrix as a termination criterion. This idea of "learning in all but $k$ dimensions" is suggested by the special case of learning uniform, well-separated mixture models where the dataset is composed of $k$ pieces, each drawn from a different bounded-covariance distribution. In this case, the empirical covariance will have $k$ large eigenvectors (caused by different cluster means), and the remaining directions will be concentrated. More generally, in the robust setting, any set of points with a large enough effect to fool the algorithm will intuitively simulate one of these clusters, and create a large eigendirection. It remains to show how to use the presence of $k$ large eigenvalues to create scores satisfying (3).

Letting $\lambda_k(\cdot)$ denote the $k^{\text{th}}$ largest eigenvalue, we choose our termination criterion as

$$\lambda_k(\text{Cov}_w(T)) = O\left(\frac{1}{\sqrt{\|w\|_1}}\right). \tag{5}$$

Here, $\text{Cov}_w(T)$ is the empirical covariance under given weights $w$. To use (5), we prove (cf. Lemma 2) that if weights $w$ are saturated (i.e. they satisfy (4)), then the weighted empirical mean satisfies

$$\|\mu_w(T) - \mu^*\|_2 = O\left(\sqrt{\|\text{Cov}_w(T)\|_{\text{op}} \frac{\|w\|_1}{\|w_S\|_1}}\right).$$

Plugging in (5) to the above bound, and using the definition of saturation (4), the mean distance bound above restricted to the space orthogonal to the top $k$ eigenvectors indeed is $O(\alpha^{-0.5}) = O(\sqrt{k})$. So, it suffices to show that the converse of (5) certifies scores satisfying (3).

A first natural attempt is to simply define scores of points via the length of their projection into the top-$k$ eigenspace of the covariance matrix, $\mathbf{V}_k \in \mathbb{R}^{d \times k}$:

$$\tau_i := \left\|\mathbf{V}_k^\top (X_i - \mu_w(T))\right\|_2^2.$$

Intuitively, if the weighted sum of these scores, i.e. the Ky Fan-$k$ norm of the covariance, is large (certified by (5) not holding), it must be because many clusters of far-out points are creating large eigenvalues. However, even then it is not clear that (3) holds, since the $k$ large directions may not be of equal magnitude (or worse, the "true" cluster may be the largest eigendirection). Our solution to this is simple: we "whiten" the top $k$ eigendirections to all have roughly equal energy, by renormalizing the top eigenspace to be the identity. In particular, we choose the scores

$$\tau_i := \left\|\mathbf{\Sigma}_k^{-\frac{1}{2}} \mathbf{V}_k^\top (X_i - \mu_w(T))\right\|_2^2, \text{ where } \mathbf{\Sigma}_k := \mathbf{V}_k^\top \text{Cov}_w(T) \mathbf{V}_k.$$

It is not difficult to show that the above scores satisfy the safety condition (3), whenever the termination condition (5) does not hold. By using this weight removal framework and iteratively maintaining the invariant (3), we show that whenever we have removed too much weight, the algorithm must terminate. Because every iteration of (2) removes at least one point, the algorithm runs in at most $n$ iterations. The bottleneck computation of each iteration is one top-$k$ eigenspace computation, i.e. $k$-PCA. These runtime and error guarantees are summarized in Theorem 3.

**Scoring via Ky Fan matrix multiplicative weights.** To obtain the main result of this paper, it remains to show how we can improve the number of iterations of our algorithm to polylogarithmic. For this, we turn to a strategy originating in [DHL19] in the large-$\alpha$ regime, which is to use the *matrix multiplicative weights* regret minimization framework to define weights for stronger performance guarantees. The intuition is that by using scores defined by more than the top eigenvector of the current covariance matrix (or in this paper, the top $k$ eigenvectors), we can capture more than one bad point at a time and obtain better worst-case iteration bounds. The main regret guarantee of MMW makes this formal. Roughly speaking, it says that if in each iteration we can downweight the current covariance so that its inner product with a certain matrix given by the MMW framework is small, then in logarithmically many iterations we can halve the operator norm.

A key technical contribution of this paper is to give a Ky Fan $k$-norm (sum of $k$ largest eigenvalues)

generalization of MMW, which typically gives operator norm guarantees. We analyze our algorithm and show that it is tolerant to the error guarantees of approximate $k$-PCA procedures such as simultaneous power iteration [MM15]. Crucial to our tightest runtime bounds are strengthenings of the analysis of a similar procedure found in [CMY20] in several places, which save multiple $k$ factors in our guarantees and may be of independent interest; we now highlight a few here.[3]

The main idea of our Ky Fan MMW regret guarantee is to bound the cost of actions $\{\mathbf{Y}_t\}_{t \geq 0}$ against a sequence of positive semidefinite "gain matrices" $\{\mathbf{G}_t\}_{t \geq 0}$ as measured by inner products. The actions $\{\mathbf{Y}_t\}_{t \geq 0}$ are given by the algorithm (depending on the gain matrices), and live in

$$\mathcal{Y} := \{\mathbf{Y} \in \mathbb{R}^{d \times d} \mid \mathbf{0} \preceq \mathbf{Y} \preceq \mathbf{I}, \mathrm{Tr}(\mathbf{Y}) = k\}.$$

The reason for this choice of action set, the "$k$-Fantope," is because it satisfies

$$\sup_{\mathbf{U} \in \mathcal{Y}} \langle \mathbf{U}, \mathbf{G} \rangle = \|\mathbf{G}\|_k,$$

where $\|\cdot\|_k$ is the Ky Fan $k$-norm, so the best action in hindsight captures this norm. Ultimately, our filtering scheme requires matrix-vector query access to each $\mathbf{Y}_t$, which are defined by *Bregman projections* onto the set $\mathcal{Y}$. It was shown in [CMY20] that the natural choice of projection, induced by a regularizer $r(\mathbf{Y})$ chosen to be matrix entropy, is a truncated exponential, where truncation occurs on the top-$k$ eigenspace. The bottleneck cost of iterations is computing this space.

To this end, we show new guarantees on the performance of approximate $k$-PCA, which allow for their use in this process. One example is that we show roughly $\frac{1}{\epsilon}$ iterations of simultaneous power iteration on a positive semidefinite matrix $\mathbf{S} \in \mathbb{R}^{d \times d}$, resulting in approximate eigenvectors $\mathbf{V} \in \mathbb{R}^{d \times k}$, are enough to guarantee (cf. Proposition 7)

$$(1 - \epsilon)\mathbf{S} \preceq \mathbf{PSP} + (\mathbf{I} - \mathbf{P})\,\mathbf{S}\,(\mathbf{I} - \mathbf{P}) \preceq (1 + \epsilon)\mathbf{S}, \text{ where } \mathbf{P} := \mathbf{VV}^\top.$$

This improves a similar analysis in [CMY20], which showed an approximation factor of $1 \pm k\epsilon$.

The main other technical piece required by our MMW algorithm is a refined divergence bound of the form (cf. Lemma 13 for a formal statement)

$$V_{\mathbf{S}}^{r^*}\left(\mathbf{S} + \eta\mathbf{G}\right) \leq \|\eta\mathbf{G}\|_{\mathrm{op}} \langle \eta\mathbf{G}, \mathbf{Y} \rangle, \text{ where } \mathbf{Y} := \nabla r^*(\mathbf{S}) \in \mathcal{Y},$$
$$\text{a strengthening of } V_{\mathbf{S}}^{r^*}\left(\mathbf{S} + \eta\mathbf{G}\right) \leq k \|\eta\mathbf{G}\|_{\mathrm{op}}^2.$$

Here, $V^{r^*}$ is the Bregman divergence in the convex conjugate of $r$. The latter bound follows easily from strong convexity of $r$ (

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

}$, we run SIFT with a reduced sample size, resulting in a $\mathrm{poly}(k)$ additive overhead in the runtime.

**Putting it all together: ListDecodableMeanEstimation.** Implicitly, the above argument assumed that we had a polynomially bounded dataset diameter; we show a simple equivalence class partitioning, PreProcess, based on one-dimensional projections efficiently yields clusters which achieve polynomially bounded diameter, so that the entire good dataset lies in the same partition (with high probability). We also give a greedy clustering step PostProcess in a low-dimensional subspace which reduces the size of the randomly sampled list to the optimal $O(k)$. Applying PreProcess, FastSIFT, and PostProcess sequentially yields our final "fast" algorithm, whose guarantees are given in Theorem 4. We also give an alternative random sampling-based procedure in Corollary 2, which trades off accuracy by roughly a $\sqrt{\log k}$ factor to remove the additive $\mathrm{poly}(k)$ term in our runtime.

## 2 Preliminaries

We give notation used in this paper in Section 2.1 and commonly-used facts in Section 2.2. We set up the list-decodable mean estimation problem and preliminary assumptions in Section 2.3.

### 2.1 Notation

**General notation.** We let $\mathcal{N}(\mu, \mathbf{\Sigma})$ denote the multivariate Gaussian distribution with specified mean and covariance, and $[d]$ denote the set of natural numbers $1 \le j \le d$. Norms and inner products are denoted by $\|\cdot\|$ and $\langle \cdot, \cdot \rangle$; when applied to a vector argument, $\|\cdot\|_p$ is the $\ell_p$ norm. The nonnegative reals are denoted $\mathbb{R}_{\ge 0}$; we also denote the (solid) probability simplex in $n$ dimensions by $\Delta^n = \{w \in \mathbb{R}^n_{\ge 0} \mid \|w\|_1 \le 1\}$. The all-ones vector in appropriate dimension is $\mathbb{1}$. Finally, unless otherwise specified all notions of approximation throughout will be multiplicative; that is, a $(1 + \epsilon)$-approximation to a quantity $\alpha$ lies in the range $[(1 - \epsilon)\alpha, (1 + \epsilon)\alpha]$.

**Matrices.** Matrices will be denoted in boldface throughout; the zero and identity matrices in appropriate dimension are $\mathbf{0}$ and $\mathbf{I}$. The set of symmetric matrices in $\mathbb{R}^{d \times d}$ is $\mathbb{S}^d$, and the positive semidefinite subset is $\mathbb{S}^d_{\ge 0}$. The Loewner order on $\mathbb{S}^d$ is denoted by $\preceq$, and $\lambda_{\max}(\cdot)$, $\lambda_{\min}(\cdot)$, and $\mathrm{Tr}(\cdot)$ are operations on $\mathbb{S}^d$ which return the largest eigenvalue, smallest eigenvalue, and trace respectively; for $k \in [d]$, the operation $\lambda_k(\cdot)$ returns the $k^{\mathrm{th}}$ largest eigenvalue of a symmetric matrix. In this paper, when applied to a matrix in $\mathbb{S}^d_{\ge 0}$, $\|\cdot\|_k$ for $k \in [d]$ is the *Ky Fan* norm, i.e. sum of the top $k$ eigenvalues. We also specially define $\|\cdot\|_{\mathrm{op}}$ and $\|\cdot\|_{\mathrm{tr}}$ to be the Ky Fan 1 and $d$ norms respectively. The inner product between symmetric matrices $\mathbf{A}$, $\mathbf{B}$ is $\langle \mathbf{A}, \mathbf{B} \rangle = \mathrm{Tr}(\mathbf{AB})$. We define the matrix exponential (on $\mathbb{S}^d$) and matrix logarithm (on $\mathbb{S}^d_{\ge 0}$) in the usual way, i.e. exp and log applied entrywise on the eigenvalues of the matrix in the appropriate basis.

**Convex analysis.** We say that twice-differentiable function $f : \mathcal{X} \to \mathbb{R}$, for $\mathcal{X} \subseteq \mathbb{R}^d$, is $\mu$-strongly convex with respect to some norm $\|\cdot\|$ if for all $x \in \mathcal{X}$ and $v \in \mathbb{R}^d$, $v^\top \nabla^2 f(x) v \succeq \mu \|v\|^2$. We say that it is $L$-smooth in $\|\cdot\|$ if its gradient is Lipschitz in the dual norm, e.g. $\|\nabla f(x) - \nabla f(x')\|_* \le L \|x - x'\|$ for all $x, x' \in \mathcal{X}$. Finally, we define the Bregman divergence, a non-Euclidean notion of distance, with respect to a convex distance-generating function $f$:

$$V_x^f(x') := f(x') - f(x) - \langle \nabla f(x), x' - x \rangle.$$

The Bregman divergence satisfies several properties which make it useful for analysis of mirror descent algorithms and their variants. In particular, it is nonnegative, convex in its argument, and

satisfies the following well-known "three-point equality":

$$\langle y - x, \nabla f(u) - \nabla f(x) \rangle = V_u^f(x) - V_u^f(y) + V_x^f(y). \tag{7}$$

**Distributions.** Let $T$ be a set of points in $\mathbb{R}^d$ with $|T| = n$, and let $w \in \Delta^n$. For any $T' \subseteq T$, $w_{T'} \in \Delta^n$ is the vector which equals $w$ on coordinates in $T'$, and is zero elsewhere. We refer to the empirical mean and covariance, parameterized by weights $w$ and subset $T' \subseteq T$, by

$$\mu_w(T') := \sum_{i \in T'} \frac{w_i}{\|w_{T'}\|_1} X_i, \; \text{Cov}_w(T') := \sum_{i \in T'} \frac{w_i}{\|w_{T'}\|_1} \left( X_i - \mu_w(T') \right) \left( X_i - \mu_w(T') \right)^\top.$$

Finally, we will also define the "unnormalized" covariance matrix by

$$\widetilde{\text{Cov}}_w(T') := \sum_{i \in T'} w_i \left( X_i - \mu_w(T') \right) \left( X_i - \mu_w(T') \right)^\top.$$

## 2.2 Useful facts

We will frequently use the following well-known facts throughout the paper. In both, $w \in \Delta^n$ is a weight vector corresponding to a set of points $T \subseteq \mathbb{R}^d$.

**Fact 1.** *We have that*

$$\mathbf{0} \preceq \sum_{i \in T} w_i (X_i - \mu_w(T))(X_i - \mu_w(T))^\top \implies \mu_w(T)\mu_w(T)^\top \preceq \sum_{i \in T} \frac{w_i}{\|w\|_1} X_i X_i^\top.$$

*Thus, for any vector $v \in \mathbb{R}^d$,*

$$(\mu_w(T) - v)(\mu_w(T) - v)^\top \preceq \sum_{i \in T} \frac{w_i}{\|w\|_1} (X_i - v)(X_i - v)^\top.$$

**Fact 2.** *For any vector $v \in \mathbb{R}^d$,*

$$\sum_{i \in [n]} w_i (X_i - v)(X_i - v)^\top = \sum_{i \in [n]} w_i (X_i - \mu_w(T))(X_i - \mu_w(T))^\top + \|w\|_1 (\mu_w(T) - v)(\mu_w(T) - v)^\top$$

$$\succeq \sum_{i \in [n]} w_i (X_i - \mu_w(T))(X_i - \mu_w(T))^\top.$$

## 2.3 List-decodable mean estimation

In the *list-decodable mean estimation* problem, we are given a set $T$ of $n$ points $\{X_i\}_{i \in T}$ in $\mathbb{R}^d$.[5] For some known $\alpha \in (0, \frac{1}{2}]$, there is a subset $S \subseteq T$ of size $\alpha n$ such that all $\{X_i\}_{i \in S}$ are independent draws from distribution $\mathcal{D}$ with mean $\mu^*$, where the covariance of $\mathcal{D}$ is identity-bounded:

$$\mathbb{E}_{x \sim \mathcal{D}} \left[ (x - \mu^*)(x - \mu^*)^\top \right] \preceq \mathbf{I}.$$

It is clear that by scaling the space, this assumption appropriately generalizes to the case when the covariance bound is $\sigma^2 \mathbf{I}$. The goal of list-decodable mean estimation is to output a list $L$, such

---

[5]In an abuse of notation, we will both let $T$ denote the set of points itself, as well as an index set for the points. Correspondingly, we will interchangeably use $X_i \in T$ and $i \in T$.

that one of the elements of the list is close to the "true mean" $\mu^*$. Our aim will be to output a list of size $|L| = O(\frac{1}{\alpha})$, which is necessary simply by identifiability of the subset $S$; it was shown as Proposition 5.4(ii) of [DKS18] that for such a list size, the minimax optimal error for the problem scales as

$$\min_{\mu \in L} \|\mu - \mu^*\|_2 = \Theta\left(\frac{1}{\sqrt{\alpha}}\right). \tag{8}$$

Regarding the sample size $n$, we additionally recall the following (note in Assumption 1 that the matrix of interest is *not* the covariance of $S$, as it is centered at the true mean $\mu^*$).

**Proposition 1** (Proposition B.1, [CSV17]). *For any constant $\epsilon \in (0,1)$, there are constants $c, C > 0$ such that with probability at least $1 - \exp(-\Omega(n))$, for $n = \frac{Cd}{\alpha}$, if an $(1+\epsilon)\alpha$ fraction of points in $\{X_i\}_{i \in T} \subseteq \mathbb{R}^d$ is drawn from $\mathcal{D}$ with covariance bounded by $c\mathbf{I}$, then Assumption 1 holds.*

**Assumption 1.** *There is a subset $S \subseteq \{X_i\}_{i \in T} \subseteq \mathbb{R}^d$ of size $\alpha n = \Theta(d)$ satisfying*

$$\frac{1}{|S|} \sum_{i \in S} (X_i - \mu^*)(X_i - \mu^*)^\top \preceq \mathbf{I}.$$

In the remainder of the paper, we will operate under Assumption 1. We will also explicitly assume that $\frac{1}{\alpha} = o(d)$, and $d \leq n = \Theta(\frac{d}{\alpha})$, for simplicity. The latter assumption is without loss of generality for any failure probability larger than $\exp(-\Omega(d))$; for any smaller failure probability, Proposition 1 implies that the assumption still holds by adjusting the sample size by a logarithmic factor. It is also fairly straightforward to see that the former assumption is also without loss of generality, since in the case $\frac{1}{\alpha} \gg d$, it suffices to sample $O(\frac{1}{\alpha} \log \frac{1}{\delta})$ random points and apply a variant of the post-processing procedure of Section 5.1 to obtain the correct list size and error guarantee; we give a formal treatment of this case in Appendix A.

Finally, throughout the variable $k$ will be reserved for values which are $\Theta(\frac{1}{\alpha})$ for explicitly stated constants. In particular, many of our algorithms will rely on performing operations such as principal components analysis in $\Theta(\frac{1}{\alpha})$ dimensions. As discussed earlier, this is because a substantial portion of the challenge in the estimation problem is reducing to the problem of learning the mean in $\Theta(\frac{1}{\alpha})$ dimensions, at which point naïve random sampling solves the problem up to logarithmic factors.

## 3 Filtering in $k$ dimensions: SIFT

In this section, we develop a simple, polynomial-time algorithm for solving the list-decodable mean estimation problem based on a "soft downweighting" approach. We outline some preliminary notions and bounds used in our algorithms and analysis in Section 3.1, which will also be used in Sections 4 and 5. We then use these tools to analyze our "slow" algorithm, SIFT, in Section 3.2.

### 3.1 Filtering preliminaries

We define two concepts which will be useful in stating guarantees of our downweighting methods.

**Definition 1** (Saturated weights). *We call weights $w \in \Delta^n$ "saturated" if $w \leq \frac{1}{n}\mathbb{1}$ entrywise, and*

$$\|w_S\|_1 \geq \alpha\sqrt{\|w\|_1}.$$

**Definition 2** (Safe scores). *We call scores $\{\tau_i\}_{i \in T} \in \mathbb{R}_{\geq 0}^n$ "safe with respect to $w \in \Delta^n$" if*

$$\sum_{i \in S} \frac{w_i}{\|w_S\|_1} \tau_i \leq \frac{1}{2} \sum_{i \in T} \frac{w_i}{\|w\|_1} \tau_i.$$

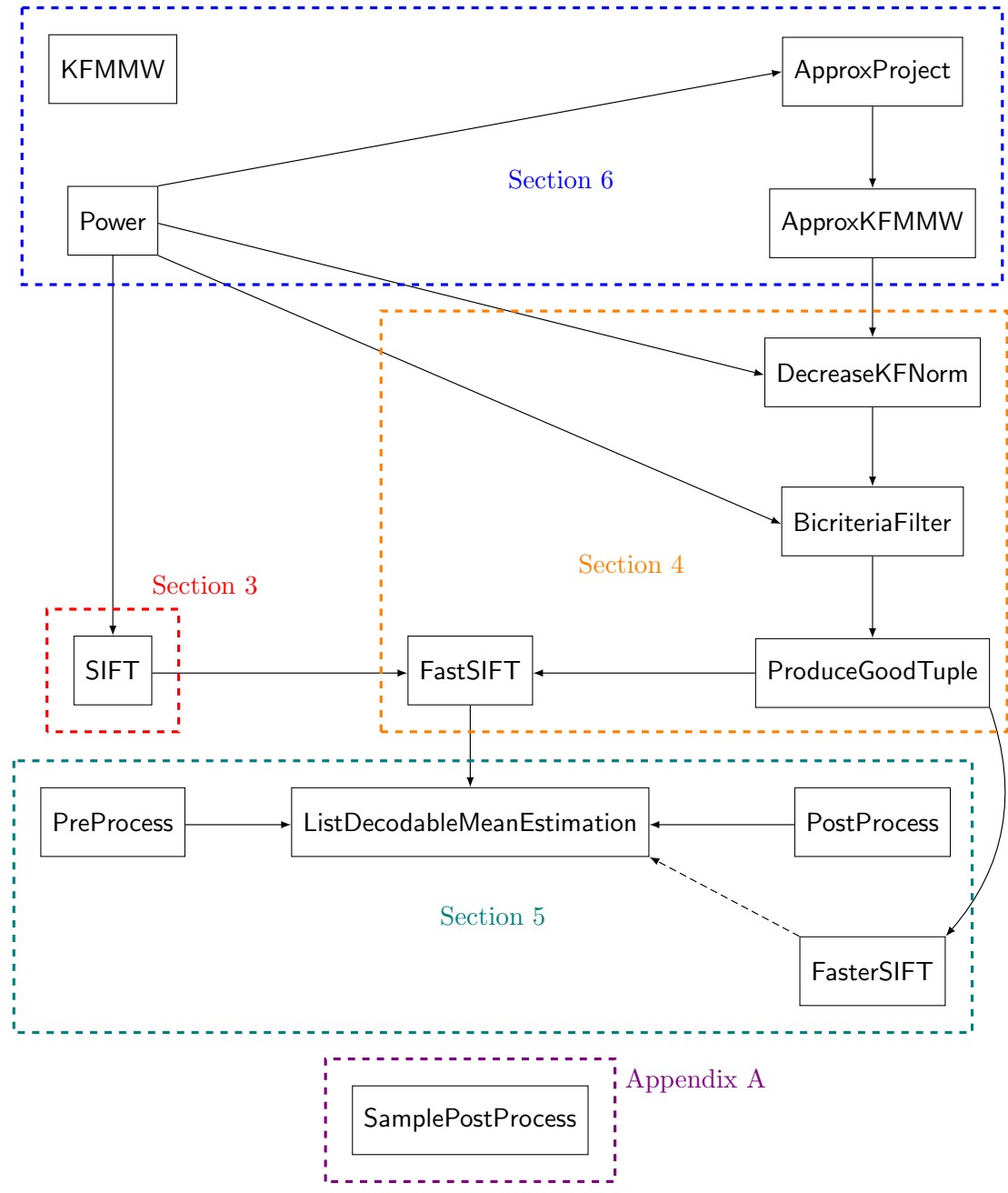

Figure 1: A picture of the dependencies of the different algorithms of this paper together with which section the algorithms are described in. Note that we have a dashed arrow from FasterSIFT to ListDecodableMeanEstimation since we present two different versions of ListDecodableMeanEstimation, where only the second (alternative) version uses FasterSIFT.

*When the weights $w$ are clear from context, we will simply call the scores $\tau$ "safe".*

In algorithms based on soft filtering in the presence of a small amount of adversarial noise (see e.g. [DKK⁺17, Li18, Ste18]), a typical goal is to remove more "good weight" than "bad weight" from an iteratively updated weight vector. However, when the overwhelming majority of the initial weight is bad, clearly this is too strong of a goal. The intuition for Definition 1 is that a weaker goal suffices

for the guarantees of our methods; while the amount of good weight is decreasing throughout, Definition 1 requires that the good weight becomes more saturated in the weight vector when more weight is removed. We now make the connection between these definitions formal.

**Lemma 1.** *Consider a set of saturated (cf. Definition 1) weights $w^{(0)}$, and updates of the form:*

1. *For $0 \le t < N$:*
   (a) *Let $\left\{\tau_i^{(t)}\right\}_{i \in T}$ be safe (cf. Definition 2) with respect to $w^{(t)}$.*
   (b) *Update for all $i \in T$:*

$$w_i^{(t+1)} \leftarrow \left(1 - \frac{\tau_i^{(t)}}{\tau_{\max}^{(t)}}\right) w_i^{(t)}, \ \ where \ \tau_{\max}^{(t)} := \max_{i \in T \mid w_i^{(t)} \ne 0} \tau_i^{(t)}. \tag{9}$$

*Then, the result of the updates $w^{(N)}$ is also saturated.*

*Proof.* First, fix some iteration $t$, and let $w := w^{(t)}$, $\tau := \tau^{(t)}$, and $w' := w^{(t+1)}$. Define

$$\delta_S := \sum_{i \in S} \frac{w_i - w_i'}{\|w_S\|_1}, \ \delta_T := \sum_{i \in T} \frac{w_i - w_i'}{\|w\|_1}.$$

Note that by the assumption that $\tau$ is safe and the iteration (9),

$$\delta_S = \frac{1}{\tau_{\max}} \sum_{i \in S} \frac{w_i}{\|w_S\|_1} \tau_i \le \frac{1}{2\tau_{\max}} \sum_{i \in T} \frac{w_i}{\|w\|_1} \tau_i = \frac{1}{2} \delta_T.$$

Hence, using $1 - \frac{1}{2}\delta_T \ge \sqrt{1 - \delta_T}$ for all $\delta_T \in [0, 1]$, we have

$$\frac{\left\|w_S^{(t+1)}\right\|_1}{\left\|w_S^{(t)}\right\|_1} = 1 - \delta_S \ge \sqrt{1 - \delta_T} = \sqrt{\frac{\left\|w_T^{(t+1)}\right\|_1}{\left\|w_T^{(t)}\right\|_1}}. \tag{10}$$

Inductively telescoping (10), using that $w^{(0)}$ was assumed to be saturated, and finally comparing with Definition 1, yields the desired conclusion that $w^{(N)}$ is saturated. $\square$

We next give three helper lemmas which help reason about how the quality of empirical estimates based on $S$ deteriorate, as the amount of weight allocated to $S$ is reduced. The first shows how the quality of the empirical mean is related to the empirical covariance and proportion of weight in $S$ (and is essentially a rephrasing of Fact A.3 in [CMY20]).

**Lemma 2.** *Let $w \in \Delta^n$ have $w \le \frac{1}{n}\mathbb{1}$ entrywise. Then,*

$$\|\mu_w(T) - \mu^*\|_2 \le \sqrt{2 \|\mathrm{Cov}_w(T)\|_{\mathrm{op}} \frac{\|w\|_1}{\|w_S\|_1} + \frac{2\alpha}{\|w\|_1}}.$$

*Proof.* Let $w^* \in \Delta^n$ be the weight vector which is $\frac{1}{|S|}$ on coordinates in $S$, and zero elsewhere. Note

that by definition $\langle w, w^* \rangle = \frac{\|w_S\|_1}{\alpha n}$. Next,

$$\|\mu_w(T) - \mu^*\|_2^2 = \max_{\|u\|_2=1} \left\langle \left( \sum_{i \in T} \frac{w_i w_i^*}{\langle w, w^* \rangle} (X_i - \mu_w(T)) \right) - \left( \sum_{i \in T} \frac{w_i w_i^*}{\langle w, w^* \rangle} (X_i - \mu^*) \right), u \right\rangle^2$$

$$\leq 2 \max_{\|u\|_2=1} \left\langle \sum_{i \in T} \frac{w_i w_i^*}{\langle w, w^* \rangle} (X_i - \mu_w(T)), u \right\rangle^2 + 2 \max_{\|u\|_2=1} \left\langle \sum_{i \in T} \frac{w_i w_i^*}{\langle w, w^* \rangle} (X_i - \mu^*), u \right\rangle^2.$$

We bound these two terms separately. First, by applying a quadratic form in $u$ to Fact 1,

$$\max_{\|u\|_2=1} \left\langle \sum_{i \in T} \frac{w_i w_i^*}{\langle w, w^* \rangle} (X_i - \mu_w(T)), u \right\rangle^2 \leq \max_{\|u\|_2=1} \sum_{i \in T} \frac{w_i w_i^*}{\langle w, w^* \rangle} \langle X_i - \mu_w(T), u \rangle^2$$

$$= \frac{\|w\|_1}{\alpha n \langle w, w^* \rangle} \max_{\|u\|_2=1} \sum_{i \in T} \frac{w_i}{\|w\|_1} \langle X_i - \mu_w(T), u \rangle^2$$

$$= \|\mathrm{Cov}_w(T)\|_{\mathrm{op}} \frac{\|w\|_1}{\|w_S\|_1}.$$

Next, by again applying Fact 1, and recalling Assumption 1,

$$\max_{\|u\|_2=1} \left\langle \sum_{i \in T} \frac{w_i w_i^*}{\langle w, w^* \rangle} (X_i - \mu^*), u \right\rangle^2 \leq \max_{\|u\|_2=1} \sum_{i \in T} \frac{w_i w_i^*}{\langle w, w^* \rangle} \langle X_i - \mu^*, u \rangle^2$$

$$\leq \frac{\|w\|_\infty}{\langle w, w^* \rangle} \max_{\|u\|_2=1} \sum_{i \in T} w_i^* \langle X_i - \mu^*, u \rangle^2 \leq \frac{\alpha n \|w\|_\infty}{\|w\|_1}.$$

$\square$

The second shows how the empirical covariance of $S$ grows relative to how much of $S$ is kept.

**Lemma 3.** *Let $w \in \Delta^n$ have $w \leq \frac{1}{n}\mathbb{1}$ entrywise. Then $\mathrm{Cov}_w(S) \preceq \frac{\alpha}{\|w_S\|_1}\mathbf{I}$.*

*Proof.* For any vector $u$ with $\|u\|_2 = 1$,

$$u^\top \mathrm{Cov}_w(S) u = \sum_{i \in S} \frac{w_i}{\|w_S\|_1} \langle u, X_i - \mu_w(S) \rangle^2$$

$$\leq \sum_{i \in S} \frac{\alpha w_i^*}{\|w_S\|_1} \langle u, X_i - \mu^* \rangle^2 \leq \frac{\alpha}{\|w_S\|_1} \left\| \sum_{i \in S} w_i^* (X_i - \mu^*)(X_i - \mu^*)^\top \right\|_{\mathrm{op}}.$$

In the second line we used Fact 2. Using Assumption 1 yields the conclusion. $\square$

The third shows how a bound on the saturation of $S$ in a weight vector can be used to bound the distance between empirical means in $S$ and $T$ via the empirical covariance matrix.

**Lemma 4.** *We have that*

$$(\mu_w(S) - \mu_w(T))(\mu_w(S) - \mu_w(T))^\top \preceq \frac{\|w\|_1}{\|w_S\|_1} \mathrm{Cov}_w(T).$$

*Proof.* This follows from the following observations (via Fact 1)

$$(\mu_w(S) - \mu_w(T))(\mu_w(S) - \mu_w(T))^\top \preceq \sum_{i \in S} \frac{w_i}{\|w_S\|_1} (X_i - \mu_w(T))(X_i - \mu_w(T))^\top$$

$$\preceq \frac{\|w\|_1}{\|w_S\|_1} \sum_{i \in T} \frac{w_i}{\|w\|_1} (X_i - \mu_w(T))(X_i - \mu_w(T))^\top$$

$$= \frac{\|w\|_1}{\|w_S\|_1} \mathrm{Cov}_w(T).$$

$\square$

## 3.2 Analysis of SIFT

We now present SIFT as Algorithm 1. It requires calls to an approximate $k$-PCA subroutine Power, the classical simultaneous power iteration method, which is stated as Algorithm 11 in Section 6.3, where we present an improved analysis of its guarantees. However, for analysis in this section it suffices to use the following guarantee. For simplicity in this section we drop the arguments $\lambda_{\max}$ and $\lambda_{\min}$ as inputs to Power, which do not play a role in Proposition 2.

**Proposition 2** (Theorem 1, [MM15])**.** *For any $\delta \in (0, 1)$ and $k \in [d]$, there is an algorithm, Power, which takes as input $k$, $\delta$, $\mathbf{A} \in \mathbb{S}_{\geq 0}^d$ and $\epsilon \in (0, 1)$, and returns with probability $1 - \delta$ a set of orthonormal vectors $\mathbf{V} \in \mathbb{R}^{d \times k}$ such that if $\mathbf{V}_{:i}$ is column $i$ of $\mathbf{V}$,*

$$\langle \mathbf{V}_{:i}, \mathbf{A}\mathbf{V}_{:i} \rangle \in [1 - \epsilon, 1 + \epsilon] \lambda_i(\mathbf{A}) \text{ for all } i \in [k],$$

$$\text{and } \left\| \left( \mathbf{I} - \mathbf{V}\mathbf{V}^\top \right) \mathbf{A} \left( \mathbf{I} - \mathbf{V}\mathbf{V}^\top \right) \right\|_{\mathrm{op}} \leq (1 + \epsilon)\lambda_{k+1}(\mathbf{A}).$$

*When $\mathbf{A}$ is given in the form $\mathbf{M}^\top \mathbf{M}$ for some $\mathbf{M} \in \mathbb{R}^{n \times d}$, the runtime of Power is*

$$O\left( \frac{ndk}{\epsilon} \log\left( \frac{d}{\delta\epsilon} \right) \right).$$

Note that Lines 6 through 10 of Algorithm 1 exactly constitute a weight removal method of the form given in Lemma 1. Consequently, to use Lemma 1 it suffices to prove that the weights $\tau_i$ used in each iteration are safe with respect to the current set of weights, which we now demonstrate.

**Lemma 5.** *In each iteration $t$ of Algorithm 1 until termination, $\tau^{(t)}$ is safe with respect to $w^{(t)}$.*

*Proof.* Throughout this proof, let $w := w^{(t)}$ and $\tau := \tau^{(t)}$. Furthermore, let $\mathbf{V}$, $\mathbf{\Sigma}$, and $\beta$ correspond to the weights $w$ at the iteration's start. We will inductively prove that $\tau$ is safe with respect to $w$, which by applying Lemma 1 implies that at the start of the iteration, $w$ is saturated (since clearly

**Algorithm 1** SIFT$(T, \delta)$

---

1: **Input:** $T \subset \mathbb{R}^d$ with $|T| = n$ satisfying Assumption 1, $\delta \in (0, 1)$
2: $w^{(0)} \leftarrow \frac{1}{n} \mathbb{1}_T$, $t \leftarrow 0$, $\beta \leftarrow 1$, $k \leftarrow \lceil \frac{4}{\alpha} \rceil$
3: $\mathbf{V} \leftarrow \mathsf{Power}(\mathrm{Cov}_{w^{(t)}}(T), k, 0.2, \frac{\delta}{2n})$
4: $\mathbf{\Sigma} \leftarrow \mathbf{V}^\top \mathrm{Cov}_{w^{(t)}}(T) \mathbf{V}$
5: **while** $\lambda_k(\mathbf{\Sigma}) \geq \frac{4}{\sqrt{\beta}}$ **do**
6: $\quad \tau_i^{(t)} \leftarrow \left\| \mathbf{\Sigma}^{-\frac{1}{2}} \mathbf{V}^\top (X_i - \mu_w(T)) \right\|_2^2$ for all $i \in T$
7: $\quad w_i^{(t+1)} \leftarrow \left( 1 - \frac{\tau_i^{(t)}}{\tau_{\max}^{(t)}} \right) w_i^{(t)}$ for all $i \in T$, where $\tau_{\max}^{(t)} := \max_{i \in T | w_i^{(t)} \neq 0} \tau_i^{(t)}$
8: $\quad t \leftarrow t + 1$, $\beta \leftarrow \left\| w^{(t)} \right\|_1$
9: $\quad \mathbf{V} \leftarrow \mathsf{Power}(\mathrm{Cov}_{w^{(t)}}(T), k, 0.2, \frac{\delta}{2n})$
10: $\quad \mathbf{\Sigma} \leftarrow \mathbf{V}^\top \mathrm{Cov}_{w^{(t)}}(T) \mathbf{V}$
11: **end while**
12: **return** $L := \{ \mathbf{V}\mathbf{V}^\top X_i + (\mathbf{I} - \mathbf{V}\mathbf{V}^\top) \mu_{w^{(t)}}(T)$ where $i \in T$ is sampled uniformly at random$\}$,
    with list size $|L| = \lceil \frac{2}{\alpha} \log \frac{2}{\delta} \rceil$

---

$w^{(0)}$ is saturated). We first compute the average score in $S$:

$$
\begin{aligned}
\sum_{i \in S} \frac{w_i}{\|w_S\|_1} \tau_i &= \sum_{i \in S} \frac{w_i}{\|w_S\|_1} \left\| \mathbf{\Sigma}^{-\frac{1}{2}} \mathbf{V}^\top (X_i - \mu_w(T)) \right\|_2^2 \\
&= \sum_{i \in S} \frac{w_i}{\|w_S\|_1} \left( \left\| \mathbf{\Sigma}^{-\frac{1}{2}} \mathbf{V}^\top (X_i - \mu_w(S)) \right\|_2^2 + \left\| \mathbf{\Sigma}^{-\frac{1}{2}} \mathbf{V}^\top (\mu_w(S) - \mu_w(T)) \right\|_2^2 \right) \\
&= \left\langle \mathbf{\Sigma}^{-1}, \mathbf{V}^\top \mathrm{Cov}_w(S) \mathbf{V} \right\rangle + \left\| \mathbf{\Sigma}^{-\frac{1}{2}} \mathbf{V}^\top (\mu_w(S) - \mu_w(T)) \right\|_2^2 \\
&\leq \left\langle \mathbf{\Sigma}^{-1}, \frac{\alpha}{\|w_S\|_1} \mathbf{I} \right\rangle + \frac{\|w\|_1}{\|w_S\|_1} \leq \frac{1}{4} \left\langle \sqrt{\beta} \mathbf{I}, \frac{1}{\sqrt{\beta}} \mathbf{I} \right\rangle + \frac{\sqrt{\beta}}{\alpha} \leq \frac{k}{2}.
\end{aligned}
$$

The first three equalities follow by expanding definitions; the first inequality is by Lemmas 3 and 4, as well as the definition of $\mathbf{\Sigma}$. The second inequality is by using the definition of saturated weights (Definition 1) twice, which implies that $\|w_S\|_1 \geq \alpha\sqrt{\beta}$, as well as the exit condition in Line 5. The third inequality follows from the definition of $k$. Finally, we conclude that $\tau$ is indeed safe, since the average score in $T$ is exactly $k$ by design:

$$
\begin{aligned}
\sum_{i \in T} \frac{w_i}{\|w\|_1} \tau_i &= \sum_{i \in T} \frac{w_i}{\|w\|_1} \left\| \mathbf{\Sigma}^{-\frac{1}{2}} \mathbf{V}^\top (X_i - \mu_w(T)) \right\|_2^2 \\
&= \left\langle \mathbf{\Sigma}^{-1}, \mathbf{V}^\top \left( \sum_{i \in T} \frac{w_i}{\|w\|_1} (X_i - \mu_w(T))(X_i - \mu_w(T))^\top \right) \mathbf{V} \right\rangle = \left\langle \mathbf{\Sigma}^{-1}, \mathbf{\Sigma} \right\rangle = k.
\end{aligned}
$$

$\square$

Finally, we prove a runtime and correctness guarantee on Algorithm 1.

**Theorem 3.** *Under Assumption 1, with probability $1 - \delta$, the output of Algorithm 1 satisfies*

$$\min_{\mu \in L} \|\mu - \mu^*\|_2^2 \leq \frac{22}{\alpha}.$$

*The overall runtime of Algorithm 1 is*

$$O\left(n^2 dk \log\left(\frac{d}{\delta}\right)\right).$$

*Proof.* We will show correctness and complexity of Algorithm 1 separately.

*Complexity guarantee.* It is clear that there are at most $n$ iterations in Algorithm 1, since at least one weight is zeroed out in Line 7 each iteration. Further, the bottleneck operation in each iteration is clearly the complexity of Power, since an eigendecomposition of $\mathbf{\Sigma}$ takes time $O(k^3) = O(ndk)$. Since $\epsilon$ is a constant in Proposition 2 and $n = O(d^2)$, this yields the complexity bound. Using a union bound, with probability $1 - \frac{\delta}{2}$, the conclusion of Proposition 2 applies in every iteration; we will condition on this event for the remainder of the proof.

We finally note that the algorithm must terminate the while loop before removing all the weight. This is because throughout the algorithm since $w$ is saturated (by Lemmas 1 and 5), $\|w\|_1 \geq \alpha^2$ holds directly by using Definition 1 and $\|w\|_1 \geq \|w_S\|_1$.

*Correctness guarantee.* As in Lemma 5, we let $w$ denote the weights on the last iteration of the algorithm (after exiting on Line 12). Denote $\mathbf{P} := \mathbf{V}\mathbf{V}^\top$ and $Y_i := \mathbf{P}X_i$ for all $i \in T$. Since

$$\sum_{i \in S} \frac{1}{\alpha n} \left(Y_i - \mathbf{P}\mu^*\right)\left(Y_i - \mathbf{P}\mu^*\right)^\top = \mathbf{P}\left(\sum_{i \in S} \frac{1}{\alpha n}\left(X_i - \mu^*\right)\left(X_i - \mu^*\right)^\top\right)\mathbf{P} \preceq \mathbf{P},$$

by Assumption 1, the expectation of $\|Y_i - \mathbf{P}\mu^*\|_2^2$ for a uniformly random sample $i \in S$ is $\frac{4}{\alpha}$ by linearity of trace. Hence, by Markov with probability at least $\frac{1}{2}$ a sample from $S$ has $\|Y_i - \mathbf{P}\mu^*\|_2^2 \leq \frac{8}{\alpha}$, so with probability at least $1 - \frac{\delta}{2}$, one of the random samples in $L$ will have an $X_i$ with $\|Y_i - \mathbf{P}\mu^*\|_2^2 \leq \frac{8}{\alpha}$. For this value of $i$, we expand via the Pythagorean theorem

$$\|(\mathbf{P}X_i + (\mathbf{I} - \mathbf{P})\mu_w(T)) - \mu^*\|_2^2 = \|Y_i - \mathbf{P}\mu^*\|_2^2 + \|(\mathbf{I} - \mathbf{P})(\mu_w(T) - \mu^*)\|_2^2$$
$$\leq \frac{8}{\alpha} + \|(\mathbf{I} - \mathbf{P})(\mu_w(T) - \mu^*)\|_2^2.$$

To bound this second term, we apply Lemma 2 on the set of points $\{(\mathbf{I} - \mathbf{P})X_i\}_{i \in T}$. This implies

$$\|(\mathbf{I} - \mathbf{P})(\mu_w(T) - \mu^*)\|_2^2 \leq \frac{2\beta}{\|w_S\|_1}\|(\mathbf{I} - \mathbf{P})\mathrm{Cov}_w(T)(\mathbf{I} - \mathbf{P})\|_{\mathrm{op}} + \frac{2\alpha}{\beta}$$
$$\leq \frac{12\sqrt{\beta}}{\|w_S\|_1} + \frac{2\alpha}{\beta} \leq \frac{14}{\alpha}.$$

Here, the last inequality used the definition of saturation, which also implies that $\beta \geq \alpha^2$. The second inequality used that the guarantees of Power and the termination condition imply that

$$\|(\mathbf{I} - \mathbf{P})\mathrm{Cov}_w(T)(\mathbf{I} - \mathbf{P})\|_{\mathrm{op}} \leq 1.2\lambda_k(\mathrm{Cov}_w(T)) \leq 1.5\lambda_k(\mathbf{\Sigma}) \leq \frac{6}{\sqrt{\beta}}.$$

Here, we use that the eigenvalues of $\mathbf{\Sigma}$ are the same as those of $\mathbf{V}\mathbf{V}^\top \mathrm{Cov}_w(T)\mathbf{V}\mathbf{V}^\top$; this calculation is given in the correctness proof of Proposition 8. Combining the above bounds yields the conclusion.

$\square$

While Theorem 3 achieves the desired error guarantee (8), it unfortunately has a quadratic dependence on the sample complexity $n$, as well as a suboptimal list size by a factor of $O(\log \frac{1}{\delta})$. We address the latter issue with a post-processing step in Section 5.1; regarding the former issue, Algorithm 1 will play a role in our final "fast" algorithm in the following Section 4, which obtains a runtime with a linear dependence on $n$ via more sophisticated weight removal.

## 4 Fast filtering in $k$ dimensions under a diameter bound

We now give an algorithm, FastSIFT, with an improved dependence on the sample size $n$ compared to the method SIFT developed in Section 3. We use the following assumption in this section.

**Assumption 2.** *All data points in $T$ lie in a Euclidean ball of radius $R$.*

We eventually show how to reduce the more general mean estimation problem to mean estimation on datasets satisfying Assumption 2 in Section 5.2 to obtain our final algorithm. The primary goal of this section is to develop a method for quickly finding a "good" tuple $(\mathbf{B}, w)$, defined as follows.

**Definition 3** (Good tuple). *We call $(\mathbf{B}, w)$ "good" if it obeys the following conditions.*

1. $\mathbf{B} \in \mathbb{R}^{d \times k'}$ *has orthogonal columns, for some $k' = O(\frac{\log R}{\alpha})$, and $w \in \Delta^n$ is saturated.*

2. *Let $\mathbf{P_B} := \mathbf{B}\mathbf{B}^\top$. The restriction of $\mathrm{Cov}_w(T)$ to the complement of $\mathbf{P_B}$, denoted by*

$$\mathrm{Cov}_w^{\mathbf{P_B^\perp}}(T) := (\mathbf{I} - \mathbf{P_B})\,\mathrm{Cov}_w(T)\,(\mathbf{I} - \mathbf{P_B})$$

*satisfies for a universal constant $c$,*

$$\left\| \mathrm{Cov}_w^{\mathbf{P_B^\perp}}(T) \right\|_{\mathrm{op}} \le \frac{c}{\sqrt{\|w\|_1}}.$$

Intuitively, a good tuple signifies that in all but $O(\frac{\log R}{\alpha})$ dimensions, we have learned the mean via the guarantee of Lemma 2. However, in the remaining dimensions we can simply run the algorithm of Section 3, which obtains an additive $\mathrm{poly}(k)$ runtime dependence. We now make this rigorous.

---

**Algorithm 2** FastSIFT$(T, \delta, \mathsf{ProduceGoodTuple})$

1: **Input:** $T = T_{\mathrm{fast}} \cup T_{\mathrm{slow}} \subset \mathbb{R}^d$ with $|T_{\mathrm{fast}}| = n$ satisfying Assumptions 1 and 2, $|T_{\mathrm{slow}}| = O(\frac{\log R}{\alpha^2})$ satisfying Assumption 1 for a fixed $O(\frac{\log R}{\alpha})$-dimensional subspace, $\delta \in (0, 1)$, subroutine ProduceGoodTuple which returns a good tuple with specified failure probability
2: $(\mathbf{B}, w) \leftarrow \mathsf{ProduceGoodTuple}(T_{\mathrm{fast}}, \frac{\delta}{2})$
3: $\mu_{\mathrm{fast}} \leftarrow (\mathbf{I} - \mathbf{B}\mathbf{B}^\top)\mu_w(T)$
4: $L_{\mathrm{slow}} \leftarrow \mathsf{SIFT}(\{\mathbf{B}\mathbf{B}^\top X_i \mid X_i \in T_{\mathrm{slow}}\}, \frac{\delta}{2})$
5: **return** $L \leftarrow \{\mu_{\mathrm{slow}} + \mu_{\mathrm{fast}} \mid \mu_{\mathrm{slow}} \in L_{\mathrm{slow}}\}$

---

**Lemma 6.** *With probability $1 - \delta$, some $\hat{\mu} \in L$ outputted by Algorithm 2 satisfies*

$$\|\hat{\mu} - \mu^*\|_2^2 \le \frac{48 + 4c}{\alpha}.$$

*The overall runtime of Algorithm 2 is the cost of running* ProduceGoodTuple$(T_{\mathrm{fast}}, \frac{\delta}{2})$ *plus*

$$O\left(\frac{d}{\alpha^3}\log(R)\log\left(\frac{dR}{\delta}\right) + \frac{1}{\alpha^6}\log^3(R)\log\left(\frac{d}{\delta}\right)\right) \text{ additional runtime overhead.}$$

*Proof.* By the proof of Theorem 3 and the second part of Definition 3, it is immediate that

$$\|(\mathbf{I} - \mathbf{P_B})(\mu_w(T) - \mu^*)\|_2^2 \le \frac{2c + 2}{\alpha}.$$

Moreover, since the size of $T_{\mathrm{slow}}$ is large enough for Proposition 1 to apply, it satisfies Assumption 1 on the $k'$-dimensional subspace whose projection matrix is $\mathbf{P_B} = \mathbf{BB}^\top$. Thus, Theorem 3 shows

$$\|\mu_{\mathrm{slow}} - \mathbf{P_B}\mu^*\|_2^2 \le \frac{22}{\alpha} \text{ for some } \mu_{\mathrm{slow}} \in L_{\mathrm{slow}}.$$

Combining these two bounds and the Pythagorean theorem yields the correctness guarantee. For the runtime overhead guarantee, it is clear the bottleneck operation is Line 4 since Line 3 can be implemented in time $O(\frac{d}{\alpha}\log R)$. For Line 4, we run Algorithm 1 entirely in the coordinate system of the columns of $\mathbf{B}$, which is isomorphic to $\mathbb{R}^{k'}$, and then left-multiply the resulting list by $\mathbf{B}$. Forming the input set $\{\mathbf{B}^\top X_i \mid X_i \in T_{\mathrm{slow}}\}$ takes time $O(|T_{\mathrm{slow}}|k'd) = O(\frac{d}{\alpha^3}\log^2 R)$; multiplying the resulting output list by $\mathbf{B}$ cannot be the dominant cost by more than a $\log\frac{1}{\delta}$ factor. $\square$

Here, we note that because we take $n = \Omega(d\alpha^{-1}) = \Omega(\alpha^{-2})$ in accordance with Proposition 1, the cost of $O(d\alpha^{-3}\log R\log\frac{dR}{\delta})$ incurred by Lemma 6 is no more than the cost of logarithmically many $k$-PCAs on the original dataset. Regarding the separation of the original dataset into $T_{\mathrm{fast}}$ and $T_{\mathrm{slow}}$, which appropriately satisfy Assumption 1, we make the following comment.

**Remark 1.** *We can form a partitioned dataset $T = T_{\mathrm{fast}} \cup T_{\mathrm{slow}}$ of the form required by Algorithm 2 by independently drawing $n$ samples to form $T_{\mathrm{fast}}$, $O(\frac{\log R}{\alpha^2})$ samples to form $T_{\mathrm{slow}}$, and applying Assumption 1 to $T_{\mathrm{fast}}$ and the projection of $T_{\mathrm{slow}}$ into a $k'$-dimensional subspace. Up to a $\log\frac{1}{\delta}$ factor in the sample complexity (for error probabilities which are smaller than $\exp(-\Omega(\alpha^{-1}))$), these are valid applications of Assumption 1 because of independence; in particular, the draws $T_{\mathrm{slow}}$ are independent of the $k'$-dimensional subspace learned by running* ProduceGoodTuple *on $T_{\mathrm{fast}}$, which only depends on randomness used in Step 2 of* FastSIFT.

We now state our strategy for the implementation of ProduceGoodTuple. Roughly speaking, ProduceGoodTuple is a composition of three subroutines at different levels, named BicriteriaFilter, DecreaseKFNorm, and KFMMW. Each subroutine is associated with one or more potential functions which show that the subroutine "one level down" is called $O(\log d)$ times.

1. ProduceGoodTuple iteratively calls BicriteriaFilter, an algorithm which takes as input saturated weights $w$ and either produces saturated weights $\|w'\|_1 \le \frac{1}{2}\|w\|_1$, or a good tuple.

2. BicriteriaFilter iteratively calls DecreaseKFNorm, an algorithm which takes as input saturated weights $w$ and maintains an updated set of orthogonal vectors $\mathbf{B}$. Each call to DecreaseKFNorm either (1) halves the $\ell_1$ norm of $w$, (2) halves the Ky Fan $k$ norm of the covariance matrix, or (3) decreases the operator norm of the covariance matrix by a constant factor and adds $k$ vectors to $\mathbf{B}$, for some $k = \Theta(\frac{1}{\alpha})$.

3. DecreaseKFNorm is based on a "win-win-win" analysis of the fine-grained guarantees of a Ky Fan norm matrix multiplicative weights procedure, developed in Section 6. We will show that

in $O(\log d)$ iterations of KFMMW, either the Ky Fan $k$ norm has halved, or one of the other two "exit conditions" required by DecreaseKFNorm has been certifiably met.

Given the guarantees of DecreaseKFNorm, correctness of ProduceGoodTuple and BicriteriaFilter follow straightforwardly. Thus, in Section 4.1, we state and prove a performance guarantee on DecreaseKFNorm, which we use to give a simple analysis of ProduceGoodTuple in Section 4.2. Combining our analysis of ProduceGoodTuple with Lemma 6 gives the main export from this section. Finally, we note that in the following development of ProduceGoodTuple and its subroutines, we will overload the input set $T$ to be $T_{\text{fast}}$ in Algorithm 2, because it is the input to ProduceGoodTuple.

### 4.1 Analysis of DecreaseKFNorm

We first state a guarantee for ApproxKFMMW as Proposition 3, which is a computationally efficient variant of KFMMW (these methods are both given and analyzed in Section 6). Proposition 3 is a restatement of Corollary 4 and Lemma 20 with $\Delta = \frac{1}{200}$, which are proven in Section 6.4.

**Proposition 3.** *There is an algorithm, ApproxKFMMW (Algorithm 13), which takes as input a sequence of matrices $\{\mathbf{G}_t\}_{t \geq 0} \subset \mathbb{S}_{\geq 0}^d$ each in the form $\mathbf{M}_t^\top \mathbf{M}_t$ for $\mathbf{M}_t \in \mathbb{R}^{n \times d}$ for explicitly given $\mathbf{M}_t$, and $k \in [d]$. Suppose that the matrices $\{\mathbf{G}_t\}_{t \geq 0}$ are weakly decreasing in Loewner order, and let $\eta \leq \frac{1}{2\|\mathbf{G}_0\|_{\text{op}}}$. For any $N \geq 1$, with probability $1 - \delta'$, ApproxKFMMW defines a sequence of matrices $\{\widehat{\mathbf{Y}}_t\}_{0 \leq t < N}$, where $\widehat{\mathbf{Y}}_t$ only depends on $\{\mathbf{G}_s\}_{0 \leq s < t}$, such that*

$$\|\mathbf{G}_N\|_k \leq \frac{2}{T} \sum_{t=0}^{N-1} \left\langle \mathbf{G}_t, \widehat{\mathbf{Y}}_t \right\rangle + \frac{k \log d}{\eta N} + \frac{k}{200\eta}.$$

*Each $\widehat{\mathbf{Y}}_t$ satisfies $\left\|\widehat{\mathbf{Y}}_t\right\|_{\text{op}} \leq 1.01$ and $\left\|\widehat{\mathbf{Y}}_t\right\|_{\text{tr}} \leq 1.01k$. The cost of the algorithm is*

$$O\left(ndkN^2 \log^2\left(\frac{dN}{\delta'}\right)\right).$$

*Furthermore, for any set of $n$ fixed vectors $\{v_i\}_{i \in [n]} \subset \mathbb{R}^d$ and any iteration $t$, 1.05-approximations to all $v_i^\top \widehat{\mathbf{Y}}_t v_i$ can be computed in time*

$$O\left(ndN \log\left(\frac{nd}{\delta'}\right)\right) \quad \text{with probability at least } 1 - \delta'.$$

We are now ready to state the algorithm DecreaseKFNorm as Algorithm 3. At a high level, the goal of DecreaseKFNorm is to implement Proposition 3 in a way so that each of the inner products $\left\langle \mathbf{G}_t, \widehat{\mathbf{Y}}_t \right\rangle$ is sufficiently small, via decreasing weights defined in terms of the matrix $\widehat{\mathbf{Y}}_t$. We will be able to successfully do this as long as the $\ell_1$ norm of the weight remains stable, and the top eigenvalue of the covariance matrix is not too much larger than the $k^{\text{th}}$ largest. When either of these conditions fail, we will exit the algorithm via a different termination condition.

The first step in the analysis of Algorithm 3 is to guarantee that any time a weight removal procedure is performed, it is with respect to safe scores, and hence the weights remain saturated throughout the course of the algorithm. We give this proof of safe weight removal as Lemma 7, and then an overall correctness and runtime guarantee in Proposition 4.

**Lemma 7.** *Throughout the course of Algorithm 3, any time weight removal is performed in Line*

---

**Algorithm 3** DecreaseKFNorm$(T, w, \gamma, \delta)$

---

1: **Input:** $T \subset \mathbb{R}^d$ with $|T| = n$ satisfying Assumptions 1 and 2, saturated $w$, $\gamma \leftarrow 1.05$-approximation to $\|\mathrm{Cov}_w(T)\|_k$ with probability at least $1 - \frac{\delta}{3(N+1)}$ for $k := \lceil \frac{612}{\alpha} \rceil$ satisfying $\gamma \geq \frac{110k}{\sqrt{\|w\|_1}}$, $\delta \in (0, 1)$

2: **Output:** Saturated $w'$, satisfying one of the following possibilities with probability $\geq 1 - \delta$:

    1. $w'$ has $\|w'\|_1 \leq \frac{1}{2} \|w\|_1$ (marked "Case 1")

    2. $\mathbf{V} \in \mathbb{R}^{d \times k}$ is also outputted, and $\left\| \mathrm{Cov}_{w'}^{\mathbf{P}_{\mathbf{V}}^{\perp}}(T) \right\|_{\mathrm{op}} \leq \frac{2}{3} \|\mathrm{Cov}_w(T)\|_{\mathrm{op}}$ (marked "Case 2")

    3. $w'$ has $\|\mathrm{Cov}_{w'}(T)\|_k \leq \frac{1}{2} \|\mathrm{Cov}_w(T)\|_k$ (marked "Case 3")

3: $N \leftarrow \lceil 425 \log d \rceil$, $w^{(0)} \leftarrow w$, $\bar{\beta} \leftarrow \|w^{(0)}\|_1$, $\eta \leftarrow \frac{1}{2.1\rho}$, where $\rho$ is a 1.05-approximation of $\left\| \widetilde{\mathrm{Cov}}_{w^{(0)}}(T) \right\|_{\mathrm{op}}$ with probability at least $1 - \frac{\delta}{3(N+1)}$

4: **for** $0 \leq t < N$ **do**

5:     $\mathbf{V} \leftarrow \mathsf{Power}(\mathrm{Cov}_{w^{(t)}}(T), k, 0.05, \frac{\delta}{3(N+1)})$

6:     $\tilde{\lambda}_1 \leftarrow \langle \mathbf{V}_{:1}, \mathrm{Cov}_{w^{(t)}}(T) \mathbf{V}_{:1} \rangle$, $\tilde{\lambda}_k \leftarrow \langle \mathbf{V}_{:k}, \mathrm{Cov}_{w^{(t)}}(T) \mathbf{V}_{:k} \rangle$

7:     **if** $\tilde{\lambda}_1 \geq 3.5 \tilde{\lambda}_k$ **then**

8:         **return** $(w^{(t)}, \mathbf{V}, \text{"Case 2"})$

9:     **end if**

10:     $\tau_i^{(t)} \leftarrow 1.05$-approximation to $\left\langle (X_i - \mu_{w^{(t)}}(T)), \widehat{\mathbf{Y}}_t (X_i - \mu_{w^{(t)}}(T)) \right\rangle$ for all $i \in T$, with probability at least $1 - \frac{\delta}{3(N+1)}$

11:     **if** $\sum_{i \in T} w_i^{(t)} \tau_i^{(t)} > \frac{\gamma \bar{\beta}}{12}$ **then**

12:         $w^{(t+1)} \leftarrow w^{(t,K)}$, where $K \leftarrow$ smallest natural number such that

$$\text{either } \left\| w^{(t,K)} \right\|_1 \leq \frac{\bar{\beta}}{2}, \text{ or } \sum_{i \in T} w_i^{(t,K)} \tau_i^{(t)} \leq \frac{\gamma \bar{\beta}}{12}, \qquad (11)$$
$$\text{where } w_i^{(t,K)} := \left( 1 - \frac{\tau_i^{(t)}}{\tau_{\max}^{(t)}} \right)^K w_i^{(t)}, \text{ and } \tau_{\max}^{(t)} := \max_{i \in T \mid w_i^{(t)} \neq 0} \tau_i^{(t)}$$

13:         **if** $\left\| w^{(t+1)} \right\|_1 \leq \frac{\bar{\beta}}{2}$ **then**

14:             **return** $(w^{(t+1)}, \text{"Case 1"})$

15:         **end if**

16:     **else**

17:         $w^{(t+1)} \leftarrow w^{(t)}$

18:     **end if**

19:     Feed $\mathbf{G}_t \leftarrow \widetilde{\mathrm{Cov}}_{w^{(t+1)}}(T)$ into the routine $\mathsf{ApproxKFMMW}$ with step size $\eta$ and $\delta' \leftarrow \frac{\delta}{3}$

20: **end for**

21: **return** $(w^{(N)}, \text{"Case 3"})$

---

12, it is with respect to safe scores, and thus $w^{(t)}$ is saturated for all $0 \leq t < N$.

*Proof.* With probability $1 - \delta$, all executions of Lines 5 and 10 throughout the algorithm succeed,

so we will condition on this event for the remainder of this proof. We also note that in any iteration $t$ where Line 12 is reached, Line 7 did not pass, and thus

$$\lambda_1\left(\mathrm{Cov}_{w^{(t)}}(T)\right) \le 1.05\tilde{\lambda}_1 < 3.675\tilde{\lambda}_k \le 4\lambda_k\left(\mathrm{Cov}_{w^{(t)}}(T)\right) \implies \|\mathrm{Cov}_{w^{(t)}}(T)\|_k \ge \frac{k}{4}\|\mathrm{Cov}_{w^{(t)}}(T)\|_{\mathrm{op}} \tag{12}$$

by the guarantees of Power in Proposition 2. Consider now a single iteration $0 \le t < N$, and suppose inductively that $w^{(t)}$ is saturated before Line 12 is executed. In every round of weight removal $0 \le \ell < K$, assuming that the $\ell_1$ norm has not halved, we can lower bound the average score in $T$ by the definition of $K$:

$$\sum_{i \in T} \frac{w_i^{(t,\ell)}}{\left\|w^{(t,\ell)}\right\|_1}\tau_i^{(t)} \ge \frac{1}{\bar{\bar{\beta}}}\sum_{i \in T} w_i^{(t,\ell)}\tau_i^{(t)} \ge \frac{\gamma}{12}.$$

Hence, to prove that the scores are safe in iteration $\ell$, it suffices to show that the average score in $S$ is at most $\frac{\gamma}{24}$. Because the weights $w^{(t,\ell)}$ are monotone in $\ell$, and the $\ell_1$ norm of $w_S^{(t,\ell)}$ inductively does not change by more than a factor of $\sqrt{2}$ by the following Lemma 7, it suffices to show that

$$\sum_{i \in S} \frac{w_i^{(t,0)}}{\left\|w_S^{(t,0)}\right\|_1}\tau_i^{(t)} \le \frac{\gamma}{34} \implies \sum_{i \in S} \frac{w_i^{(t,\ell)}}{\left\|w_S^{(t,\ell)}\right\|_1}\tau_i^{(t)} \le \frac{\gamma\sqrt{2}}{34} < \frac{\gamma}{24}.$$

We now prove this bound on the average score in $S$ with respect to $w^{(t,0)} = w^{(t)}$, which will conclude the proof. To see this bound, we have

$$\sum_{i \in S} \frac{w_i^{(t)}}{\left\|w_S^{(t)}\right\|_1}\tau_i^{(t)} \le 1.05\left\langle \widehat{\mathbf{Y}}_t, \sum_{i \in S} \frac{w_i^{(t)}}{\left\|w_S^{(t)}\right\|_1}\left(X_i - \mu_{w^{(t)}}(T)\right)\left(X_i - \mu_{w^{(t)}}(T)\right)^\top \right\rangle$$

$$= 1.05\left\langle \widehat{\mathbf{Y}}_t, \mathrm{Cov}_{w^{(t)}}(S) \right\rangle + 1.05\left\langle \widehat{\mathbf{Y}}_t, \left(\mu_{w^{(t)}}(S) - \mu_{w^{(t)}}(T)\right)\left(\mu_{w^{(t)}}(S) - \mu_{w^{(t)}}(T)\right)^\top \right\rangle$$

$$\le 1.07k\|\mathrm{Cov}_{w^{(t)}}(S)\|_{\mathrm{op}} + 1.07\left\|\left(\mu_{w^{(t)}}(S) - \mu_{w^{(t)}}(T)\right)\left(\mu_{w^{(t)}}(S) - \mu_{w^{(t)}}(T)\right)^\top\right\|_{\mathrm{op}}$$

$$\le \frac{1.07k\alpha}{\left\|w_S^{(t)}\right\|_1} + \frac{9\gamma}{k\alpha} \le \frac{1.6k}{\sqrt{\bar{\bar{\beta}}}} + \frac{9\gamma}{k\alpha} \le \frac{\gamma}{34}.$$

Here, the first inequality is by the approximation guarantees on the scores $\tau_i^{(t)}$. The second inequality used matrix Hölder twice, as well as trace and operator norm bounds on $\widehat{\mathbf{Y}}_t$ due to Proposition 3, and finally the fact that the trace and operator norm agree for any rank-1 matrix. The fourth inequality is by the helper Lemma 7 and saturation of $w^{(0)}$, and the fifth is by our choices of $k$ and lower bound on $\gamma \ge \frac{110k}{\sqrt{\bar{\bar{\beta}}}}$. The third inequality used Lemmas 3 and 4, the latter of which implies

$$\left\|\left(\mu_{w^{(t)}}(S) - \mu_{w^{(t)}}(T)\right)\left(\mu_{w^{(t)}}(S) - \mu_{w^{(t)}}(T)\right)^\top\right\|_{\mathrm{op}} \le \frac{\left\|w^{(t)}\right\|_1}{\left\|w_S^{(t)}\right\|_1}\|\mathrm{Cov}_{w^{(t)}}(T)\|_{\mathrm{op}}$$

$$\le \frac{1}{\alpha}\cdot\frac{4}{k}\|\mathrm{Cov}_{w^{(t)}}(T)\|_k \le \frac{8.4\gamma}{k\alpha}.$$

The second inequality used our assumption (12), and the last used that $\widetilde{\mathrm{Cov}}_{w^{(t)}}(T)$ is monotonically

decreasing in the Loewner order, and thus since until termination, the normalization factor $\left\|w^{(t)}\right\|_1$ does not change by more than a factor of two, and $\gamma$ is a 1.05-approximation to $\left\|\mathrm{Cov}_{w^{(0)}}(T)\right\|_k$,

$$\left\|\mathrm{Cov}_{w^{(t)}}(T)\right\|_k = \frac{1}{\left\|w^{(t)}\right\|_1}\left\|\widetilde{\mathrm{Cov}}_{w^{(t)}}(T)\right\|_k \leq \frac{2}{\bar{\beta}}\left\|\widetilde{\mathrm{Cov}}_{w^{(0)}}(T)\right\|_k \leq 2.1\gamma.$$

$\square$

In proving Lemma 7, we used the following helper lemma.

**Lemma 8.** *Consider any algorithm of the form in Lemma 1. Suppose in some iteration $t$, $\left\|w^{(t)}\right\|_1 \geq \frac{1}{2}\left\|w^{(0)}\right\|_1$. Then, $\left\|w_S^{(t)}\right\|_1 \geq \frac{1}{\sqrt{2}}\left\|w_S^{(0)}\right\|_1$.*

*Proof.* This is immediate from telescoping (10), which was used in the proof of Lemma 1. $\square$

Finally, we prove overall correctness of Algorithm 3.

**Proposition 4.** *Algorithm 3 succeeds with probability at least $1-\delta$, in the sense that each of Cases 1-3 returns correctly. The overall complexity is bounded by*

$$O\left(ndk\log^2(d)\log^2\left(\frac{dR}{\delta}\right)\right).$$

*Proof.* We will show correctness and complexity of Algorithm 3 separately.

*Correctness guarantee.* As argued in the proof of Lemma 7, with probability $1-\delta$ every weight removal is safe, so Lemma 7 shows that $w^{(t)}$ is saturated throughout the algorithm. By a union bound, we also assume that all approximations are correct in the remainder of the proof. It is obvious that if the algorithm terminates in Line 14, the requirement of Case 1 is met. If the algorithm terminates in Line 8, the guarantees of Power (Proposition 2) imply that

$$\begin{aligned}
\lambda_1\left(\mathrm{Cov}_{w^{(t)}}(T)\right) &\geq \frac{1}{1.05}\tilde{\lambda}_1 \geq \frac{3.5}{1.05}\tilde{\lambda}_k \geq \frac{3.5}{1.05^2}\lambda_k\left(\mathrm{Cov}_{w^{(t)}}(T)\right) \\
&\geq \frac{3.5}{1.05^3}\left\|(\mathbf{I}-\mathbf{V}\mathbf{V}^\top)\mathrm{Cov}_{w^{(t)}}(T)(\mathbf{I}-\mathbf{V}\mathbf{V}^\top)\right\|_{\mathrm{op}} \geq 3\left\|\mathrm{Cov}_{w^{(t)}}^{\mathbf{P}_\mathbf{V}^\perp}(T)\right\|_{\mathrm{op}}.
\end{aligned}$$
(13)

However, since the algorithm did not terminate on Line 14 in the previous iteration, we also have

$$\lambda_1\left(\mathrm{Cov}_{w^{(t)}}(T)\right) = \frac{1}{\left\|w^{(t)}\right\|_1}\lambda_1\left(\widetilde{\mathrm{Cov}}_{w^{(t)}}(T)\right) \leq \frac{2}{\bar{\beta}}\lambda_1\left(\widetilde{\mathrm{Cov}}_{w^{(0)}}(T)\right) = 2\lambda_1\left(\mathrm{Cov}_{w^{(0)}}(T)\right).$$

Combining the above two calculations gives the correctness proof for Case 2, as

$$\left\|\mathrm{Cov}_{w^{(t)}}^{\mathbf{P}_\mathbf{V}^\perp}(T)\right\|_{\mathrm{op}} \leq \frac{1}{3}\lambda_1\left(\mathrm{Cov}_{w^{(t)}}(T)\right) \leq \frac{2}{3}\lambda_1\left(\mathrm{Cov}_{w^{(0)}}(T)\right).$$

Finally, we show correctness in Case 3, where $N$ iterations of the algorithm have passed without terminating on either of Lines 8 (which halves operator norm) or 14 (which halves weight). In this case, we apply Proposition 3, which is valid since the $\mathbf{G}_t$ are monotonically decreasing, and $\eta\mathbf{G}_0 \preceq \frac{1}{2}\mathbf{I}$ by the approximation guarantee on $\rho$. Here, we also note that all our matrices $\mathbf{G}_t$ are

covariance matrices with known weights, so they can be expressed in the form $\mathbf{M}_t^\top \mathbf{M}_t$ for explicitly given $\mathbf{M}_t \in \mathbb{R}^{n \times d}$. Proposition 3 additionally requires a bound on each $\left\langle \mathbf{G}_t, \widehat{\mathbf{Y}}_t \right\rangle$; to this end,

$$
\begin{aligned}
\left\langle \mathbf{G}_t, \widehat{\mathbf{Y}}_t \right\rangle &= \sum_{i \in T} w_i^{(t+1)} \left\langle \left(X_i - \mu_{w^{(t+1)}}(T)\right), \widehat{\mathbf{Y}}_t \left(X_i - \mu_{w^{(t+1)}}(T)\right) \right\rangle \\
&\leq \sum_{i \in T} w_i^{(t+1)} \left\langle \left(X_i - \mu_{w^{(t)}}(T)\right), \widehat{\mathbf{Y}}_t \left(X_i - \mu_{w^{(t)}}(T)\right) \right\rangle \leq 1.05 \sum_{i \in T} w_i^{(t+1)} \tau_i^{(t)} \leq \frac{1.05 \gamma \bar{\beta}}{12}.
\end{aligned}
$$

In the first inequality, we used Fact 2; in the second, we used the assumption on the scores $\tau^{(t)}$; and in the third, we used the second guarantee in (11) since we did not terminate on Line 14. Now, applying this bound in every iteration $0 \leq t < N$ in Proposition 3, and defining $\mathbf{G}_N = \mathbf{G}_{N-1}$,

$$
\begin{aligned}
\|\mathbf{G}_N\|_k &\leq \frac{1.05 \gamma \bar{\beta}}{6} + \frac{2.1 k \rho \log d}{N} + \frac{2.1 k \rho}{200} \\
&\leq \frac{1.05 \gamma \bar{\beta}}{6} + \frac{2.21 k \left\|\widetilde{\mathrm{Cov}}_{w^{(0)}}(T)\right\|_{\mathrm{op}} \log d}{N} + \frac{2.21 k \left\|\widetilde{\mathrm{Cov}}_{w^{(0)}}(T)\right\|_{\mathrm{op}}}{200} \\
&\leq \frac{1.05 \gamma \bar{\beta}}{6} + \frac{9 \left\|\widetilde{\mathrm{Cov}}_{w^{(0)}}(T)\right\|_k \log d}{N} + \frac{9 \left\|\widetilde{\mathrm{Cov}}_{w^{(0)}}(T)\right\|_k}{200} \\
&\leq \frac{1.05^2 \|\mathrm{Cov}_{w^{(0)}}(T)\|_k \bar{\beta}}{6} + \frac{9 \left\|\widetilde{\mathrm{Cov}}_{w^{(0)}}(T)\right\|_k \log d}{N} + \frac{9 \left\|\widetilde{\mathrm{Cov}}_{w^{(0)}}(T)\right\|_k}{200}.
\end{aligned}
$$

The first inequality was by Proposition 3 and the definition of $\eta$; the second was by the approximation guarantee on $\rho$; the third was by the fact that the first iteration did not terminate on Line 8, so we can apply the bound (12); and the fourth was by the definition of $\gamma$. Next, dividing both sides by $\bar{\beta}$ and using that termination on Line 14 has not occurred,

$$
\begin{aligned}
\frac{1}{2} \left\|\mathrm{Cov}_{w^{(N)}}(T)\right\|_k &= \frac{1}{2 \left\|w^{(N)}\right\|_1} \left\|\widetilde{\mathrm{Cov}}_{w^{(N)}}(T)\right\|_k \\
&\leq \frac{1}{\bar{\beta}} \|\mathbf{G}_N\|_k \leq \|\mathrm{Cov}_{w^{(0)}}(T)\|_k \left(\frac{1.05^2}{6} + \frac{9 \log d}{N} + \frac{9}{200}\right).
\end{aligned}
$$

Here, we used that $\frac{1}{\bar{\beta}} \left\|\widetilde{\mathrm{Cov}}_{w^{(0)}}(T)\right\|_k = \|\mathrm{Cov}_{w^{(0)}}(T)\|_k$ twice, by definition of $\bar{\beta}$. Rearranging and using the definition of $N \geq 425 \log d$ then yields correctness of Case 3.

*Complexity guarantee.* For $N = O(\log d)$, the total cost of running ApproxKFMMW is

$$
O\left(ndk \log^2(d) \log^2\left(\frac{d}{\delta}\right)\right),
$$

as given by Proposition 3. It is straightforward to check that the costs of Lines 5 and 10, given by Propositions 2 and 3, do not dominate this. Finally, since the cost of checking (11) for a value of $K$ is linear in $n$, it suffices to provide an upper bound on $K$ and then binary search. For this, we have

$$
\sum_{i \in T} w_i^{(t,K)} \tau_i^{(t)} \leq \sum_{i \in T} \exp\left(-\frac{K \tau_i^{(t)}}{\tau_{\max}^{(t)}}\right) w_i^{(t)} \tau_i^{(t)} \leq \frac{1}{eK} \sum_{i \in T} w_i^{(t)} \tau_{\max}^{(t)} \leq \frac{\tau_{\max}^{(t)}}{eK}.
$$

Here, the first inequality used the definition of $w_i^{(t,K)}$, the second used that $x \exp(-Cx) \leq \frac{1}{eC}$ for all nonnegative $x$, where we chose $C = \frac{K}{\tau_{\max}^{(t)}}$, and the third used $w^{(t)} \in \Delta^n$. Since the definition of saturated weights implies that $\sqrt{\bar{\beta}} \geq \alpha$, it follows that the threshold in Line 11 satisfies

$$\frac{\gamma\bar{\beta}}{12} \geq \frac{110k\alpha}{12} \geq 5000.$$

Also, Assumption 2 and $\left\|\widehat{\mathbf{Y}}_t\right\|_{\mathrm{op}} \leq 1.01$ imply that all scores are bounded by $1.01R^2$, so we conclude $K \leq R^2$. Thus, the complexity of the binary search is $O(n \log R)$ and does not dominate. $\qquad\square$

## 4.2 Analysis of ProduceGoodTuple

At this point, the statements and analyses of both BicriteriaFilter and ProduceGoodTuple are straightforward, as we have done most of the heavy lifting in proving Proposition 4. We state both here and prove their correctness and a runtime guarantee in Proposition 5.

**Proposition 5.** *Algorithm 5,* ProduceGoodTuple, *correctly outputs a good tuple with probability at least* $1 - \delta$. *Its overall complexity is*

$$O\left(\frac{nd}{\alpha} \log^2(d) \log^2\left(\frac{dR}{\delta}\right) \log(R) \log\left(\frac{1}{\alpha}\right)\right).$$

*Proof.* We will show correctness and complexity of Algorithm 5 separately.

*Correctness guarantee.* We first claim that if BicriteriaFilter meets its specifications, then so does ProduceGoodTuple. This is since every time BicriteriaFilter returns in Case 1, the $\ell_1$ norm of $w$ is halved, but it can never be smaller than $\alpha^2$ since $w$ is always saturated, so Case 1 occurs $\leq 2 \log \frac{1}{\alpha}$ times. Finally, note that Case 2 of BicriteriaFilter indeed constitutes a good tuple, with $c = 128$.

It remains to prove that BicriteriaFilter meets its specifications. We first claim that the while loop of Lines 5-23 is not run more than $M$ times. To see this, whenever DecreaseKFNorm returns in Case 1, the loop immediately terminates, so it suffices to bound the number of times DecreaseKFNorm returns in Case 2 or Case 3 before exiting on Line 13. Observe that every time Case 2 occurs, the operator norm of $\mathrm{Cov}_w(T)$ is decreased by $\frac{1}{3}$, but by Assumption 2 it is bounded by $R^2$ initially, and as soon as it is smaller than 100, then the algorithm will exit on Line 13. Thus, the number of times Case 2 occurs is at most $3 \log(\frac{R^2}{100})$; similarly, Case 3 occurs at most $2 \log(\frac{R^2}{100})$ times since it halves the Ky Fan norm each time. Combining these yields the claimed bound of $M$ loops.

Thus, the failure probability of BicriteriaFilter is met; it remains to prove that in each case, it returns correctly. If the algorithm returns on Line 7, this is clear. If the algorithm returns on Line 16, note that its input $w$ has $\|w\|_1 \leq \bar{\beta}$ by monotonicity of filtering, so it must be that the output of DecreaseKFNorm has $\ell_1$ norm at most $\frac{1}{2}\bar{\beta}$ by Case 1 of DecreaseKFNorm. The only other place the algorithm can return is in Line 13. However, in this case it is clear that $\mathbf{B}$ has at most $k \cdot (3 \log(\frac{R^2}{100}) + 1) = O(\frac{\log R}{\alpha})$ columns, since every time Line 18 is executed only $k$ columns are appended, and we earlier bounded the number of times Line 18 can occur. Finally, by combining the definition of $\gamma$, the fact that we always project $T$ into the orthogonal complement of $\mathbf{BB}^\top$ in Line 9, and the fact that Proposition 2 implies that $\lambda_{k+1}(\mathrm{Cov}_w(T)) \leq (1.05)^2 \frac{\gamma}{k}$ (see e.g. the calculation (13)), we see that when $(w, \mathbf{B})$ is returned,

$$\left\|\mathrm{Cov}_w^{\mathbf{P}_{\mathbf{B}}^\perp}(T)\right\|_{\mathrm{op}} \leq \frac{(1.05)^3 \gamma}{k} \leq \frac{128}{\sqrt{\|w\|_1}}.$$

**Algorithm 4** BicriteriaFilter$(T, \delta, w)$

1: **Input:** $T \subset \mathbb{R}^d$ with $|T| = n$ satisfying Assumptions 1 and 2, $\delta \in (0,1)$, saturated $w$
2: **Output:** Saturated $w'$, satisfying one of the following possibilities with probability $\geq 1 - \delta$:

    1. $w'$ has $\|w'\|_1 \leq \frac{1}{2} \|w\|_1$ (marked "Case 1")

    2. $\mathbf{B} \in \mathbb{R}^{d \times k'}$ is also outputted, for $k' = O(\frac{\log R}{\alpha})$, and

$$\left\| \mathrm{Cov}_{w'}^{\mathbf{P}_{\mathbf{B}}^{\perp}}(T) \right\|_{\mathrm{op}} \leq \frac{128}{\sqrt{\|w'\|_1}} \text{ (marked "Case 2")}$$

3: $\mathbf{B} \leftarrow [], k \leftarrow \lceil \frac{612}{\alpha} \rceil, \bar{\beta} \leftarrow \|w\|_1$
4: $\delta' \leftarrow \frac{\delta}{M}$, for $M = 5 \log(\frac{R^2}{100})$
5: **while true do**
6:     **if** $\|w\|_1 \leq \frac{1}{2} \bar{\beta}$ **then**
7:         **return** $(w, \text{"Case 1"})$
8:     **end if**
9:     $T \leftarrow$ projection of $T$ into orthogonal complement of $\mathbf{B}\mathbf{B}^{\top}$
10:     $\gamma \leftarrow 1.05$-approximation to $\|\mathrm{Cov}_w(T)\|_k$ with probability $\geq 1 - \frac{\delta'}{3(N+1)}$, for $N = \lceil 150 \log d \rceil$
11:     **if** $\gamma < \frac{110k}{\sqrt{\|w\|_1}}$ **then**
12:         Append the columns of $\mathsf{Power}(\mathrm{Cov}_w(T), k, 0.05, \frac{\delta'}{3(N+1)})$ to $\mathbf{B}$
13:         **return** $(w, \mathbf{B}, \text{"Case 2"})$
14:     **end if**
15:     **if** $\mathsf{DecreaseKFNorm}(T, w, \gamma, \delta')$ returns "Case 1" **then**
16:         **return** $(\mathsf{DecreaseKFNorm}(T, w, \gamma, \delta'), \text{"Case 1"})$
17:     **else if** $\mathsf{DecreaseKFNorm}(T, w, \gamma, \delta')$ returns "Case 2" **then**
18:         $(w, \mathbf{V}) \leftarrow \mathsf{DecreaseKFNorm}(T, w, \gamma, \delta')$
19:         Append the columns of $\mathbf{V}$ to $\mathbf{B}$
20:     **else**
21:         $w \leftarrow \mathsf{DecreaseKFNorm}(T, w, \gamma, \delta')$
22:     **end if**
23: **end while**

---

**Algorithm 5** ProduceGoodTuple$(T, \delta)$

1: **Input:** $T \subset \mathbb{R}^d$ with $|T| = n$ satisfying Assumptions 1 and 2, $\delta \in (0,1)$
2: **Output:** Good tuple $(\mathbf{B}, w)$ (cf. Definition 3) with probability $\geq 1 - \delta$
3: $w \leftarrow \frac{1}{n} \mathbb{1}$
4: **while true do**
5:     **if** $\mathsf{BicriteriaFilter}\left(T, \frac{\delta}{2 \log \frac{1}{\alpha}}, w\right)$ returns "Case 1" **then**
6:         $w \leftarrow \mathsf{BicriteriaFilter}\left(T, \frac{\delta}{2 \log \frac{1}{\alpha}}, w\right)$
7:     **else**
8:         **return** $\mathsf{BicriteriaFilter}\left(T, \frac{\delta}{2 \log \frac{1}{\alpha}}, w\right)$
9:     **end if**
10: **end while**

In the above equation, we overload $T$ to mean the original dataset (rather than after projection in Line 9). This proves correctness of BicriteriaFilter in all cases. Finally, we remark that all parts of DecreaseKFNorm operate correctly after the projection in Line 9. The only place this may cause difficulty is in dependences on smallest eigenvalues in implementing ApproxKFMMW, because the gain matrices are not full rank. However, it is straightforward to check that the guarantees of the subroutines ApproxProject and Power as given in Section 6 will depend on the smallest eigenvalues of gain matrices restricted to $\mathrm{Span}(\mathbf{I} - \mathbf{B}\mathbf{B}^\top)$, if all operations are performed in this space.

*Complexity guarantee.* By our earlier analysis, ProduceGoodTuple incurs a multiplicative $O(\log \frac{1}{\alpha})$ overhead on the cost of BicriteriaFilter, so it suffices to understand this latter complexity. The dominant cost is clearly the (at most $M$) calls to DecreaseKFNorm, and the projection steps in Line 9. Line 9 involves orthogonalizing each of $n$ vectors against $O(k \log R)$ vectors in $d$ dimensions, so its complexity is $O(ndk \log R \cdot M)$, which does not dominate. The overall cost bound follows from combining Proposition 4 with a multiplicative $O(M \log \frac{1}{\alpha})$ overhead factor. □

By combining Lemma 6 with Proposition 5, we have the following guarantee on FastSIFT.

**Corollary 1.** *With probability $1 - \delta$, some $\hat{\mu} \in L$ outputted by Algorithm 2 satisfies*

$$\|\hat{\mu} - \mu^*\|_2^2 \leq \frac{560}{\alpha}.$$

*The overall runtime of Algorithm 2 is*

$$O\left(\frac{nd}{\alpha} \log^2(d) \log^2\left(\frac{dR}{\delta}\right) \log(R) \log\left(\frac{1}{\alpha}\right) + \frac{1}{\alpha^6} \log^3(R) \log\left(\frac{dR}{\delta}\right)\right).$$

## 5 Cleanup

In this section, we give implementations of pre-processing and post-processing procedures on the dataset which will be used in attaining our final guarantees. In particular, Section 5.1 shows how to reduce the size of our final output list, and Section 5.2 shows how to naïvely cluster the dataset to have diameter polynomially bounded in problem parameters. Finally, we put all the pieces together in giving our final result on list decodable mean estimation in Section 5.3, as well as a variant on this procedure which obtains a slight runtime-accuracy tradeoff, in Section 5.4.

### 5.1 Merging candidate means

We give a simple greedy algorithm for taking the output of Algorithm 2 (FastSIFT) and reducing its size to be $O(\frac{1}{\alpha})$, without affecting the guarantee (8) by more than a constant factor. The algorithm and analysis bear some resemblance to the strategy in [DKK20], but we include it for completeness. In this section, denote $k := \lceil \frac{4}{\alpha} \rceil$ as in Algorithm 1. We recall from the description of SIFT that the output $L$ of FastSIFT has the property that elementwise, all $\hat{\mu} \in L$ are of the form

$$\mu_{\mathrm{fixed}} + \mathbf{V}\mathbf{V}^\top \mathbf{B}\mathbf{B}^\top X_i = \mu_{\mathrm{fixed}} + \mathbf{P} X_i, \text{ where } X_i \in T_{\mathrm{slow}}, \ \mathbf{P} := \mathbf{V}\mathbf{V}^\top, \quad (14)$$

since columns of $\mathbf{V} \in \mathbb{R}^{d \times k}$ are contained in $\mathrm{Span}(\mathbf{B})$, and $\mu_{\mathrm{fixed}}$ lies in the orthogonal complement of $\mathrm{Span}(\mathbf{V})$.[6] To see this, note that all input points to SIFT are of the form $\mathbf{B}\mathbf{B}^\top X_i$ (Line 4 of FastSIFT), and because SIFT then works in the coordinate system of $\mathbf{B}$, every element of the

---

[6]In the implementation, we will have $\mathbf{V} \in \mathbb{R}^{k' \times k}$ where $k'$ is the column dimensionality of $\mathbf{B}$ since it is expressed in the coordinate system of $\mathbf{B}$, but we write it this way for consistency with the whole algorithm.

output list will have this form. In particular, $\mu_{\text{fixed}}$ is the sum of $\mu_{\text{fast}}$ (Line 3, Algorithm 2) and the empirical mean in the last iteration of SIFT projected into $(\mathbf{I} - \mathbf{P})$ (Line 12, Algorithm 1).

Because the proof of Lemma 6 (with $c = 128$, cf. Proposition 5) shows that

$$\|\mu_{\text{fixed}} - (\mathbf{I} - \mathbf{P})\mu^*\|_2^2 \leq \frac{512 + 28}{\alpha} = \frac{540}{\alpha}, \; \|\mathbf{P}(X_i - \mu^*)\|_2^2 \leq \frac{8}{\alpha} \text{ for some } \hat{\mu} \in L, \qquad (15)$$

it suffices to reduce the number of $\mathbf{P}X_i$ while maintaining one with squared $\ell_2$ distance $O(\frac{1}{\alpha})$ from $\mathbf{P}\mu^*$. We now give our post-processing procedure. In the following, define $n' := |T_{\text{slow}}| = O(\frac{\log R}{\alpha^2})$.

---

**Algorithm 6** PostProcess$(L, \alpha)$

---

1: **Input:** $L$, the output of Algorithm 2 (FastSIFT) decomposed as (14), satisfying (15)
2: **Output:** $\widetilde{L}$, a subset of $L$ with $|\widetilde{L}| \leq \frac{2}{\alpha}$
3: $\widetilde{L} \leftarrow \emptyset$
4: Let $\widetilde{L}$ be a maximal subset of $L$ of points $\hat{\mu} = \mu_{\text{fixed}} + \mathbf{P}X_i$, such that $\|\mathbf{P}(X_i - X_j)\|_2^2 \leq \frac{32}{\alpha}$ for at least $\frac{n'\alpha}{2}$ of the $X_j \in T_{\text{slow}}$, and $\|\hat{\mu} - \hat{\mu}'\|_2^2 \geq \frac{128}{\alpha}, \forall \hat{\mu}' \in \widetilde{L}$
5: **return** $\widetilde{L}$

---

**Lemma 9.** *The output of Algorithm 6 has $|\widetilde{L}| \leq \frac{2}{\alpha}$, and at least one $\hat{\mu} \in \widetilde{L}$ has*

$$\|\hat{\mu} - \mu^*\|_2^2 \leq \frac{1052}{\alpha}.$$

*The overall runtime of the algorithm is*

$$O\left(\frac{1}{\alpha^4} \log(R) \log\left(\frac{1}{\delta}\right)\right).$$

*Proof.* We first prove the bound on the list size. Note that every element $\hat{\mu} \in \widetilde{L}$ is associated with at least $\frac{n'\alpha}{2}$ elements in $T_{\text{slow}}$; call this the "cluster" of $\hat{\mu}$. By the separation assumption on pairs in $\widetilde{L}$, the clusters of all $\hat{\mu}, \hat{\mu}' \in \widetilde{L}$ are distinct, so there can only be at most $\frac{2}{\alpha}$ clusters as desired.

We now show the error guarantee. By the decomposition (14), the assumption (15), and the Pythagorean theorem, it suffices to show that for some $\hat{\mu} = \mathbf{P}X_j + \mu_{\text{fixed}}$ in the output list,

$$\|\mathbf{P}(X_j - \mu^*)\|_2^2 \leq \frac{512}{\alpha}. \qquad (16)$$

By assumption, there is a particular $\hat{\mu} = \mathbf{P}X_i + \mu_{\text{fixed}} \in L$ which satisfies the bound (15). We will designate this $\hat{\mu}$ as $\hat{\mu}_{\text{good}}$ throughout the proof, and fix the index $i$ to be associated with $\hat{\mu}_{\text{good}}$. Next, we recall that at least $\frac{n'\alpha}{2}$ of the points $X_j \in T_{\text{slow}}$ have

$$\|\mathbf{P}(X_j - \mu^*)\|_2^2 \leq \frac{8}{\alpha}.$$

This was shown in the first part of Theorem 3, and is a straightforward application of Markov and Assumption 1. By triangle inequality to $\mu^*$ and the definition of $\hat{\mu}_{\text{good}}$, $X_i$ satisfies

$$\|\mathbf{P}(X_i - X_j)\|_2^2 \leq \frac{32}{\alpha} \text{ for at least } \frac{n'\alpha}{2} \text{ of the } X_j \in T_{\text{slow}}.$$

Now, assume that (16) does not occur; this clearly also means that $\hat{\mu}_{\text{good}}$ cannot belong to $\widetilde{L}$. However, this is a contradiction, since triangle inequality implies that if no point in $\widetilde{L}$ satisfies (16), then $\hat{\mu}_{\text{good}}$ would be added to the list by maximality of the subset.

Finally, we show the complexity guarantee. Throughout, we use the assumption that $L$ has already been decomposed as (14), and all components in $\mathbf{P}$ are expressed in the coordinate system of $\mathbf{V}$, so all distance comparisons take time $O(k)$. We can first eliminate all points which do not meet the clustering criteria (e.g. do not have enough points nearby) in one pass, in time $O(|L||T_{\text{slow}}|k)$. Afterwards, a naïve greedy algorithm suffices for forming a list $\widetilde{L}$ in Line 4, e.g. iteratively looping over $L$ and performing the check against points in $|\widetilde{L}|$ sequentially until a loop adds no elements to $\widetilde{L}$. This costs $O(|L||\widetilde{L}|^2 k)$, which yields the runtime since we argued $|\widetilde{L}| = O(k)$. $\qquad\square$

## 5.2 Bounding dataset diameter

In Section 4, we developed an algorithm for list-decodable mean estimation under Assumption 2. We now demonstrate how to reduce a general dataset satisfying Assumption 1 to this case. Our strategy will be to divide the original dataset into multiple portions of bounded diameter, such that with high probability all of the points in $S$ satisfying Assumption 1 lie in the same set. To do so, we perform a random one-dimensional projection, which is likely to preserve distances up to a polynomial factor, and then use an equivalence class partition as our clustering. We state two simple facts which are helpful in the analysis.

**Lemma 10.** *No two points $X_i, X_j \in S$ have $\|X_i - X_j\|_2 \geq 2\sqrt{n}$.*

*Proof.* It suffices to show that every point in $S$ has distance at most $\sqrt{n}$ from $\mu^*$. If this were not the case, it is clear Assumption 1 cannot hold by virtue of the corresponding rank-one term. $\quad\square$

**Lemma 11.** *Let $T$ be a set of $n$ points in $\mathbb{R}^d$, and sample $g \sim \mathcal{N}(0, \mathbf{I})$. With probability at least $1 - \delta$, for every pair of distinct points $X_i, X_j \in T$,*

$$\frac{1}{4 \log \frac{n}{\delta}} \left( \langle g, X_i - X_j \rangle \right)^2 \leq \|X_i - X_j\|_2^2 \leq \frac{n^4}{\delta^2} \left( \langle g, X_i - X_j \rangle \right)^2. \tag{17}$$

*Proof.* Fix a pair $X_i, X_j \in T$; we show that each of the bounds in (17) holds with probability at least $1 - \frac{\delta}{n^2}$, and then the conclusion holds by a union bound over both tails and all pairs. Since the distribution of $\langle g, X_i - X_j \rangle$ is $\mathcal{N}(0, \|X_i - X_j\|_2^2)$, the lower bound in (17) is a straightforward application of sub-Gaussian concentration. The upper bound comes from the fact that the probability mass of $\mathcal{N}(0, 1)$ in the range $[-\sqrt{\epsilon}, \sqrt{\epsilon}]$ is bounded by

$$\frac{1}{\sqrt{2\pi}} \int_{-\sqrt{\epsilon}}^{\sqrt{\epsilon}} \exp\left(-\frac{1}{2}t^2\right) dt \leq \sqrt{\epsilon}.$$

Hence, the probability that $Z \sim \mathcal{N}(0, \|X_i - X_j\|_2^2)$ has $Z^2 \leq \frac{\delta^2}{n^4}$ is bounded by $\frac{\delta}{n^2}$. $\qquad\square$

At this point, we are ready to give our pre-processing procedure.

**Lemma 12.** PreProcess *meets its output specifications. The overall runtime is*

$$O\left(nd + n \log n\right).$$

**Algorithm 7** PreProcess($T, \delta$)

---

1: **Input:** $T \subset \mathbb{R}^d$ with $|T| = n$ satisfying Assumption 1, $\delta \in (0, 1)$
2: **Output:** Partition of $T$ into disjoint clusters $\{T_j\}_{j \in [m]}$, such that all of $S$ is contained in a single cluster, and every cluster has radius $\leq \frac{4n^4}{\delta^2}$, with probability $1 - \delta$
3: $g \sim \mathcal{N}(0, \mathbf{I})$
4: $v_i \leftarrow \langle g, X_i \rangle$ for all $X_i \in T$
5: Partition $T$ into equivalence classes $\{T_j\}_{j \in [m]}$, where indices $i, i'$ are in the same $T_j$ if there is a path of distinct $i_1 = i, i_2, \ldots i_\ell = i'$ so that each consecutive $|v_{i_a} - v_{i_{a+1}}| \leq 4\sqrt{n \log \frac{n}{\delta}}$
6: **return** Clusters in $\{T_j\}_{j \in [m]}$ with at least $\alpha n$ points

---

*Proof.* The runtime bound is immediate; Lines 3 and 4 clearly take time $O(nd)$, and Line 5 can be performed by sorting the values $\{v_i\}_{i \in T}$ and greedily forming clusters, creating disjoint paths from the smallest value to the largest. To show correctness, condition on the conclusion of Lemma 11 occuring (giving the failure probability). We begin with the claim that all of $S$ is contained in a single cluster; to see this, if $X_i, X_j \in S$, then combining Lemma 10 and Lemma 11 implies that

$$(v_i - v_j)^2 \leq \left(4 \log \frac{n}{\delta}\right)(4n) \implies |v_i - v_j| \leq 4\sqrt{n \log \frac{n}{\delta}}.$$

Furthermore, suppose two points $X_i, X_{i'}$ are in the same cluster, witnessed by a path of length $\ell \leq n$ starting at $i_1 = i$ and ending at $i_\ell = i'$. Then, by triangle inequality

$$|v_i - v_{i'}| \leq \sum_{a=1}^{\ell-1} |v_{i_a} - v_{i_{a+1}}| \leq 4\sqrt{n^3 \log \frac{n}{\delta}}$$

$$\implies (\langle g, X_i - X_{i'}\rangle)^2 \leq 16n^3 \log \frac{n}{\delta} \implies \|X_i - X_{i'}\|_2^2 \leq \frac{16n^7}{\delta^2} \log \frac{n}{\delta} \leq \frac{16n^8}{\delta^4}.$$

In the last implication, we used the upper bound in Lemma 11. $\qquad\square$

## 5.3 Putting it all together

Finally, we put together the pieces we have developed to give our final algorithm.

---

**Algorithm 8** ListDecodableMeanEstimation($T, \delta$)

---

1: **Input:** $T \subset \mathbb{R}^d$ with $|T| = n$ satisfying Assumption 1, $T_{\text{slow}} \subset \mathbb{R}^d$ with $|T_{\text{slow}}| = O(\frac{\log d/\delta}{\alpha^2})$ satisfying Assumption 1 for $\frac{1}{\alpha}$ fixed $O(\frac{\log d/\delta}{\alpha})$-dimensional subspaces (cf. Remark 1, where we use $R = \text{poly}(d, \delta^{-1})$ as below, where $n = \text{poly}(d)$), $\delta \in (0, 1)$
2: **Output:** $L \subset \mathbb{R}^d$ with $|L| \leq \frac{2}{\alpha}$ satisfying (8) with probability $\geq 1 - \delta$
3: $\{T_j\}_{j \in [m]} \leftarrow \text{PreProcess}(T, \frac{\delta}{2})$
4: $L_j \leftarrow \text{FastSIFT}(T_j, \frac{\delta\alpha}{2}, \text{ProduceGoodTuple})$, for all $j \in [m]$, with $R = \frac{4n^4}{\delta^2}$, and $\alpha_j = \frac{\alpha|T|}{|T_j|}$, reusing the same datapoints $T_{\text{slow}}$ for each call to FastSIFT
5: $L_j \leftarrow \text{PostProcess}(L_j, \alpha_j)$, for all $j \in [m]$
6: **return** $L \leftarrow \bigcup_{j \in P} L_j$, where $P = \{j \in [m] \mid |L_j| \leq \frac{2}{\alpha_j}\}$

---

**Theorem 4.** *Under Assumption 1, with probability at least $1 - \delta$, ListDecodableMeanEstimation outputs a list of size at most $\frac{2}{\alpha}$, and attains error*

$$\min_{\mu \in L} \|\mu - \mu^*\|_2 = O\left(\frac{1}{\sqrt{\alpha}}\right).$$

*The overall runtime is*

$$O\left(\frac{nd}{\alpha} \log^2(d) \log^3\left(\frac{d}{\delta}\right) \log\left(\frac{1}{\alpha}\right) + \frac{1}{\alpha^6} \log^4\left(\frac{d}{\delta}\right)\right).$$

*Proof.* We will show correctness and complexity of Algorithm 8 separately.

*Correctness guarantee.* First, note there are at most $\alpha^{-1}$ clusters outputted by PreProcess, so by a union bound, with probability at least $1 - \delta$, both PreProcess and all FastSIFT calls succeed. Note that whichever cluster $T_j$ that contains all of $S$ indeed satisfies Assumption 1, with $|S| = \alpha_j |T_j|$, by definition of $\alpha_j$. Thus, Corollary 1 and Lemma 9 imply that index $j$ will belong to the output set $P$, and an element of $L_j$ will meet the error guarantee (8). The list size follows from

$$|L| \leq \sum_{j \in [m]} \frac{2}{\alpha_j} = \frac{2}{\alpha}.$$

Finally, we remark that we can reuse the same slow dataset $T_{\text{slow}}$ for each of the at most $\frac{1}{\alpha}$ runs of FastSIFT in Line 4, corresponding to different clusters, up to a $\frac{1}{\alpha}$ factor in the failure probability of Proposition 1. This is because (as in Remark 1), the low-dimensional subspaces produced by ProduceGoodTuple are each independent of any randomness used in generating the set $T_{\text{slow}}$.

*Complexity guarantee.* The cost of PostProcess given in Lemma 9 never dominates the cost of FastSIFT given in Corollary 1; similarly, it is clear that the cost of PreProcess given in Lemma 12 never dominates. Thus, it suffices to bound the costs of all calls to FastSIFT in Line 4. To this end, we bound contributions of the two terms in the runtime of Corollary 1. Because each $\alpha_j \geq \alpha$ and the sum of the sizes of the $\{T_j\}_{j \in [m]}$ is $n$,

$$\sum_{j \in [m]} \frac{|T_j| d}{\alpha_j} \leq \frac{|T| d}{\alpha} = \frac{nd}{\alpha}.$$

Similarly, denoting $k_j = \frac{1}{\alpha_j}$ and $k = \frac{1}{\alpha}$, since $\sum_{j \in [m]} k_j = k$ by design,

$$\sum_{j \in [m]} k_j^6 \leq \left(\sum_{j \in [m]} k_j\right)^6 = \frac{1}{\alpha^6}.$$

$\square$

## 5.4 Trading off accuracy for runtime

In this section, we give a simple alternative to the algorithm ListDecodableMeanEstimation which removes the lower-order term in the runtime (so that the complexity is just the cost of polylogarithmically many calls to a $k$-PCA routine), at the cost of a slight loss in the accuracy term. We first note that unless $\alpha^{-1} = \omega\left(\sqrt{d}\right)$, the term with dependence $\alpha^{-6}$ will not dominate the complexity

of Theorem 4. This is because we choose our sample complexity (following Assumption 1) to be on the order of $\frac{d}{\alpha}$, so that asymptotically,

$$\frac{1}{\alpha^6} > \frac{nd}{\alpha} \implies d^2 < \frac{1}{\alpha^4}.$$

We now give the main result of this section, which shows in this regime of $\alpha^{-1}$, it suffices to randomly sample in the last stage of each run of FastSIFT rather than apply SIFT. The following Algorithm 9 (FasterSIFT) is a simple modification of FastSIFT, which is the same for the first three lines, as well as the last. The only difference is that in Line 4, the list $L_{\text{slow}}$ is formed by random sampling points from $T_{\text{slow}}$ and projecting into the subspace $\mathbf{BB}^\top$.

---

**Algorithm 9** FasterSIFT$(T, \delta, \text{ProduceGoodTuple})$

---

1: **Input:** $T = T_{\text{fast}} \cup T_{\text{slow}} \subset \mathbb{R}^d$ with $|T_{\text{fast}}| = n$ satisfying Assumptions 1 and 2, $|T_{\text{slow}}| = O(\frac{\log R}{\alpha^2})$ satisfying Assumption 1 for a fixed $O(\frac{\log R}{\alpha})$-dimensional subspace, $\delta \in (0,1)$, subroutine ProduceGoodTuple which returns a good tuple with specified failure probability
2: $(\mathbf{B}, w) \leftarrow \text{ProduceGoodTuple}(T_{\text{fast}}, \frac{\delta}{2})$
3: $\mu_{\text{fast}} \leftarrow (\mathbf{I} - \mathbf{BB}^\top)\mu_w(T)$
4: $L_{\text{slow}} \leftarrow \{\mathbf{BB}^\top X_i \text{ where } i \in T_{\text{slow}} \text{ is sampled uniformly at random}\}$, with list size $|L_{\text{slow}}| = \lceil \frac{2}{\alpha} \log \frac{4}{\delta\alpha} \rceil$
5: **return** $L \leftarrow \{\mu_{\text{slow}} + \mu_{\text{fast}} \mid \mu_{\text{slow}} \in L_{\text{slow}}\}$

---

**Corollary 2.** *Consider running* ListDecodableMeanEstimation *with a modification: in Line 4, use* FasterSIFT *(Algorithm 9) in place of* FastSIFT *(Algorithm 2). The resulting list has size at most $\frac{2}{\alpha}$. Under Assumption 1, with probability at least $1 - \delta$, the overall runtime is*

$$O\left(\frac{nd}{\alpha} \log^2(d) \log^3\left(\frac{d}{\delta}\right) \log\left(\frac{1}{\alpha}\right)\right),$$

*and the error guarantee is*

$$\min_{\mu \in L} \|\mu - \mu^*\|_2 = O\left(\sqrt{\frac{\log \frac{1}{\delta\alpha}}{\alpha}}\right).$$

*Proof.* We first discuss list size and error guarantee. It suffices to show that for the cluster $T_j$ containing all of $S$, we can modify Lemma 9 to obtain a list size $\frac{2}{\alpha_j}$ and error guarantee on the order of $\sqrt{\log(1/\delta\alpha)/\alpha}$. To see this, all arguments in Lemma 9 follow identically, except that the random sampling occured in a $O(\frac{\log R}{\alpha_j})$-dimensional space. Hence, the error guarantee is correspondingly amplified, where we recall $R = \text{poly}(d, \delta^{-1})$, but the list size argument is the same (e.g. we only keep means which contain at least $O(|T_j|\alpha_j)$ points within their cluster, and all clusters are disjoint).

We now discuss runtime. The cost of all runs of FastSIFT remains the same, up until the step where SIFT is run; clearly, the cost of random sampling is cheaper than running ProduceGoodTuple, once the projections into the coordinate system of $\mathbf{B}$ have already been formed. Finally, the only place that we can lose runtime due to working in a larger-dimensional subspace is in the complexity of PostProcess, where operations are done in $O(\frac{\log R}{\alpha})$ dimensions. Mirroring the proof of Lemma 9, this only adds a $\log R$ overhead, and it is straightforward to check that the cost of all runs of PostProcess do not dominate, since for $d \geq \alpha^{-1}$ and our choice of $n$, $\frac{nd}{\alpha} \geq \frac{1}{\alpha^4}$. □

## 6 Ky Fan matrix multiplicative weights

We give a regret guarantee for a Ky Fan matrix multiplicative weights procedure, as well as its efficient implementation. We first state a general-purpose regret bound in Section 6.1, using a key divergence bound shown in Section 6.2. We then show how to use a more fine-grained analysis of simultaneous power iteration developed in Section 6.3 to prove correctness and a complexity bound on our overall method (tolerant to approximation error), given in Section 6.4.

Throughout this entire section, all variables (unless otherwise specified) will be either $d$-dimensional vectors or $d \times d$ matrices, and we let $k \in [d]$ be some smaller dimensionality.

### 6.1 Regret bound

Throughout this section, we define a "dual set" and regularizer inducing dual variables as follows:

$$\mathcal{Y} := \{\mathbf{Y} \mid \mathbf{0} \preceq \mathbf{Y} \preceq \mathbf{I}, \ \mathrm{Tr}(\mathbf{Y}) = k\}, \ r(\mathbf{Y}) := \langle \mathbf{Y}, \log \mathbf{Y} \rangle - \mathrm{Tr}(\mathbf{Y}). \tag{18}$$

Finally, we define the projection operator for any symmetric matrix $\mathbf{S}$,

$$\nabla r^*(\mathbf{S}) := \mathrm{argmin}_{\mathbf{Y} \in \mathcal{Y}} \{\langle -\mathbf{S}, \mathbf{Y} \rangle + r(\mathbf{Y})\}, \ \text{where} \ r^*(\mathbf{S}) := \max_{\mathbf{Y} \in \mathcal{Y}} \{\langle \mathbf{S}, \mathbf{Y} \rangle - r(\mathbf{Y})\}. \tag{19}$$

Here, we remark that it is a direct application of convex duality and the following fact (which is standard, and follows from e.g. the arguments of [Yu13]) that $\nabla r^*$ is unique, and is the gradient of $r^*$, the Fenchel dual of $r$ over the set $\mathcal{Y}$.

**Fact 3.** *Function $r$ defined in* (18) *is $\frac{1}{k}$-strongly convex over $\mathcal{Y}$ in $\|\cdot\|_{\mathrm{tr}}$, and has range $k \log \frac{d}{k}$.*

We prove a helper lemma about the structure of $\nabla r^*$, using its closed form derived in [CMY20].

**Fact 4** ([CMY20], Lemma 7.3)**.** *Given symmetric matrix $\mathbf{S}$ with eigenvalues $\lambda_1 \geq \lambda_2 \geq \ldots \geq \lambda_d$ and corresponding eigenvectors $\{v_j\}_{j \in [d]}$, we can compute $\nabla r^*(\mathbf{S})$ as follows. Define*

$$\tau(\mathbf{S}) := \max \left\{ \tau \ \middle| \ \tau > 0, \ \frac{\exp(\tau)}{\sum_{j \in [d]} \exp(\min(\tau, \lambda_j))} \leq \frac{1}{k} \right\}. \tag{20}$$

*Then,*

$$\nabla r^*(\mathbf{S}) = \sum_{j \in [d]} \frac{k \exp(\min(\tau(\mathbf{S}), \lambda_j))}{\sum_{j' \in [d]} \exp(\min(\tau(\mathbf{S}), \lambda_{j'}))} v_j v_j^\top.$$

---

**Algorithm 10** KFMMW$(k, \{\mathbf{G}_t\}_{t \geq 0}, \eta)$

---

1: **Input:** Gain matrices $\{\mathbf{G}_t\}_{t \geq 0}$, step size $\eta > 0$
2: $\mathbf{Y}_0 \leftarrow \frac{k}{d}\mathbf{I}$, $\mathbf{S}_0 \leftarrow \nabla r(\mathbf{Y}_0) = \log(\frac{k}{d})\mathbf{I}$
3: **for** $t \geq 0$ **do**
4:     $\mathbf{S}_{t+1} \leftarrow \mathbf{S}_t + \eta \mathbf{G}_t$
5:     $\mathbf{Y}_{t+1} \leftarrow \nabla r^*(\mathbf{S}_{t+1})$
6: **end for**

---

In other words, $\nabla r^*$ exponentiates its argument and normalizes the trace to be $k$, with the exception of "large" coordinates which are truncated so that the resulting matrix is operator norm bounded (as in the definition of $\mathcal{Y}$). We now give a "refined regret bound" for Algorithm 10 when all

gain matrices $\{\mathbf{G}_t\}_{t \geq 0}$ are positive and bounded. The bound is refined in the sense that it depends directly on the inner products $\langle \mathbf{G}_t, \mathbf{Y}_t \rangle$ rather than a looser, more standard bound such as $k \|\mathbf{G}_t\|_{\mathrm{op}}$ (cf. discussion in [ZLO15]). In proving Proposition 6, we will rely on a new bound on Bregman divergences with respect to $r^*$, which is stated here, and proven in the following Section 6.2.

**Lemma 13.** *For symmetric matrix* $\mathbf{S}$, *positive semidefinite* $\mathbf{G}$, *and scalar* $\eta > 0$ *let* $\mathbf{S}' = \mathbf{S} + \eta \mathbf{G}$. *Suppose that* $\|\eta \mathbf{G}\|_{\mathrm{op}} \leq \frac{1}{2}$. *Then,*

$$V_{\mathbf{S}}^{r^*}\left(\mathbf{S}'\right) \leq \langle \eta \mathbf{G}, \nabla r^*(\mathbf{S})\rangle.$$

**Proposition 6.** *Suppose the input gain matrices to Algorithm 10 satisfy the bound, for all* $t \geq 0$,

$$\mathbf{0} \preceq \eta \mathbf{G}_t \preceq \frac{1}{2}\mathbf{I}.$$

*Then, we have the guarantee for all* $T \geq 1$, *and all* $\mathbf{U} \in \mathcal{Y}$,

$$\frac{1}{T}\sum_{t=0}^{T-1}\langle \mathbf{G}_t, \mathbf{U}\rangle \leq \frac{2}{T}\sum_{t=0}^{T-1}\langle \mathbf{G}_t, \mathbf{Y}_t\rangle + \frac{k \log d}{\eta T}.$$

*Proof.* Fix some $\mathbf{U} \in \mathcal{Y}$ throughout this proof, and note that by Fact 3, $V_{\mathbf{Y}_0}^{r}(\mathbf{U}) \leq k \log d$ as $\mathbf{Y}_0$ minimizes $r$. Moreover, fix $\boldsymbol{\Psi} := \nabla r(\mathbf{U})$; it is a straightforward computation that the inverse mapping $\nabla r^*(\boldsymbol{\Psi}) = \mathbf{U}$ holds, via Fact 4. For each iteration $t$,

$$\begin{aligned}
\langle \eta \mathbf{G}_t, \mathbf{U} - \mathbf{Y}_t\rangle &= \langle \mathbf{S}_{t+1} - \mathbf{S}_t, \nabla r^*(\boldsymbol{\Psi}) - \nabla r^*(\mathbf{S}_t)\rangle \\
&= V_{\boldsymbol{\Psi}}^{r^*}(\mathbf{S}_t) - V_{\boldsymbol{\Psi}}^{r^*}(\mathbf{S}_{t+1}) + V_{\mathbf{S}_t}^{r^*}(\mathbf{S}_{t+1}) \leq V_{\boldsymbol{\Psi}}^{r^*}(\mathbf{S}_t) - V_{\boldsymbol{\Psi}}^{r^*}(\mathbf{S}_{t+1}) + \langle \eta \mathbf{G}_t, \mathbf{Y}_t\rangle.
\end{aligned} \tag{21}$$

The second equality is the well-known three-point equality of Bregman divergence and follows from expanding definitions, and in the last inequality we used Lemma 13. Telescoping (21) across all iterations and dividing by $\eta T$, we arrive at the bound

$$\frac{1}{T}\sum_{t=0}^{T-1}\langle \mathbf{G}_t, \mathbf{U} - \mathbf{Y}_t\rangle \leq \frac{1}{T}\sum_{t=0}^{T-1}\langle \mathbf{G}_t, \mathbf{Y}_t\rangle + \frac{V_{\boldsymbol{\Psi}}^{r^*}(\mathbf{S}_0)}{\eta T}.$$

The conclusion follows by rearrangement and using that (from Fact 3 and $\nabla r(\mathbf{Y}_0) = \mathbf{S}_0$)

$$\begin{aligned}
V_{\boldsymbol{\Psi}}^{r^*}(\mathbf{S}_0) &= r^*(\mathbf{S}_0) - r^*(\boldsymbol{\Psi}) - \langle \mathbf{U}, \mathbf{S}_0 - \boldsymbol{\Psi}\rangle \\
&= (\langle \mathbf{Y}_0, \mathbf{S}_0\rangle - r(\mathbf{Y}_0)) - (\langle \mathbf{U}, \boldsymbol{\Psi}\rangle - r(\mathbf{U})) - \langle \mathbf{U}, \mathbf{S}_0 - \boldsymbol{\Psi}\rangle \\
&= r(\mathbf{U}) - r(\mathbf{Y}_0) - \langle \nabla r(\mathbf{Y}_0), \mathbf{U} - \mathbf{Y}_0\rangle = V_{\mathbf{Y}_0}^{r}(\mathbf{U}) \leq k \log d.
\end{aligned}$$

$\square$

In Section 6.4, where we will only have approximate access to the $\{\mathbf{Y}_t\}_{t \geq 0}$, we give a simple bound showing that the guarantee in Proposition 6 does not significantly deteriorate as Corollary 4.

## 6.2 Refined divergence bound

In this section, we prove Lemma 13. The proof is patterned from calculations in [CDST19, JLL$^+$20] tailored towards the specific properties of the functions $r$, $r^*$ in (18), (19). We define the vector

variants of these functions, denoted $r_{\mathrm{vec}} : \mathcal{Y}_{\mathrm{vec}} \to \mathbb{R}$ and $r_{\mathrm{vec}}^* : \mathbb{R}^d \to \mathbb{R}$, by

$$r_{\mathrm{vec}}(y) := \langle y, \log y \rangle - \|y\|_1, \; r_{\mathrm{vec}}^*(s) := \min_{y \in \mathcal{Y}_{\mathrm{vec}}} \left\{ \langle -s, y \rangle + r(y) \right\},$$

where $\mathcal{Y}_{\mathrm{vec}}$ is the set of nonnegative vectors with $\ell_1$ norm $k$ and maximum entry bounded by 1. Here, we use $\log y$ to denote the entrywise logarithm of a vector.

**Lemma 14.** *For $s \in \mathbb{R}^d$, overload $\tau(s)$ to mean* (20) *applied to a matrix whose eigenvalues are given by $s$. Then, $r_{\mathrm{vec}}^*$ is twice-differentiable at $s$ if and only if no coordinate of $s$ is equal to $\tau(s)$.*

*Proof.* Suppose without loss throughout this proof that $s$ is sorted so $s_1 \geq \ldots \geq s_d$; we may do this since $r_{\mathrm{vec}}^*$ is symmetric in its arguments. Also, define (overloading (20) appropriately for vectors)

$$N(s) := \sum_{j \in [d]} \exp(\min(\tau(s), s_j)) \implies [\nabla r_{\mathrm{vec}}^*(s)]_j = \frac{k \exp(\min(\tau(s), s_j))}{N(s)}. \tag{22}$$

This implication is via a direct modification of the calculations leading to Fact 4 (alternatively, this follows from Corollary 3.3 of [Lew96] since $r^*$ is a spectral function).

*Twice-differentiable case.* We first prove that $r_{\mathrm{vec}}^*(s)$ is twice-differentiable when no coordinate of $s$ is $\tau(s)$; suppose that for some $0 \leq \ell \leq k - 1$, exactly $\ell$ coordinates of $s$ are (strictly) larger than $\tau(s)$.[7] If $\ell = 0$, it is clear that $r_{\mathrm{vec}}^*$ is twice-differentiable, so we focus on the case $\ell \neq 0$; in this case, by the definition of $\tau(s)$ (summing over indices larger and smaller than $\tau$ separately),

$$N(s) = k \exp(\tau(s)) = \ell \exp(\tau(s)) + \sum_{j \notin [\ell]} \exp(s_j) \implies \exp(\tau(s)) = \frac{\sum_{j \notin [\ell]} \exp(s_j)}{k - \ell}.$$

We thus compute

$$\frac{\partial}{\partial s_j} \exp(\tau(s)) = \begin{cases} 0 & j \in [\ell] \\ \frac{\exp(s_j)}{k - \ell} & j \notin [\ell] \end{cases}, \; \frac{\partial}{\partial s_j} N(s) = \begin{cases} 0 & j \in [\ell] \\ \frac{k \exp(s_j)}{k - \ell} & j \notin [\ell] \end{cases}. \tag{23}$$

It is then a straightforward calculation that $\nabla_{ij}^2 r_{\mathrm{vec}}^*(s)$ exists in all cases, upon differentiating coordinates of $\nabla r_{\mathrm{vec}}^*$ as computed in (22). In particular,

$$\nabla_{ij}^2 r_{\mathrm{vec}}^*(s) = \begin{cases} \frac{k \exp(s_i)}{N(s)} - \frac{k \exp(s_i)^2}{N(s)^2} & i = j \notin [\ell] \\ -\frac{k \exp(s_i) \exp(s_j)}{N(s)^2} & i, j \notin [\ell], i \neq j \\ 0 & \text{otherwise} \end{cases}. \tag{24}$$

This also shows that all $\nabla_{ij}^2 r_{\mathrm{vec}}^*$ are continuous in a small neighborhood of $s$, so we conclude $r_{\mathrm{vec}}^*$ is twice-differentiable at $s$.

*Non-twice-differentiable case.* Next, suppose we are in the case where some coordinate $s_\ell = \tau(s)$. We claim that $\frac{\partial}{\partial s_\ell} \frac{\partial}{\partial s_\ell} r_{\mathrm{vec}}^*(s)$ does not exist. In particular, perturbing $s_\ell$ in a positive direction does not affect $\tau(s)$, and thus does not affect $N(s)$ either, so the derivative from above of $\frac{\partial}{\partial s_\ell} r_{\mathrm{vec}}^*(s)$ with respect to $s_\ell$ vanishes. To compute the derivative from below, suppose without loss of generality that $s_\ell \geq \tau(s)$ but $s_{\ell+1} < \tau(s)$. We handle the case where $\ell \geq 2$ here, and discuss $\ell = 1$ at the end.

---

[7]From the definition of $\tau$, we cannot have $\ell \geq k$ since otherwise the sum of the $k$ largest elements is too large.

We first compute the effect on negatively perturbing $s_\ell$ on $\tau(s)$; for vanishing $\delta > 0$, let $s' = s - \delta e_\ell$. Since $\tau$ is weakly monotone in its argument, clearly $s_j \geq \tau(s) > s'_\ell$ for $j \in [\ell - 1]$, so since

$$k \exp(s'_\ell) \leq \ell \exp(s'_\ell) + (k - \ell) \exp(\tau(s)) = \ell \exp(s'_\ell) + \sum_{j \notin [\ell]} \exp(s_j) = \sum_{j \in [d]} \exp\left(\min(s'_\ell, s'_j)\right),$$

we have by the definition (20) that $\tau(s') \geq s'_\ell$. Next, by

$$k \exp(\tau(s')) = (\ell - 1) \exp(\tau(s')) + \exp\left(s'_\ell\right) + \sum_{j \notin [\ell]} \exp(s_j)$$

$$\implies \exp(\tau(s')) = \frac{\exp\left(s'_\ell\right) + \sum_{j \notin [\ell]} \exp(s_j)}{k - (\ell - 1)} = \frac{\exp\left(s'_\ell\right) + (k - \ell) \exp(\tau(s))}{k - (\ell - 1)},$$

we see that $\tau(s') < \tau(s)$ since $s'_\ell$ decreased. It is straightforward to see from this that since

$$\left[\frac{\partial}{\partial s_\ell}\right]_- \exp(\tau(s)) = \frac{\exp(s_\ell)}{k - (\ell - 1)} \implies \left[\frac{\partial}{\partial s_\ell}\right]_- \sum_{j \in [d]} \exp\left(\min(\tau(s), s_j)\right) = \frac{k \exp(s_\ell)}{k - (\ell - 1)},$$

where $[\frac{\partial}{\partial s_\ell}]_-$ is the derivative from below, we have

$$\left[\frac{\partial}{\partial s_\ell}\right]_- \frac{k \exp(s_\ell)}{\sum_{j \in [d]} \exp\left(\min(\tau(s), s_j)\right)}$$

$$= \frac{k}{\left(\sum_{j \in [d]} \exp\left(\min(\tau(s), s_j)\right)\right)^2} \left(\exp(s_\ell) \sum_{j \in [d]} \exp\left(\min(\tau(s), s_j)\right) - \frac{k \exp(s_\ell)^2}{k - (\ell - 1)}\right) \neq 0.$$

The last inequality is by

$$\sum_{j \in [d]} \exp\left(\min(\tau(s), s_j)\right) = k \exp(\tau(s)) \neq \frac{k}{k - (\ell - 1)} \exp(\tau(s)) = \frac{k}{k - (\ell - 1)} \exp(s_\ell).$$

Thus, the derivatives from above and below do not agree as desired. Finally, consider when $\ell = 1$; the above calculations imply that $\tau(s') = \infty$ (since then no element needs to be truncated). Hence,

$$\left[\frac{\partial}{\partial s_\ell}\right]_- \frac{k \exp(s_\ell)}{\sum_{j \in [d]} \exp\left(\min(\tau(s), s_j)\right)} = \frac{k}{\left(\sum_{j \in [d]} \exp\left(s_j\right)\right)^2} \left(\exp(s_\ell) \sum_{j \in [d]} \exp\left(s_j\right) - \exp(s_\ell)^2\right) \neq 0.$$

$\square$

We next prove a bound on quadratic forms with respect to the (matrix) Hessian of $r^*$, at symmetric matrices $\mathbf{S}$ where the function is twice-differentiable. We crucially use formulas for the derivatives of spectral functions (permutation-invariant scalar-valued functions on symmetric matrices which depend only on the eigenvalues), from [Lew96, LS01].

**Lemma 15.** *Let $\mathbf{S} = \mathbf{U}^\top \mathbf{diag}\,(s)\,\mathbf{U}$ be a symmetric matrix with eigenvalues $s$ sorted so that $s_1 \geq \ldots \geq s_d$, and $\mathbf{U}$ is an orthonormal basis. Then, $r^*$ is twice-differentiable at $\mathbf{S}$ if and only if no coordinate of $s$ equals $\tau(\mathbf{S})$. Further, when $r^*$ is twice-differentiable at $\mathbf{S}$, for any positive*

*semidefinite* $\mathbf{G}$,

$$\nabla^2 r^*(\mathbf{S})[\mathbf{G}, \mathbf{G}] \leq \langle \nabla r^*(\mathbf{S}), \mathbf{G}^2 \rangle.$$

*Proof.* The first claim is a direct consequence of Lemma 14 and the first part of Theorem 3.3 of [LS01], which states that when $r^*$ is a spectral function of $\mathbf{S}$, it is twice-differentiable at $\mathbf{S}$ if and only if $r^*_{\text{vec}}$ is twice-differentiable at $s$. Moreover, Theorem 3.3 of [LS01] gives the formula

$$\nabla^2 r^*(\mathbf{S})[\mathbf{G}, \mathbf{G}] = \nabla^2 r^*_{\text{vec}}(s) \left[ \text{diagvec}(\widetilde{\mathbf{G}}), \text{diagvec}(\widetilde{\mathbf{G}}) \right] + \left\langle \mathcal{A}, \widetilde{\mathbf{G}} \circ \widetilde{\mathbf{G}} \right\rangle,$$

$$\text{where } \widetilde{\mathbf{G}} = \mathbf{U}\mathbf{G}\mathbf{U}^\top, \; \mathcal{A}_{ij} = \begin{cases} 0 & i = j \\ \frac{\nabla_i r^*_{\text{vec}}(s) - \nabla_j r^*_{\text{vec}}(s)}{s_i - s_j} & i \neq j \end{cases}, \tag{25}$$

$\circ$ is the Hadamard (entrywise) product, and $\text{diagvec} : \mathbb{R}^{d \times d} \to \mathbb{R}^d$ returns the vector whose entries are the diagonal of the input matrix. Here, we assume that no two entries of $s$ are identical since it is clear that the scalar-valued Hessian is continuous at $s$ by the formula (24), so Theorem 4.2 of [LS01] shows that $\nabla^2 r^*$ is also continuous at $\mathbf{S}$ (thus we can perturb $\mathbf{S}$ infinitesimally so the eigenvalues are unique). Now, let $\tilde{s} = \min(\tau(s), s)$ entrywise. We first have

$$\nabla^2 r^*_{\text{vec}}(s) \left[ \text{diagvec}(\widetilde{\mathbf{G}}), \text{diagvec}(\widetilde{\mathbf{G}}) \right] \leq \mathbf{diag} \left( \left\{ \frac{k \exp(s_i)}{N(s)} \right\}_{s_i \leq \tau(s)} \right) \left[ \text{diagvec}(\widetilde{\mathbf{G}}), \text{diagvec}(\widetilde{\mathbf{G}}) \right]$$

$$\leq \frac{k}{N(s)} \sum_{i \in [d]} \exp(\tilde{s}_i) \left( \widetilde{\mathbf{G}}_{ii} \right)^2. \tag{26}$$

Here, we used that $\nabla^2 r^*_{\text{vec}}(s)$ is a diagonal matrix minus a rank-one term, restricted to eigenvalues which are at most $\tau(s)$ as calculated in (24). Next, we claim that for any tuple $i \neq j \in [d]$,

$$\frac{\exp(\tilde{s}_i) - \exp(\tilde{s}_j)}{s_i - s_j} \leq \frac{\exp(\tilde{s}_i) + \exp(\tilde{s}_j)}{2}.$$

Without loss of generality assume $s_i > s_j$. This claim is obvious for any tuple where $s_i > s_j \geq \tau(s)$. For all other cases, we recall the identity $\frac{\exp(a) - \exp(b)}{a - b} \leq \frac{\exp(a) + \exp(b)}{2}$ for all $a \neq b$ (cf. Lemma B.3, [JLL$^+$20]). Then, if $s_j \leq s_i < \tau(s)$, a direct application of this identity yields the claim; for the final case where $s_j < \tau(s) \leq s_i$, this follows from also using $s_i - s_j \geq \tilde{s}_i - \tilde{s}_j$. Continuing,

$$\left\langle \mathcal{A}, \widetilde{\mathbf{G}} \circ \widetilde{\mathbf{G}} \right\rangle = \frac{k}{N(S)} \sum_{i \neq j \in [d]} \frac{\exp(\tilde{s}_i) - \exp(\tilde{s}_j)}{s_i - s_j} \left( \widetilde{\mathbf{G}}_{ij} \right)^2$$

$$\leq \frac{k}{N(S)} \sum_{i \neq j \in [d]} \frac{\exp(\tilde{s}_i) + \exp(\tilde{s}_j)}{2} \left( \widetilde{\mathbf{G}}_{ij} \right)^2. \tag{27}$$

Combining (26) and (27) in the formula (25),

$$\nabla^2 r^*(\mathbf{S})[\mathbf{G},\mathbf{G}] \leq \frac{k}{N(S)} \sum_{i,j \in [d]} \frac{\exp(\tilde{s}_i) + \exp(\tilde{s}_j)}{2} \left(\widetilde{\mathbf{G}}_{ij}\right)^2$$

$$= \frac{k}{N(S)} \sum_{i,j \in [d]} \exp(\tilde{s}_i) \left(\widetilde{\mathbf{G}}_{ij}\right)^2 = \frac{k}{N(S)} \sum_{i \in [d]} \exp(\tilde{s}_i) \left(\sum_{j \in [d]} \left(\widetilde{\mathbf{G}}_{ij}\right)^2\right)$$

$$= \sum_{i \in [d]} \frac{k \exp(\tilde{s}_i)}{N(S)} \left[\widetilde{\mathbf{G}}^2\right]_{ii} = \left\langle \mathbf{diag}\left(\nabla r^*_{\text{vec}}(s)\right), \widetilde{\mathbf{G}}^2 \right\rangle.$$

Finally, note that $\widetilde{\mathbf{G}}^2 = \mathbf{U}\mathbf{G}^2\mathbf{U}^\top$, so the last expression is equal to $\left\langle \mathbf{U}^\top \mathbf{diag}\left(\nabla r^*_{\text{vec}}(s)\right)\mathbf{U}, \mathbf{G}^2 \right\rangle$ by the cyclic property of trace. We conclude by the fact that $\nabla r^*(\mathbf{S}) = \mathbf{U}^\top \mathbf{diag}\left(\nabla r^*_{\text{vec}}(s)\right)\mathbf{U}$, since $r^*$ is a spectral function, due to Corollary 3.3 of [Lew96]. $\qquad\square$

We conclude with the desired proof of Lemma 13.

**Lemma 13.** *For symmetric matrix* $\mathbf{S}$, *positive semidefinite* $\mathbf{G}$, *and scalar* $\eta > 0$ *let* $\mathbf{S}' = \mathbf{S} + \eta\mathbf{G}$. *Suppose that* $\|\eta\mathbf{G}\|_{\text{op}} \leq \frac{1}{2}$. *Then,*

$$V_{\mathbf{S}}^{r^*}\left(\mathbf{S}'\right) \leq \langle \eta\mathbf{G}, \nabla r^*(\mathbf{S})\rangle.$$

*Proof.* We first claim that without loss of generality, everywhere on the straight-line path from $\mathbf{S}$ to $\mathbf{S}'$ except for a measure-zero set (in $\mathbb{R}^1$), $r^*$ is twice-differentiable. To see this, the Alexandrov theorem says that since $r^*$ is convex, it is twice-differentiable everywhere except a measure-zero set in the space of its argument. However, by perturbing $\mathbf{S}$ and $\mathbf{S}'$ by a random matrix with eigenvalues distributed uniformly at random $\in [-\delta, \delta]$, for vanishing $\delta > 0$, with probability one the line between perturbed matrices only intersects the non-twice-differentiable set on a measure-zero set (this follows from the disintegration theorem). Thus, by continuity of $V^{r^*}$ in both arguments (since $\nabla r^*$ is Lipschitz by Lemma 15.3 of [Sha07], as $r^*$ is the dual of a strongly convex function), we assume $\mathbf{S}, \mathbf{S}'$ have this property, so we may write

$$V_{\mathbf{S}}^{r^*}(\mathbf{S}') = \int_0^1 \int_0^s \nabla^2 r^*(\mathbf{S}_t)[\mathbf{G},\mathbf{G}]dtds$$

$$\leq \int_0^1 \int_0^s \left\langle \nabla r^*(\mathbf{S}_t), \eta^2\mathbf{G}^2 \right\rangle dtds \leq \frac{1}{2} \int_0^1 \int_0^s \left\langle \nabla r^*(\mathbf{S}_t), \eta\mathbf{G} \right\rangle dtds. \qquad (28)$$

Here, for $t \in [0,1]$ we define $\mathbf{S}_t = \mathbf{S} + t\eta\mathbf{G}$, and used Lemma 15 in the second line (almost everywhere) as well as the assumed bound on $\|\eta\mathbf{G}\|_{\text{op}}$ so that $\eta^2\mathbf{G}^2 \preceq \frac{1}{2}\eta\mathbf{G}$. Define $p(t) := r^*(\mathbf{S}_t)$ and $v(t) := V_{\mathbf{S}}^{r^*}(\mathbf{S}_t)$; then,

$$\int_0^s \left\langle \nabla r^*(\mathbf{S}_t), \eta\mathbf{G} \right\rangle dt = p(s) - p(0) = v(s) + \langle \nabla r^*(\mathbf{S}), s\eta\mathbf{G}\rangle \leq v(1) + \langle \nabla r^*(\mathbf{S}), s\eta\mathbf{G}\rangle.$$

In the last inequality, we used that $v$ is increasing, which can be seen via

$$tv'(t) = \langle t\eta\mathbf{G}, \nabla r^*(\mathbf{S}_t) - \nabla r^*(\mathbf{S})\rangle \geq 0.$$

Substituting back into (28),

$$V_{\mathbf{S}}^{r^*}(\mathbf{S}') \leq \frac{1}{2}\int_0^1 \left(v(1) + \langle\nabla r^*(\mathbf{S}), s\eta\mathbf{G}\rangle\right)ds \leq \frac{1}{2}v(1) + \frac{1}{2}\langle\nabla r^*(\mathbf{S}), \eta\mathbf{G}\rangle.$$

Rearranging and using that $V_{\mathbf{S}}^{r^*}(\mathbf{S}') = v(1)$ yields the desired bound. $\qquad\square$

## 6.3 Refined $k$-PCA guarantees

We show a refined bound on the guarantees of simultaneous power iteration for approximately learning the top $k$ eigenvectors of a positive semidefinite matrix (i.e. $k$-PCA). In particular, the main result of this section (Proposition 7) strengthens Theorem 6.1 in [CMY20] by a factor of $k$.

---

**Algorithm 11** Power($\mathbf{A}, \lambda_{\max}, \lambda_{\min}, k, \epsilon, \delta$)

---

1: **Input:** Positive semidefinite $\mathbf{A} \in \mathbb{R}^{d\times d}$ with $\lambda_{\min}\mathbf{I} \preceq \mathbf{A} \preceq \lambda_{\max}\mathbf{I}$, accuracy $\epsilon \in (0, 1)$, $k \in [d]$, $\delta \in (0, 1)$
2: $N \leftarrow \Theta\left(\frac{1}{\epsilon}\log\left(\frac{d}{\delta\epsilon}\cdot\frac{\lambda_{\max}}{\lambda_{\min}}\right)\right)$ for a sufficiently large universal constant
3: $\mathbf{G} \in \mathbb{R}^{d\times k}$ entrywise $\sim \mathcal{N}(0, 1)$
4: **return** $\mathbf{V} \in \mathbb{R}^{d\times k}$, an orthonormal basis for the column span of $\mathbf{A}^N\mathbf{G}$

---

For the remainder of this section, we will fix a particular positive semidefinite matrix $\mathbf{A} = \mathbf{U}^\top\mathbf{diag}(\lambda)\mathbf{U}$, where $\mathbf{U} \in \mathbb{R}^{d\times d}$ is orthonormal and $\lambda_1 \geq \lambda_2 \geq \ldots \geq \lambda_d$ are the ordered eigenvalues of $\mathbf{A}$. We will also define three sets which partition $[d]$:

$$\begin{aligned}
L &:= \{j \in [d] \mid \lambda_j > (1 + \tfrac{\epsilon}{4})\lambda_{k+1}\}, \\
M &:= \{j \in [d] \mid (1 + \tfrac{\epsilon}{4})\lambda_{k+1} \geq \lambda_j \geq (1 - \tfrac{\epsilon}{4})\lambda_{k+1}\}, \\
S &:= \{j \in [d] \mid \lambda_j < (1 - \tfrac{\epsilon}{4})\lambda_{k+1}\}.
\end{aligned}\tag{29}$$

In particular, $L$, $M$, and $S$ are the "large", "medium", and "small" eigenvalues of $\mathbf{A}$. We first give two key structural results, which say that with high probability, the span of $\mathbf{V}$ contains essentially all the $\ell_2$ mass of any vector in $L$, and essentially none of the $\ell_2$ mass of any vector in $S$.

**Lemma 16.** *Let* $\mathbf{P} := \mathbf{V}\mathbf{V}^\top$ *where* $\mathbf{V}$ *is the output of Algorithm 11. With probability at least* $1 - \frac{\delta}{3} - \exp(-Ck)$ *for a universal constant* $C$, *for all* $j \in S$, $\|\mathbf{P}u_j\|_2 \leq \frac{\lambda_d^2}{\lambda_1^2}\cdot\frac{\epsilon^2}{64d^2}$, *where* $u_j$ *is row* $j$ *of* $\mathbf{U}$, *and we follow notation in* (29).

*Proof.* By rotational invariance of Gaussian matrices, it suffices to consider the case where $\mathbf{A}$ is diagonal and $\mathbf{U}$ is the identity; henceforth in this lemma, $u_j$ is the $j^{\text{th}}$ standard basis vector. Recall that $\mathbf{P}$ is the projection onto the column span of $\mathbf{A}^N\mathbf{G}$. We explicitly compute

$$\mathbf{P} = \mathbf{A}^N\mathbf{G}\left(\mathbf{G}^\top\mathbf{A}^{2N}\mathbf{G}\right)^{-1}\mathbf{G}^\top\mathbf{A}^N \implies \|\mathbf{P}u_j\|_2^2 = u_j^\top\mathbf{A}^N\mathbf{G}\left(\mathbf{G}^\top\mathbf{A}^{2N}\mathbf{G}\right)^{-1}\mathbf{G}^\top\mathbf{A}^Nu_j.\tag{30}$$

Here, we used that $\mathbf{P}^2 = \mathbf{P}$. Now, notice that (where $\mathbf{G}_{j:}$ is row $j$ of $\mathbf{G}$)

$$\mathbf{G}^\top\mathbf{A}^{2N}\mathbf{G} = \sum_{j\in[d]}\lambda_j^{2N}\mathbf{G}_{j:}\mathbf{G}_{j:}^\top \succeq \lambda_k^{2N}\sum_{j\in[k]}\mathbf{G}_{j:}\mathbf{G}_{j:}^\top.$$

However, Theorem 1.1 of [RV09] shows that with probability $\frac{\delta}{6} + \exp(-Ck)$ for some constant $C$, the smallest eigenvalue of a $k \times k$ Gram matrix for independent Gaussian entries is at least $\frac{\delta}{6\sqrt{k}}$. Assuming that this happens, we then continue to bound

$$\lambda_k^{2N} \sum_{j \in [k]} \mathbf{G}_{j:} \mathbf{G}_{j:}^\top \succeq \frac{\lambda_k^{2N} \delta}{6\sqrt{k}} \mathbf{I} \implies \|\mathbf{P} u_j\|_2^2 \leq \frac{6\sqrt{k}}{\delta \lambda_k^{2N}} u_j^\top \mathbf{A}^N \mathbf{G} \mathbf{G}^\top \mathbf{A}^N u_j$$

$$= \frac{6\sqrt{k}}{\delta} \left( \frac{\lambda_j}{\lambda_k} \right)^{2N} \left[ \mathbf{G} \mathbf{G}^\top \right]_{jj}$$

$$\leq \frac{6\sqrt{k}}{\delta} \exp\left( -\frac{\epsilon N}{2} \right) \sum_{i \in [k]} \mathbf{G}_{ji}^2.$$

In the first implication, we combined the lower bound we just derived with (30). Using standard chi-squared concentration bounds (cf. Lemma 1, [LM00]), the probability that $\sum_{i \in [k]} \mathbf{G}_{ji}^2 \geq 2k + 3 \log \frac{6}{\delta}$ is no more than $\frac{\delta}{6}$. Performing a union bound, with failure probability at most $\frac{\delta}{3} + \exp(-Ck)$, we have that for sufficiently large $N = \Theta\left( \frac{1}{\epsilon} \log \left( \frac{d}{\delta \epsilon} \cdot \frac{\lambda_{\max}}{\lambda_{\min}} \right) \right)$, since $k \leq d$,

$$\|\mathbf{P} u_j\|_2^2 \leq \frac{12 k^{1.5} + 18\sqrt{k} \log \frac{6}{\delta}}{\delta} \exp\left( -\frac{\epsilon N}{2} \right) \leq \frac{\lambda_{\min}^2}{\lambda_{\max}^2} \cdot \frac{\epsilon^2}{64 d^2} \leq \frac{\lambda_d^2}{\lambda_1^2} \cdot \frac{\epsilon^2}{64 d^2}.$$

Finally, adjusting the failure probability of the chi-squared tail bound by a factor of $d$, the conclusion follows by union bounding over all $j \in S$. $\qquad \square$

**Lemma 17.** *Let* $\mathbf{P} := \mathbf{V} \mathbf{V}^\top$ *where* $\mathbf{V}$ *is the output of Algorithm 11. With probability at least* $1 - \frac{\delta}{3} - d \exp(-Ck)$ *for a universal constant* $C$, *for all* $j \in L$, $\|\mathbf{P} u_j\|_2^2 \geq 1 - \frac{\lambda_d^2}{\lambda_1^2} \cdot \frac{\epsilon^2}{64 d^2}$, *where* $u_j$ *is row* $j$ *of* $\mathbf{U}$, *and we follow notation in* (29).

*Proof.* By definition of $L$, it is clear that $j \in [k]$. Again we consider the case where $\mathbf{A}$ is diagonal and $\mathbf{U}$ is the identity without loss of generality. Let $\mathbf{G}_{[k]:}$ be the first $k$ rows of $\mathbf{G}$, and let

$$\widetilde{\mathbf{G}} := \mathbf{G} \left( \mathbf{G}_{[k]:} \right)^{-1}.$$

Observe that the first $k$ rows of $\widetilde{\mathbf{G}}$ are exactly $\mathbf{I}$. Also, with probability at least $1 - \frac{\delta}{6} - \exp(-Ck)$, the largest singular value of $\left( \mathbf{G}_{[k]:} \right)^{-1}$ is bounded above by $\frac{6\sqrt{k}}{\delta}$, again by Theorem 1.1 of [RV09]. Condition on this event for the remainder of the proof. Since in this case $\mathbf{G}_{[k]:}$ is invertible, where Span denotes column span,

$$\mathrm{Span}\left( \widetilde{\mathbf{G}} \right) = \mathrm{Span}\left( \mathbf{G} \right) \implies \mathrm{Span}\left( \mathbf{A}^N \widetilde{\mathbf{G}} \right) = \mathrm{Span}\left( \mathbf{A}^N \mathbf{G} \right).$$

Fix some $j \in L$. To show the conclusion, it suffices to show that there exists a unit vector $v^*$ in the span of $\mathbf{A}^N \widetilde{\mathbf{G}}$ with $(\langle u_j, v^* \rangle)^2 \geq 1 - \frac{\lambda_d^2}{\lambda_1^2} \cdot \frac{\epsilon^2}{64 d^2}$. To see this, let $\{v_i\}_{i \in [k]}$ be any orthonormal basis for $\mathrm{Span}\left( \mathbf{A}^N \widetilde{\mathbf{G}} \right)$ with $v_1 = v^*$; then

$$\|\mathbf{P} u_j\|_2^2 = u_j^\top \mathbf{P} u_j = \sum_{i \in [k]} (\langle u_j, v_i \rangle)^2 \geq (\langle u_j, v^* \rangle)^2 \geq 1 - \frac{\lambda_d^2}{\lambda_1^2} \cdot \frac{\epsilon^2}{64 d^2}.$$

We will choose $v^*$ to be the normalization of $\mathbf{A}^N \widetilde{\mathbf{G}}_{:j}$ which has unit $\ell_2$ norm, where $\widetilde{\mathbf{G}}_{:j}$ is column $j$ of $\widetilde{\mathbf{G}}$. By standard chi-squared concentration bounds, with probability at least $1 - \frac{\delta}{6}$, all rows $i \notin [k]$ of the matrix $\mathbf{G}$ have squared $\ell_2$ norm at most

$$2k + 3 \log \frac{6d}{\delta}.$$

Here, we adjusted the failure probability of Lemma 1 in [LM00] by a factor of $d$ and union bounded over all $i \notin [k]$. Now, this implies that for all $i \notin [k]$,

$$\left\| \left( \left( \mathbf{G}_{[k]:} \right)^{-1} \right)^\top \mathbf{G}_{i:}^\top \right\|_2^2 \leq \frac{72k^2 + 108k \log \frac{6d}{\delta}}{\delta^2} \implies \widetilde{\mathbf{G}}_{ij}^2 \leq \frac{72k^2 + 108k \log \frac{6d}{\delta}}{\delta^2} \text{ for all } j \in [k].$$

We conclude that the column vector $\widetilde{\mathbf{G}}_{:j}$ has the property that

$$\widetilde{\mathbf{G}}_{ij}^2 \begin{cases} = 1 & i = j \\ = 0 & i \neq j,\ i \in [k] \\ \leq \frac{72k^2 + 108k \log \frac{6d}{\delta}}{\delta^2} & i \notin [k]. \end{cases}$$

Here, the first two cases are by design, and the last is by our earlier derivation. Thus, by choosing $N = \Theta\left( \frac{1}{\epsilon} \log \left( \frac{d}{\delta\epsilon} \cdot \frac{\lambda_{\max}}{\lambda_{\min}} \right) \right)$ to be sufficiently large (as in the ending of the proof of Lemma 16), we see that $\mathbf{A}^N \widetilde{\mathbf{G}}_{:j}$ places all but a negligible amount of $\ell_2^2$ mass on coordinate $j$, where we use that $\lambda_j^N \geq (1 + \frac{\epsilon}{4})^N \lambda_i^N$ for all $i \notin [k]$. □

We also give a simple helper calculation for demonstrating Loewner orderings.

**Lemma 18.** *Let $\mathbf{A}, \mathbf{B} \in \mathbb{R}^{d \times d}$ be positive semidefinite and suppose for any fixed unit test vector $v \in \mathbb{R}^d$ and some $\epsilon \in (0, 1)$,*

$$\left| v^\top (\mathbf{A} - \mathbf{B}) v \right| \leq \epsilon v^\top \mathbf{B} v.$$

*Then, $(1 - \epsilon)\mathbf{B} \preceq \mathbf{A} \preceq (1 + \epsilon)\mathbf{A}$.*

*Proof.* The upper bound follows from

$$v^\top \mathbf{A} v \leq v^\top \mathbf{B} v + \left| v^\top (\mathbf{A} - \mathbf{B}) v \right| \leq (1 + \epsilon) v^\top \mathbf{B} v.$$

The lower bound follows similarly. □

Our main bound follows from an application of the above three results.

**Proposition 7.** *Let $\mathbf{P} := \mathbf{V}\mathbf{V}^\top$ where $\mathbf{V}$ is the output of Algorithm 11. With probability at least $1 - \frac{2\delta}{3} - 2\exp(-Ck)$ for a universal constant $C$,*

$$(1 - \epsilon) \left( \mathbf{PAP} + (\mathbf{I} - \mathbf{P}) \mathbf{A} (\mathbf{I} - \mathbf{P}) \right) \preceq \mathbf{A} \preceq (1 + \epsilon) \left( \mathbf{PAP} + (\mathbf{I} - \mathbf{P}) \mathbf{A} (\mathbf{I} - \mathbf{P}) \right). \tag{31}$$

*Proof.* Condition on the conclusions of Lemmas 16 and 17 holding for the rest of this proof. We note

$$\mathbf{A} - \left( \mathbf{PAP} + (\mathbf{I} - \mathbf{P}) \mathbf{A} (\mathbf{I} - \mathbf{P}) \right) = \mathbf{PA} (\mathbf{I} - \mathbf{P}) + (\mathbf{I} - \mathbf{P}) \mathbf{AP}.$$

Hence, applying Lemma 18, for any fixed unit test vector $v \in \mathbb{R}^d$, this proposition asks to show

$$2 \left| y^\top \mathbf{A} x \right| \leq \epsilon \left( x^\top \mathbf{A} x + y^\top \mathbf{A} y \right), \text{ where } x := \mathbf{P} v \text{ and } y := (\mathbf{I} - \mathbf{P}) v.$$

Recall that $\mathbf{A} = \sum_{j \in [d]} \lambda_j u_j u_j^\top$. Letting $\tilde{x} := \mathbf{U} x$ and $\tilde{y} := \mathbf{U} y$, it suffices to show

$$\left| \sum_{j \in [d]} \lambda_j \tilde{x}_j \tilde{y}_j \right| \leq \frac{\epsilon}{2} \sum_{j \in [d]} \lambda_j \left( \tilde{x}_j^2 + \tilde{y}_j^2 \right). \tag{32}$$

Since $\langle \tilde{x}, \tilde{y} \rangle = \langle x, y \rangle = 0$ by the definition of $x, y$,

$$\left| \sum_{j \in [d]} \lambda_j \tilde{x}_j \tilde{y}_j \right| = \left| \sum_{j \in [d]} (\lambda_j - \lambda_{k+1}) \tilde{x}_j \tilde{y}_j \right| \leq \frac{\epsilon}{4} \sum_{j \in M} \lambda_{k+1} |\tilde{x}_j \tilde{y}_j| + \left| \sum_{j \notin M} (\lambda_j - \lambda_{k+1}) \tilde{x}_j \tilde{y}_j \right|.$$

Here we used the definition of $j \in M$, so that $|\lambda_j - \lambda_{k+1}| \leq \frac{\epsilon}{4} \lambda_{k+1}$. We first bound

$$\frac{\epsilon}{4} \sum_{j \in M} \lambda_{k+1} |\tilde{x}_j| |\tilde{y}_j| \leq \frac{\epsilon}{4(1 - \frac{\epsilon}{4})} \sum_{j \in M} \lambda_j |\tilde{x}_j| |\tilde{y}_j| \leq \frac{\epsilon}{4} \sum_{j \in M} \lambda_j \left( \tilde{x}_j^2 + \tilde{y}_j^2 \right) \leq \frac{\epsilon}{4} \sum_{j \in [d]} \lambda_j \left( \tilde{x}_j^2 + \tilde{y}_j^2 \right). \tag{33}$$

Moreover, by Lemma 16, for each $j \in S$, we have

$$|\tilde{x}_j| = \left| u_j^\top \mathbf{P} v \right| \leq \|\mathbf{P} u_j\|_2 \|v\|_2 \leq \frac{\lambda_d}{\lambda_1} \cdot \frac{\epsilon}{8d},$$

and similarly for each $j \in L$, $\tilde{y}_j \leq \frac{\lambda_d}{\lambda_1} \cdot \frac{\epsilon}{8d}$ by Lemma 17. Thus, since all $|\tilde{x}_j|$ and $|\tilde{y}_j|$ are at most 1,

$$\begin{aligned} \left| \sum_{j \in S} (\lambda_j - \lambda_{k+1}) \tilde{x}_j \tilde{y}_j \right| &\leq \lambda_1 \sum_{j \in S} |\tilde{x}_j| \leq \frac{\epsilon}{8} \lambda_d \leq \frac{\epsilon}{8} \sum_{j \in [d]} \lambda_j \left( \tilde{x}_j^2 + \tilde{y}_j^2 \right), \\ \left| \sum_{j \in L} (\lambda_j - \lambda_{k+1}) \tilde{x}_j \tilde{y}_j \right| &\leq \lambda_1 \sum_{j \in L} |\tilde{y}_j| \leq \frac{\epsilon}{8} \lambda_d \leq \frac{\epsilon}{8} \sum_{j \in [d]} \lambda_j \left( \tilde{x}_j^2 + \tilde{y}_j^2 \right). \end{aligned} \tag{34}$$

Finally, combining (33) and (34), we have the desired bound (32). $\qquad \square$

An unfortunate consequence of Proposition 7 is that its failure probability is exponentially related to $k$, rather than $d$. However, for sufficiently small $k = O(\log \frac{1}{\delta})$, we can use an alternative analysis of the power method due to [CMY20] to conclude that the desired bound (31) holds.

**Corollary 3.** *There is an algorithm (either Algorithm 11 of this paper, or Algorithm 5 of [CMY20]) which takes as input positive semidefinite $\mathbf{A} \in \mathbb{R}^{d \times d}$ with $\lambda_{\min} \mathbf{I} \preceq \mathbf{A} \preceq \lambda_{\max} \mathbf{I}$, $k \in [d]$, and accuracy parameter $\epsilon \in (0, 1)$, and returns with probability at least $1 - \delta$ a set of orthonormal vectors $\mathbf{V} \in \mathbb{R}^{d \times k}$ such that for $\mathbf{P} := \mathbf{V} \mathbf{V}^\top$, (31) holds. The number of matrix-vector products to $\mathbf{A}$ required is*

$$O \left( \frac{k}{\epsilon} \log^2 \left( \frac{d}{\delta \epsilon} \frac{\lambda_{\max}}{\lambda_{\min}} \right) \right).$$

*Proof.* In the case where $k \geq \frac{\log 6/\delta}{C}$ for $C$ the universal constant in Proposition 7, the conclusion

is immediate from Proposition 7. In the other case, we have that $k = O(\log \frac{1}{\delta})$. Hence, we can run Algorithm 5 of [CMY20] with an accuracy parameter which is $O(k)$ times smaller, and use their Theorem 6.1 to obtain the desired conclusion. The iteration complexity of Algorithm 5 of [CMY20] depends linearly on the inverse accuracy, so the bound loses an additional logarithmic factor. □

## 6.4 Implementation

In this section, we give an algorithm (Algorithm 12) which takes as input a matrix $\mathbf{S}$ and produces a matrix $\widehat{\mathbf{Y}}$ such that for some choice of input $\Delta \geq 0$, we have

$$\left\| \widehat{\mathbf{Y}} - \nabla r^*(\mathbf{S}) \right\|_{\mathrm{tr}} \leq k\Delta. \tag{35}$$

We will use this at the end of the section to give a complete (computationally efficient) implementation of an approximate variant of Algorithm 10, and give its guarantees as Corollary 4.

---

**Algorithm 12** ApproxProject$(\mathbf{S}, \lambda_{\max}, \lambda_{\min}, k, \Delta, \delta)$

---

1: **Input:** Positive semidefinite $\mathbf{S} = \mathbf{M}^\top \mathbf{M} \in \mathbb{R}^{d \times d}$ for some explicitly given $\mathbf{M} \in \mathbb{R}^{n \times d}$ with $\lambda_{\min} \mathbf{I} \preceq \mathbf{S} \preceq \lambda_{\max} \mathbf{I}$, $k \leq d \leq n$, accuracy $\Delta \in (0, 1)$, $k \in [d]$, $\delta \in (0, 1)$
2: **Output:** $\widehat{\mathbf{Y}}$ satisfying $\left\| \widehat{\mathbf{Y}} - \nabla r^*(\mathbf{S}) \right\|_{\mathrm{tr}} \leq k\Delta$ with probability $\geq 1 - \delta$
3: $\mathbf{V} \leftarrow \mathsf{Power}(\mathbf{S}, \lambda_{\max}, \lambda_{\min}, k, \frac{\Delta}{8\lambda_{\max}}, \frac{\delta}{2})$ (or when $k \leq \frac{\log 12/\delta}{C}$, use Algorithm 5 of [CMY20])
4: $\{u_j\}_{j \in [k]} \leftarrow$ eigenvectors of $\mathbf{V}^\top \mathbf{M}^\top \mathbf{M} \mathbf{V} \in \mathbb{R}^{k \times k}$, left-multiplied by $\mathbf{V}$
5: For $j \in [k]$, $\tilde{\lambda}_j \leftarrow u_j^\top \mathbf{P} \mathbf{S} \mathbf{P} u_j$ where $\mathbf{P} := \mathbf{V}\mathbf{V}^\top$
6: $\widehat{\mathbf{S}} \leftarrow \sum_{j \in [k]} \tilde{\lambda}_j u_j u_j^\top + (1 - \frac{\Delta}{4\lambda_{\max}})(\mathbf{I} - \mathbf{P})\mathbf{S}(\mathbf{I} - \mathbf{P})$
7: $\widehat{T} \leftarrow (1 \pm \frac{\Delta}{8})$-approximation to $\mathrm{Tr} \exp\left( (1 - \frac{\Delta}{4\lambda_{\max}})(\mathbf{I} - \mathbf{P})\mathbf{S}(\mathbf{I} - \mathbf{P}) \right)$ with probability $1 - \frac{\delta}{2}$
8: $\widehat{\tau} \leftarrow$ fixed point of $k \exp(\widehat{\tau}) = \sum_{j \in [k]} \exp(\min(\widehat{\tau}, \tilde{\lambda}_j)) + \widehat{T}$
9: $\widehat{\mathbf{Y}} \leftarrow \frac{k}{\sum_{j \in [k]} \exp(\min(\widehat{\tau}, \tilde{\lambda}_j)) + \widehat{T}} \left( \sum_{j \in [k]} \exp(\min(\widehat{\tau}, \tilde{\lambda}_j)) u_j u_j^\top + \exp\left( (1 - \frac{\Delta}{4\lambda_{\max}})(\mathbf{I} - \mathbf{P})\mathbf{S}(\mathbf{I} - \mathbf{P}) \right) \right)$
10: **return** $\widehat{\mathbf{Y}}$

---

**Proposition 8.** *With probability at least $1 - \delta$, the output $\widehat{\mathbf{Y}}$ of Algorithm 12 satisfies (35). The complexity of Lines 3-8 of the algorithm is*

$$O\left( ndk \cdot \frac{\lambda_{\max}}{\Delta^2} \log^2\left( \frac{d}{\Delta\delta} \frac{\lambda_{\max}}{\lambda_{\min}} \right) \right).$$

*Moreover, for any $\epsilon \in (0, 1)$, the complexity of providing $(1 \pm \epsilon)$-approximate access to quadratic forms through $\widehat{\mathbf{Y}}$ for any $n$ fixed vectors $\{v_i\}_{i \in [n]} \subset \mathbb{R}^d$ with probability at least $1 - \delta$ is*

$$O\left( nd \cdot \frac{\lambda_{\max}}{\epsilon^2} \log\left( \frac{1}{\epsilon} \right) \log\left( \frac{nd}{\delta} \right) \right).$$

*Proof.* We will show correctness and complexity of Algorithm 12 separately.

*Correctness guarantee.* We begin with proving correctness, which we complete in two parts. In particular, we show that the following two bounds hold:

$$\left\| \nabla r^*(\mathbf{S}) - \nabla r^*(\widehat{\mathbf{S}}) \right\|_{\mathrm{tr}} \leq \frac{k\Delta}{2}, \quad \left\| \widehat{\mathbf{Y}} - \nabla r^*(\widehat{\mathbf{S}}) \right\|_{\mathrm{tr}} \leq \frac{k\Delta}{2}. \tag{36}$$

By combining the two parts of (36) and applying the triangle inequality, we have the desired conclusion. To show the former bound, because the convex conjugate of any $\frac{1}{k}$-strongly convex function in $\|\cdot\|_{\mathrm{tr}}$ is $k$-smooth in $\|\cdot\|_{\mathrm{op}}$ (cf. Lemma 15.3, [Sha07]), and Fact 3 states that $r$ is strongly convex, it suffices to show that

$$\left\|\widehat{\mathbf{S}} - \mathbf{S}\right\|_{\mathrm{op}} \leq \frac{\Delta}{2} \implies \left\|\nabla r^*(\widehat{\mathbf{S}}) - \nabla r^*(\mathbf{S})\right\|_{\mathrm{tr}} \leq k \left\|\widehat{\mathbf{S}} - \mathbf{S}\right\|_{\mathrm{op}} \leq \frac{k\Delta}{2}. \tag{37}$$

Assume first that Power was used in computing Line 3. By Proposition 7, we have that

$$\|\mathbf{S} - (\mathbf{PSP} + (\mathbf{I} - \mathbf{P})\,\mathbf{S}\,(\mathbf{I} - \mathbf{P}))\|_{\mathrm{op}} \leq \frac{\Delta}{8\lambda_{\max}} \|\mathbf{S}\|_{\mathrm{op}} \leq \frac{\Delta}{8}. \tag{38}$$

Next, we claim that $\{u_j\}_{j \in [k]}$ are the eigenvectors of $\mathbf{PSP}$, so that

$$\sum_{j \in [k]} \tilde{\lambda}_j u_j u_j^\top = \mathbf{PSP}.$$

To see this, let $w_j \in \mathbb{R}^k$ be an eigenvector of $\mathbf{V}^\top \mathbf{M}^\top \mathbf{M} \mathbf{V}$ with eigenvalue $\tilde{\lambda}_j$, and let $u_j = \mathbf{V} w_j$, as in Line 4 of Algorithm 12. Then indeed we have (since $\mathbf{V}^\top \mathbf{V}$ is the identity)

$$\mathbf{PSP} u_j = \mathbf{V} \mathbf{V}^\top \mathbf{S} \mathbf{V} \mathbf{V}^\top \mathbf{V} w_j = \mathbf{V}\left(\mathbf{V}^\top \mathbf{M}^\top \mathbf{M} \mathbf{V} w_j\right) = \tilde{\lambda}_j \mathbf{V} w_j = \tilde{\lambda}_j u_j.$$

We then compute, using the definition of $\widehat{\mathbf{S}}$ in Line 6,

$$\left\|(\mathbf{PSP} + (\mathbf{I} - \mathbf{P})\,\mathbf{S}\,(\mathbf{I} - \mathbf{P})) - \widehat{\mathbf{S}}\right\|_{\mathrm{op}} = \frac{\Delta}{4\lambda_{\max}} \|(\mathbf{I} - \mathbf{P})\mathbf{S}(\mathbf{I} - \mathbf{P})\|_{\mathrm{op}} \leq \frac{3\Delta}{8\lambda_{\max}} \|\mathbf{S}\|_{\mathrm{op}} \leq \frac{3\Delta}{8}. \tag{39}$$

Here, we used Proposition 7 once more to (loosely) upper bound $(\mathbf{I} - \mathbf{P})\mathbf{S}(\mathbf{I} - \mathbf{P})$ by $1.5\mathbf{S}$. Combining (37), (38), and (39) gives the first conclusion in (36).

We next claim the top eigenvalue of $(\mathbf{I} - \mathbf{P})\mathbf{S}(\mathbf{I} - \mathbf{P})$ is at most $(1 + \frac{\Delta}{4\lambda_{\max}})\tilde{\lambda}_k$; this follows from the second and third parts of Theorem 1 of [MM15]. Thus, by scaling down $(\mathbf{I} - \mathbf{P})\mathbf{S}(\mathbf{I} - \mathbf{P})$ by a factor $1 - \frac{\Delta}{4\lambda_{\max}}$, we have that its largest eigenvalue is smaller than the smallest of $\mathbf{PSP}$. Let $\{\tilde{\lambda}_j\}_{j \in [d] \setminus [k]}$, $\{u_j\}_{j \in [d] \setminus [k]}$ complete an eigendecomposition of $\widehat{\mathbf{S}}$. We conclude that none of the eigenvalues of $\widehat{\mathbf{S}} - \mathbf{PSP}$ will be truncated in the projection since they are not in the top $k$, so Fact 4 yields that

$$\nabla r^*(\widehat{\mathbf{S}}) = \frac{k}{\sum_{j \in [k]} \min(\sigma, \alpha_j) + T}\left(\sum_{j \in [k]} \min(\sigma, \alpha_j) u_j u_j^\top + \sum_{j \notin [k]} \alpha_j u_j u_j^\top\right),$$
$$\text{where } T := \mathrm{Tr}\exp\left(\left(1 - \frac{\Delta}{4\lambda_{\max}}\right)(\mathbf{I} - \mathbf{P})\mathbf{S}(\mathbf{I} - \mathbf{P})\right),$$
$$\text{and } \sigma := \exp(\tau(\tilde{\lambda})), \ \alpha_j := \exp(\tilde{\lambda}_j) \text{ for all } j \in [d].$$

Specifically, this form is clear for the first $k$ eigenvectors, and for the remainder the $\nabla r^*$ operation applies an exponentiation and scaling (since they will not be truncated), which does not affect the relevant basis. Finally, by applying the following Lemma 19 with $\gamma = \frac{\Delta}{6}$, and using that the eigenvectors of our returned $\widehat{\mathbf{Y}}$ align exactly with those of $\widehat{\mathbf{S}}$, we have the desired second bound in (36). We remark that the only place we used the fact that Power was used in Line 3 thus far in this proof was in citing Theorem 1 of [MM15]; however, if Algorithm 5 of [CMY20] is used, a similar

statement on the top eigenvalue of $\widehat{\mathbf{S}} - \mathbf{PSP}$ follows by their Remark 6.9.

*Complexity guarantee.* When $\mathbf{S}$ is given in the form $\mathbf{M}^\top \mathbf{M}$, the cost of a matrix-vector product in $\mathbf{S}$ is $O(nd)$. So, from Corollary 4 the cost of Line 3 is bounded by

$$O\left(ndk \cdot \frac{\lambda_{\max}}{\Delta} \log^2\left(\frac{d}{\Delta\delta} \frac{\lambda_{\max}}{\lambda_{\min}}\right)\right).$$

In Line 4, the cost of forming the matrix $\mathbf{MV}$ is $O(ndk)$, and forming its Gram matrix and performing an eigendecomposition takes time $O(k^2 d + k^3)$; left-multiplying all resulting vectors by $\mathbf{V}$ also takes time $O(k^2 d)$. The cost of Line 5 for each $j \in [k]$ is $O(nd + kd)$, so the overall cost is also $O(ndk)$. Line 8 is a scalar optimization problem and will not dominate the complexity (tolerance to error in a binary search is guaranteed via Lemma 19). The only remaining cost is in Line 6.

To estimate $\operatorname{Tr}\exp(\mathbf{A})$ to $1 \pm \gamma$ accuracy for a positive semidefinite matrix $\mathbf{0} \preceq \mathbf{A} \preceq \lambda_{\max}\mathbf{I}$ (here, we note Proposition 7 guarantees $\mathbf{A} = (1 - \frac{\Delta}{4\lambda_{\max}})(\mathbf{I} - \mathbf{P})\mathbf{S}(\mathbf{I} - \mathbf{P}) \preceq \mathbf{S} \preceq \lambda_{\max}\mathbf{I}$), we will use two facts well-known in the approximate semidefinite programming literature. First, Theorem 4.1 of [SV14] shows that a degree-$O(\lambda_{\max} \log \frac{1}{\gamma})$ polynomial $p$ has the property that

$$\left(1 - \frac{\gamma}{3}\right) \exp(\mathbf{A}) \preceq p(\mathbf{A}) \preceq \left(1 + \frac{\gamma}{3}\right) \exp(\mathbf{A}).$$

Moreover, the Johnson-Lindenstrauss lemma (e.g. the implementation given in [Ach03]) shows that to estimate $\operatorname{Tr}\exp(\mathbf{A})$ it suffices to sample a random $\pm\frac{1}{\sqrt{r}}$ matrix $\mathbf{G} \in \mathbb{R}^{d \times r}$ for some $r = O(\log(\frac{d}{\delta})\gamma^{-2})$ and then compute

$$\sum_{j \in [r]} \left\| p\left(\frac{1}{2}\mathbf{A}\right) \mathbf{G}_{:j} \right\|_2^2 \approx_{1\pm\frac{\gamma}{3}} \sum_{j \in [r]} \left\| \exp\left(\frac{1}{2}\mathbf{A}\right) \mathbf{G}_{:j} \right\|_2^2 = \operatorname{Tr}\left(\exp\left(\frac{1}{2}\mathbf{A}\right) \mathbf{GG}^\top \exp\left(\frac{1}{2}\mathbf{A}\right)\right).$$

This last quantity is a $1 \pm \frac{\gamma}{3}$ approximation of $\operatorname{Tr}\exp(\mathbf{A})$ with probability $1 - \frac{\delta}{2}$. The cost of this whole procedure is dominated by $O(r\lambda_{\max} \log \frac{1}{\gamma})$ matrix-vector multiplies to $\mathbf{A}$; for our choice of $\mathbf{A}$, each multiplication costs $O(nd)$ time, leading to an overall complexity of (as $\gamma = \Theta(\Delta)$)

$$O\left(nd \cdot \frac{\lambda_{\max}}{\Delta^2} \log\left(\frac{1}{\Delta}\right) \log\left(\frac{d}{\delta}\right)\right).$$

For any $v \in \mathbb{R}^d$, essentially the same strategy of sampling a random $\mathbf{G} \in \mathbb{R}^{d \times r}$ and computing

$$v^\top p\left(\frac{1}{2}\mathbf{A}\right) \mathbf{GG}^\top p\left(\frac{1}{2}\mathbf{A}\right) v = \sum_{j \in [r]} \left(\left\langle \mathbf{G}_{:j}, p\left(\frac{1}{2}\mathbf{A}\right) v \right\rangle\right)^2.$$

suffices for estimating the quadratic form in $\exp(\mathbf{A})$ to $1 \pm \epsilon$ accuracy, where now $r$ and the polynomial degree depend on $\epsilon$ rather than $\gamma$. We can first apply the polynomial to each column of $\mathbf{G}$ and then compute inner products with $v$. To compute approximate quadratic forms in $\widehat{\mathbf{Y}}$, every part of $\widehat{\mathbf{Y}}$ is explicitly given except for the component $\exp(\mathbf{A})$ for $\mathbf{A} = (1 - \frac{\Delta}{4\lambda_{\max}})(\mathbf{I} - \mathbf{P})\mathbf{S}(\mathbf{I} - \mathbf{P})$, which we can approximate with the above strategy in the desired time.

Finally, for a batch of $n$ vectors $\{v_i\}_{i \in [n]}$, note that we can first compute all the vectors $p(\frac{1}{2}\mathbf{A})\mathbf{G}_{:j}$ in the desired time, at which point the cost of computing each quadratic form is reduced to $O(dr)$. Thus, the overall complexity is $O(ndr)$, where we adjust the logarithm in the definition of $r$ by a factor of $n$ to union bound the failure probability. □

We now provide the helper Lemma 19, which we remark crucially improves the error analysis in Section 7.3 of [CMY20] by a factor of $k$, allowing us to avoid an additional $\text{poly}(k)$ dependence.

**Lemma 19.** *Let $\gamma \in (0, 1)$, $k \in [d]$. Given nonnegative $\{\alpha_j\}_{j \in [d]}$ sorted with $\alpha_1 \geq \ldots \geq \alpha_d$, let $T := \sum_{j \notin [k]} \alpha_j$, and let $\widehat{T} \in [(1 - \gamma)T, (1 + \gamma)T]$. Define $\sigma$ and $\widehat{\sigma}$ to be fixed points of*

$$k\sigma = \sum_{j \in [k]} \min(\sigma, \alpha_j) + T, \; k\widehat{\sigma} = \sum_{j \in [k]} \min(\widehat{\sigma}, \alpha_j) + \widehat{T}.$$

*Then, we have*

$$\sum_{j \in [d]} \left| \frac{k \min(\sigma, \alpha_j)}{\sum_{i \in [k]} \min(\sigma, \alpha_i) + T} - \frac{k \min(\widehat{\sigma}, \alpha_j)}{\sum_{i \in [k]} \min(\widehat{\sigma}, \alpha_i) + \widehat{T}} \right| \leq 3k\gamma.$$

*Proof.* We first comment briefly on the existence of $\sigma, \widehat{\sigma}$. Note that in the setting of the lemma,

$$f(\widehat{\sigma}) := \frac{\widehat{\sigma}}{\sum_{i \in [k]} \min(\widehat{\sigma}, \alpha_i) + \widehat{T}}$$

is an increasing, continuous function of $\widehat{\sigma}$ in the range $[0, \infty)$ which satisfies $f(0) = 0$ and $f(\infty) = \infty$, so there must be a unique $\widehat{\sigma}$ satisfying $f(\widehat{\sigma}) = \frac{1}{k}$. Existence of $\sigma$ is proven similarly. Next, we claim

$$\widehat{\sigma} \in [(1 - \gamma)\sigma, (1 + \gamma)\sigma]. \tag{40}$$

By our earlier argument, it suffices to show that $f((1 + \gamma)\sigma) \geq \frac{1}{k}$ and $f((1 - \gamma)\sigma) \leq \frac{1}{k}$, so that an appeal to continuity and monotonicity of $f$ yields (40). To see the former bound, note that

$$\begin{aligned} f((1 + \gamma)\sigma) &= \frac{(1 + \gamma)\sigma}{\sum_{i \in [k]} \min((1 + \gamma)\sigma, \alpha_i) + \widehat{T}} \\ &\geq \frac{(1 + \gamma)\sigma}{(1 + \gamma) \sum_{i \in [k]} \min(\sigma, \alpha_i) + (1 + \gamma)T} \\ &= \frac{\sigma}{\sum_{i \in [k]} \min(\sigma, \alpha_i) + T} = \frac{1}{k}. \end{aligned}$$

The last equality used the definition of $\sigma$; the only inequality used $\widehat{T} \leq (1 + \gamma)T$ by assumption, and $\min((1 + \gamma)\sigma, \alpha_i) \leq \min((1 + \gamma)\sigma, (1 + \gamma)\alpha_i) = (1 + \gamma) \min(\sigma, \alpha_i)$. Similarly,

$$\begin{aligned} f((1 - \gamma)\sigma) &= \frac{(1 - \gamma)\sigma}{\sum_{i \in [k]} \min((1 - \gamma)\sigma, \alpha_i) + \widehat{T}} \\ &\leq \frac{(1 - \gamma)\sigma}{(1 - \gamma) \sum_{i \in [k]} \min(\sigma, \alpha_i) + (1 - \gamma)T} = \frac{1}{k}. \end{aligned}$$

Here we used $(1 - \gamma) \min(\sigma, \alpha_i) \leq \min((1 - \gamma)\sigma, \alpha_i)$. Now, we claim (40) implies that for all $j \in [d]$,

$$\left| \frac{\min(\sigma, \alpha_j)}{\sum_{i \in [k]} \min(\sigma, \alpha_i) + T} - \frac{\min(\widehat{\sigma}, \alpha_j)}{\sum_{i \in [k]} \min(\widehat{\sigma}, \alpha_i) + \widehat{T}} \right| \leq \frac{3\gamma \min(\sigma, \alpha_j)}{\sum_{i \in [k]} \min(\sigma, \alpha_i) + T}. \tag{41}$$

To see this, we may upper and lower bound for each $j \in [d]$,

$$(1 - \gamma) \min(\sigma, \alpha_j) \leq \min(\widehat{\sigma}, \alpha_j) \leq (1 + \gamma) \min(\sigma, \alpha_j)$$

$$\implies (1 - \gamma) \left( \sum_{i \in [k]} \min(\sigma, \alpha_i) + T \right) \leq \sum_{i \in [k]} \min(\widehat{\sigma}, \alpha_i) + \widehat{T} \leq (1 + \gamma) \left( \sum_{i \in [k]} \min(\sigma, \alpha_i) + T \right) \quad (42)$$

$$\implies \frac{(1 - 3\gamma) \min(\sigma, \alpha_j)}{\sum_{i \in [k]} \min(\sigma, \alpha_i) + T} \leq \frac{\min(\widehat{\sigma}, \alpha_j)}{\sum_{i \in [k]} \min(\widehat{\sigma}, \alpha_i) + \widehat{T}} \leq \frac{(1 + 3\gamma) \min(\sigma, \alpha_j)}{\sum_{i \in [k]} \min(\sigma, \alpha_i) + T}.$$

This yields (41), which upon summing and using that $\min(\sigma, \alpha_j) = \alpha_j$ for all $j \notin [k]$, and the definition of $T$, yields the final conclusion after multiplying by $k$. $\qquad \square$

We additionally state one helper guarantee on the properties of $\widehat{\mathbf{Y}}$.

**Lemma 20.** *With probability at least $1 - \delta$, the output $\widehat{\mathbf{Y}}$ of Algorithm 12 satisfies*

$$\left\| \widehat{\mathbf{Y}} \right\|_{\mathrm{tr}} \leq \left( 1 + \frac{\Delta}{2} \right) k, \ \left\| \widehat{\mathbf{Y}} \right\|_{\mathrm{op}} \leq 1 + \frac{\Delta}{2}.$$

*Proof.* By the definition of $\nabla r^*$, we have that $\nabla r^*(\widehat{\mathbf{S}}) \in \mathcal{Y}$ so has operator norm at most 1 and trace norm at most $k$. For the first conclusion, (36) implies

$$\left\| \widehat{\mathbf{Y}} \right\|_{\mathrm{tr}} \leq \left\| \nabla r^*(\widehat{\mathbf{S}}) \right\|_{\mathrm{tr}} + \frac{k\Delta}{2}.$$

For the second conclusion, (42) and the operator norm bound on $\mathcal{Y}$ imply

$$\lambda_{\max} \left( \widehat{\mathbf{Y}} \right) \leq (1 + 3\gamma) \lambda_{\max} \left( \nabla r^*(\widehat{\mathbf{S}}) \right) \leq 1 + \frac{\Delta}{2}.$$

$\qquad \square$

Finally, we state our complete algorithm, an approximate version of the KFMMW method.

---

**Algorithm 13** ApproxKFMMW$(k, \{\mathbf{G}_t\}_{0 \leq t \leq T}, \eta, \Delta, \delta)$

---

1: **Input:** Gain matrices $\{\mathbf{G}_t\}_{0 \leq t < T}$, step size $\eta > 0$, accuracy $\Delta \in (0, 1)$, $\delta \in (0, 1)$
2: $\mathbf{Y}_0 \leftarrow \frac{k}{d}\mathbf{I}$, $\mathbf{S}_0 \leftarrow \nabla r(\mathbf{Y}_0) = \log(\frac{k}{d})\mathbf{I}$
3: **for** $0 \leq t < T$ **do**
4: $\quad \mathbf{S}_{t+1} \leftarrow \mathbf{S}_t + \eta \mathbf{G}_t$
5: $\quad \widehat{\mathbf{Y}}_{t+1} \leftarrow$ ApproxProject$(\mathbf{S}_{t+1} + (1 + \log(\frac{d}{k}))\mathbf{I}, t + 2, 1, k, \Delta, \frac{\delta}{T})$
6: **end for**

---

**Corollary 4.** *Suppose the input gain matrices to Algorithm 13 satisfy the bound, for all $t \geq 0$,*

$$\mathbf{0} \preceq \eta \mathbf{G}_t \preceq \frac{1}{2}\mathbf{I}.$$

*Further, suppose that the $\{\mathbf{G}_t\}_{t \geq 0}$ are weakly decreasing in Loewner order. With probability $1 - \delta$,*

$$\|\mathbf{G}_T\|_k \leq \frac{2}{T} \sum_{t=0}^{T-1} \left\langle \mathbf{G}_t, \widehat{\mathbf{Y}}_t \right\rangle + \frac{k \log d}{\eta T} + \frac{k\Delta}{\eta}.$$

*The complexity of Algorithm 13 is*

$$O\left(ndk \cdot \frac{T^2}{\Delta^2} \log^2\left(\frac{dT}{\Delta\delta}\right)\right),$$

*and the cost of providing $(1 \pm \epsilon)$-approximate access to quadratic forms for any fixed $n$ vectors $\{v_i\}_{i \in [n]} \subset \mathbb{R}^d$ through any $\widehat{\mathbf{Y}}_t$ for any $\epsilon \in (0, 1)$, with failure probability at most $\delta$, is*

$$O\left(nd \cdot \frac{T}{\epsilon} \log\left(\frac{1}{\epsilon}\right) \log\left(\frac{nd}{\delta}\right)\right).$$

*Proof.* We claim first that it suffices to show that the conclusion of Proposition 8 holds in each iteration $t$. To see why this is enough, matrix Hölder on the conclusion of Proposition 6 yields

$$\langle \mathbf{G}_t, \mathbf{Y}_t \rangle \leq \left\langle \mathbf{G}_t, \widehat{\mathbf{Y}}_t \right\rangle + \|\mathbf{G}_t\|_{\mathrm{op}} \left\|\widehat{\mathbf{Y}}_t - \mathbf{Y}_t\right\|_{\mathrm{tr}} \leq \left\langle \mathbf{G}_t, \widehat{\mathbf{Y}}_t \right\rangle + k\Delta \|\mathbf{G}_t\|_{\mathrm{op}} \leq \left\langle \mathbf{G}_t, \widehat{\mathbf{Y}}_t \right\rangle + \frac{k\Delta}{2\eta}$$

$$\implies \frac{1}{T} \sum_{t=0}^{T-1} \langle \mathbf{G}_t, \mathbf{U} \rangle \leq \frac{2}{T} \sum_{t=0}^{T-1} \left\langle \mathbf{G}_t, \widehat{\mathbf{Y}}_t \right\rangle + \frac{k \log d}{\eta T} + \frac{k\Delta}{\eta}.$$

In the last inequality in the first line, we used the assumption that $\eta\mathbf{G} \preceq \frac{1}{2}\mathbf{I}$. Supremizing over $\mathbf{U}$, and using monotonicity of the gain matrices, yields the conclusion. Next, we prove that calling ApproxProject is valid. Note $\nabla r^*$ is invariant under shifts by the identity, so it suffices to first shift $\mathbf{S}_0$ to be $\mathbf{I}$ and hence $\lambda_{\min} = 1$ is a valid bound. Since for all $t \geq 0$, the change in $\mathbf{S}_t$ (i.e. $\eta\mathbf{G}_t$) is positive semidefinite and bounded by $\mathbf{I}$, we can set $\lambda_{\max} = t + 2$ in the call. Finally, the failure probability comes from a union bound over all iterations, and the overall complexity is $T$ times the cost of a single ApproxProject operation, given by Proposition 8. $\square$

## Acknowledgments

We thank Morris Yau for clarifying conversations about the prior work [CMY20]. Ilias Diakonikolas is supported by NSF Award CCF-1652862 (CAREER), a Sloan Research Fellowship, and a DARPA Learning with Less Labels (LwLL) grant. Daniel Kane is supported by NSF CAREER Award ID 1553288 and a Sloan fellowship. Kevin Tian is supported by NSF CAREER Award CCF-1844855 and NSF Grant CCF-1955039.

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

# A  List-decodable mean estimation for $\alpha^{-1} = \Omega(d)$

We give a simple algorithm for list-decodable mean estimation in the regime $\alpha^{-1} = \Omega(d)$.

---

**Algorithm 14** SamplePostProcess$(T, \delta)$

---

1: **Input:** $T \subset \mathbb{R}^d$ with $|T| = n$ satisfying Assumption 1, $\alpha \leq \frac{1}{Cd}$ for a universal constant $C$, $\delta \in (0, 1)$
2: **Output:** $L \subset \mathbb{R}^d$ with $|L| \leq \frac{3}{\alpha}$ satisfying (8) with probability $\geq 1 - \delta$
3: $N \leftarrow \left\lceil \frac{36 \log(2/\delta)}{\alpha} \right\rceil$
4: $\widetilde{L} \leftarrow \{X_i\}_{i \in [N]}$, where each $X_i$ is an independent uniform sample from $T$
5: $\mathbf{G} \in \mathbb{R}^{d \times c} \leftarrow$ entrywise $\pm \frac{1}{\sqrt{c}}$ uniformly at random, for $c = \Theta(\log(\frac{1}{\alpha\delta}))$
6: Let $L$ be a maximal subset of $\widetilde{L}$ such that for each $X_i \in L$, $\left\| \mathbf{G}^\top (X_i - X_j) \right\|_2^2 \leq 8.8d$ for at least $\frac{\alpha N}{3}$ of the $X_j \in \widetilde{L}$, and $\left\| \mathbf{G}^\top (X_i - X_j) \right\|_2^2 \geq 35.2d, \forall X_j \in L$
7: **return** $L$

---

**Proposition 9.** *Algorithm 14,* SamplePostProcess, *meets its output specifications in runtime*

$$O\left( \frac{1}{\alpha^2} \log^4 \left( \frac{1}{\alpha\delta} \right) \right).$$

*Proof.* It is straightforward by Assumption 1 (cf. correctness proof of Theorem 3) that at least $\frac{\alpha n}{2}$ of the points $X_i \in T$ satisfy

$$\|X_i - \mu^*\|_2^2 \leq 2d. \tag{43}$$

For each $i \in [N]$ indexing the set $\widetilde{L}$, let $E_i$ be the event that $X_i$ satisfies the bound (43); each of these events is an independent Bernoulli variable with mean at least $\frac{\alpha}{2}$. Thus, by applying a Chernoff bound, with probability at least $1 - \frac{\delta}{2}$, at least $\frac{\alpha N}{3}$ of the points in $\widetilde{L}$ satisfy (43). Next, by the Johnson-Lindenstrauss lemma of [Ach03], for a sufficiently large dimensionality $c$, with probability at least $1 - \frac{\delta}{2}$, all of the $\left\| \mathbf{G}^\top (X_i - X_j) \right\|_2^2$ are within a 1.1 factor of the corresponding $\|X_i - X_j\|_2^2$. Condition on both of these events for the remainder of the proof.

By definition of the greedy process in Line 6, we have the output size guarantee, since each element of $\widetilde{L}$ is associated with a (disjoint) cluster of $\frac{\alpha N}{3}$ points, by the separation property. So, for correctness, it suffices to prove that (8) is met for a universal constant (depending on $C$). Call $\widetilde{S}$ the set of points in $T$ satisfying (43). If any point in $\widetilde{S}$ is chosen in $L$, then indeed

$$\|X_i - \mu^*\|_2^2 \leq 2d \leq \frac{2}{C\alpha},$$

so (8) is met with constant $\sqrt{\frac{2}{C}}$. Further, observe that the only thing preventing any point in $\widetilde{S}$ from being chosen is the separation condition for $L$. This is because by triangle inequality and the definition (43), any pair of points $X_i, X_j \in \widetilde{S}$ satisfies $\|X_i - X_j\|_2^2 \leq 8d$, so after multiplication by $\mathbf{G}^\top$ they pass the clustering requirement. Thus, suppose no point in $\widetilde{S}$ is in $L$. For any $X_i \in \widetilde{S} \cup \widetilde{L}$, this implies there exists a $X_j \in \widetilde{L}$ with

$$\left\| \mathbf{G}^\top (X_i - X_j) \right\|_2^2 \leq 35.2d \implies \|X_i - X_j\|_2^2 \leq 40d.$$

By triangle inequality, this implies that (8) is met with constant $\sqrt{\frac{84}{C}}$, via

$$\|X_j - \mu^*\|_2^2 \leq 84d \leq \frac{84}{C\alpha}.$$

Finally, the runtime is dominated by the cost of multiplying all points in $\widetilde{L}$ by $\mathbf{G}^\top$, and performing all pairwise distance comparisons of the $\{\mathbf{G}^\top X_i\}_{i \in [N]}$. Both of these fit in the allotted time budget. $\qquad\square$

We make a final remark that up to logarithmic factors, the runtime in Proposition 9 is not larger than $\frac{nd}{\alpha}$ asymptotically, since we take sample size $n \geq \alpha^{-1}$. Thus, in the regime $\alpha^{-1} = \Omega(d)$, we obtain the correct list size and error bound up to constants, in time $\widetilde{O}(\frac{nd}{\alpha})$ as desired.

## B    Runtime of [CMY20]

For notational convenience in this section, we denote $k := \alpha^{-1}$. We give a brief discussion of the dependence on $k$ in the runtime of [CMY20], as it is not explicitly stated there.

**Cluster removal: $O(k)$ overhead.** At a high level, the [CMY20] algorithm is composed of an "outer loop" which is repeated $O(k)$ times. Each iteration of the outer loop removes roughly an $\alpha$ fraction of the overall weight, and this could occur $O(k)$ times.

**Ky Fan positive SDP: $\widetilde{O}(k^2)$ overhead.** Each run of the outer loop is composed of polylogarithmically many iterations which decrease a particular potential function. The potential function used is the objective value of a Ky Fan norm positive SDP over a truncated simplex. Each iteration of the outer loop run is dominated by the cost of approximating the positive SDP. The statement of the SDP solver, Algorithm 3 of [CMY20], shows that the solver takes $\widetilde{O}(k^2)$ iterations.

**Approximate Bregman projections: $\widetilde{O}(k^3)$ overhead.** To implement iterations of the SDP solver, [CMY20] apply approximate Bregman projections based on simultaneous power iteration, similar to the ones we develop in Section 6. However, their analysis was loose in terms of the accuracy needed for the simultaneous power iteration. The two places this is most apparent are:

1. Theorem 6.1 of [CMY20] loses a factor of $k$ when compared to Proposition 7.

2. Lemma 7.11 of [CMY20] loses a factor of $k$ when compared to Lemma 19 (note that the statement of our lemma is scaled up by $k$).

Under looser analyses, the cost of each projection step is dominated by the cost of computing the trace product of an approximate matrix exponential and the empirical covariance. Because of the extra $k$ factor in Lemma 7.11 of [CMY20], the multiplicative accuracy of matrix exponential-vector products must be on the order of $\frac{1}{k}$. The form of the approximate exponential is essentially the same as that in Line 9 of Algorithm 12, so following the strategy of Proposition 8, it suffices to implement $O(k)$ (corresponding to the degree of a Taylor expansion) matrix-vector multiplies in a matrix, each of which costs $O(nd)$ to apply. This matrix exponential-vector product is applied to $\widetilde{O}(k^2)$ vectors, via the Johnson-Lindenstrauss lemma for the higher accuracy threshold.

In summary, we calculate the dependence on $k$ to be roughly $k^C$ for $C \geq 6$ in [CMY20]. We remark a $k^2$ factor can be saved in the Bregman projection step by simply swapping in our more fine-grained analysis, so the cost of each projection is $\widetilde{O}(ndk)$, leading to an overall $k^4$ dependence. However (as discussed in the introduction), the presence of $k$-dimensional operations and a clustering outer loop suggests that this approach is likely to depend at least quadratically on $k$.