# OpenReview forum: "List-Decodable Mean Estimation in Nearly-PCA Time"
_NeurIPS.cc/2021/Conference — NeurIPS 2021 Spotlight_

### Official Review · Reviewer_Zu6e · 2021-07-15

**Rating:** 7
**Confidence:** 3

**Summary:**

This article is abound list-decodable learning in which we consider the mean estimation problem when more than half of the data are corrupted data. In this framework, the mean cannot be recovered exactly but one can obtain a list of $k$ potential mean and say that the mean of the inliers is among these $k$ values. This article exhibit an algorithm that attain optimal statistical bounds (up to log factors) in a linear running time (linear in the sample size but also the dimension and in $k$).

**Limitations And Societal Impact:**

The article would greatly benefit from some practical experiments. The theory is stated up to a constant and in practice if the constant in the big O are large, then even though the algorithm is nearly PCA time, it may not be computable. Please provide a proof of concept, even on a corrupted mixture of spherical Gaussian this would already be a good illustration to show that the algorithm work in practice.

**Main Review:**

The article is quite significant because it exhibit a considerable faster algorithm than previous works for the same optimal error guarantees.  The article is clear and the quality of the redaction is ok.

Remark : I would ask the authors to be more mindful of the abbreviations they use. Their are a lot of abbreviations that are not transparent, for instance the notation PCA is never made clear in  the article : what algorithm does it represent exactly ?  Also SIFT is not transparent and in fact there already exist a SIFT algorithm in image processing so this might not be the best choice of name for the algorithm. Similarly the reader also have to remember what is MMW...

**Time Spent Reviewing:**

2

---

> ### Author Response · Authors · 2021-08-11
> **Response to Reviewer Zu6e**
>
> Thank you for your careful reading, and your feedback regarding the exposition. We will make sure to be more explicit in a revision in terms of the abbreviations we use, especially those which are less well-known such as MMW. We will also add a pointer to the existence of the prior algorithm named SIFT in the image processing literature; thank you for this observation. To address your question regarding PCA, every time we use this abbreviation in our article we mean principal component analysis, a technique for identifying maximum variance directions.
>
> We addressed your concerns regarding practicality in a meta-comment. We hope you found our discussion clarifying, and that it elevates your view of our paper. Thanks once again for your reviewing efforts.

---

> > ### Comment · Reviewer_Zu6e · 2021-08-23
> > **Acknowledgement of Authors' responses**
> >
> > Thank you for the answers.
> >
> > Ok regarding the names of the algorithms..
> >
> > I will not increase my score because given the complexity of the proposed algorithm I am really curious whether it works in practice or not. Even in toy datasets, complexity bounds are given but no practical running time on laptop for instance. This is my main concern about this article and even though this is a theoretical article, the main advantage of this article is the low complexity of the proposed algorithm hence for the article to rate higher, the reader must be convinced that the algorithm is really practically usable.

---

### Official Review · Reviewer_1cxi · 2021-07-15

**Rating:** 10
**Confidence:** 5

**Summary:**

This paper studies the fundamental task of list-decodable mean estimation in high dimensional space. This paper proposes a novel algorithm for bounded covariance distributions with optimal sample complexity and near-optimal error guarantee, running time in nearly-PCA time.


**Limitations And Societal Impact:**

Limitations are not mentioned. However, this paper is a super fancy theory paper, in order to make it more practical might take more efforts. Societal impact is not mentioned and since this paper is pure theory, it does not have a direct societal impact.

**Main Review:**

Strength: This paper improved all the previous results (e.g. [Charikar, Steinhardt, Valiant, STOC’17], [Diakonikolas, Kane, Kongsgaard NeurIPS’20], [Cheapanamjeri, Mohanty, Yau
FOCS’20]) running time. This paper is a truly breakthrough result.

More impressively, this paper proposed a super fancy technique called Subspace Isotropic FilTering
(SIFT) . To the best of my knowledge, I have never seen such a thing before, this is completely new.

Clearly, this paper should be at least an oral presentation at NeurIPS.

Minor:

In Reference, [AK+05], should be [AK05], the et al should be removed. This might be a bug of google scholar citation. Clearly, this paper only has two authors.

In Reference, the [DKK20] conference information is missing. It is a NeurIPS 2020 paper.

In Reference, the [LY20] conference information is missing. It is a NeurIPS 2020 paper.

In Reference, the [JLT20] conference information is missing, it is a NeurIPS 2020 paper.

This paper mentioned SDP in a number of places. Several recent, state-of-art SDP solvers should be cited ``A Faster Interior Point Method for Semidefinite Programming’’ and “Solving Tall Dense SDPs in the Current Matrix Multiplication Time”.


**Time Spent Reviewing:**

15

---

> ### Author Response · Authors · 2021-08-11
> **Response to Reviewer 1cxi**
>
> Thank you very much for your encouraging feedback; we are happy to hear that you found our result a breakthrough, and that you found our techniques interesting.
>
> We will be sure to incorporate your feedback regarding changes in our citations, as well as the suggested pointers to the SDP literature; we agree these are relevant to our method.

---

### Official Review · Reviewer_F5h5 · 2021-07-16

**Rating:** 8
**Confidence:** 3

**Summary:**

This paper studies the list-decodable mean estimation problem and proposes an algorithm that runs in nearly $k$-PCA time, which can be seen as a natural barrier to this problem.

**Limitations And Societal Impact:**

See the "Main Review".

**Main Review:**

This paper studies the list-decodable mean estimation problem: that is, if only $\frac{1}{\alpha}$-fraction of the data is the good point, we want to output a list of $O(\frac{1}{\alpha})$ solutions, one of which is guaranteed to be close to the true mean of the distribution. This problem can be seen as an extended to the outlier robust mean estimation problem and has received significant attention in recent years.
In this work, the authors give a new algorithm for the bounded covariance distribution that runs in $\tilde{O}(\frac{nd}{\alpha})$ time, which nearly matches the time to perform $k$-PCA on the data, which can be seen a natural bottleneck of this problem. This result is better than the previous one in [CMY20](FOCS 2020)(runs in time $\tilde{O}(\frac{nd}{\alpha^C})$ where $C \ge 6$) and only increase the error by logarithmic factor.

This paper first gives a simple version of the algorithm that runs in $\tilde{O}(\frac{n^2 d}{\alpha})$ but is conceptually new. Then, following the result in [DHL19], the author uses the matrix multiplicative weights regret minimization framework to reduce the runtime to $\tilde{O}(\frac{nd}{\alpha})$. One technical contribution here is to give a Ky Fan k-norm generalization of MMW.

This paper is technically very strong and the writing is well. The authors provide a lot of intuition for the proofs. This paper is a pure theory paper. However, a set of accompanying experiment results would be also nice.

Some minor questions:

- For the second "=" in the equation which starts in line 211, is it should be "$\le$" instead of "="?




**Time Spent Reviewing:**

8-10

---

> ### Author Response · Authors · 2021-08-11
> **Response to Reviewer F5h5**
>
> Thank you for your thoughtful comments; we are glad you enjoyed the writing of the paper, and found the intuition we provided for our proofs useful. We also would like to point out (in reference to the “Main Review” feedback) that in addition to our result which improves [CMY20] by a factor of at least $\alpha^{-5}$ and only slightly loses in the estimation error, we provide an alternative algorithm which improves [CMY20] by a factor of at least $\min(\alpha^{-5}, nd)$ which matches the estimation error up to at most a constant (see Theorem 4 in the supplement).
>
> Additionally, thank you for your clarifying question regarding Line 211. This equality is indeed intended to be an equality, see Fact 2 in the supplement for the derivation.

---

### Author Response · Authors · 2021-08-11
**Meta-comment: experimental evaluation**

Thanks very much to the reviewers for the helpful feedback. We would like to address the feedback regarding the inclusion of experiments here, which was mentioned by all reviewers.

Our main contribution is a theoretical proof-of-concept of the practical tractability of the fundamental problem we study, by giving both improved runtimes and new, arguably simpler, techniques decoupling filtering and clustering. Our conceptual approach is new, and our runtimes at minimum remove overheads of roughly $n/\alpha$ or $\alpha^{-5}$ from prior bounds. We aim for these results to serve as a first step towards practical solutions to list-decodable mean estimation. Our new baseline algorithm, SIFT (Theorem 2 in the submission) is indeed only a few lines of code, and is the simplest algorithm we are aware of for solving our problem. Our algorithm with the strongest theoretical guarantees (Theorem 1 in the submission), shows how to use semidefinite programming to improve the theoretical performance of SIFT, an idea which has already experimentally yielded benefits in the minority-outlier regime (see [DHL19]).

We strongly agree that developing practical list-decodable mean estimators is an important undertaking, and plan to do so in follow-up work. However, this is likely an endeavor requiring substantial care in its own right, and thus we believe it is outside the focus of our submission.

---

### Decision · Program_Chairs · 2021-09-27

**Decision:**

Accept (Spotlight)

**Comment:**

This paper gives a nearly "PCA" time algorithm for the problem of list-decodable mean estimation. There are two relevant parameters in the problem: the underlying dimension d, the number of samples n, and the fraction of inliers \alpha. There's a significant history of works on this problem -- starting with polynomial (but large exponent) running time in both d and 1/\alpha. A significant leap was made in a recent work that gave linear time algorithm in nd but a large polynomial (1/\alpha^5) dependence on \alpha. This paper resolves the issue completely and gives an algorithm with a quasi-linear dependence in nd/\alpha.

List-decodable mean estimation is a fundamental question in algorithmic robust statistics (it generalizes mean estimation and spherical mixture clustering problems) and finding efficient algorithms for the problem is an important research direction. This paper makes a foundational contribution to this area by giving a nearly optimal algorithm for the problem. While there were concerns of whether the nearly optimal running time in this paper translates into practical gains, the authors' response convincingly clarified that a provably linear time algorithm paves the way for a practically fast one.

We recommend acceptance.